JCB — Journal of Cell Biology

# The β-cell primary cilium is an autonomous Ca²⁺ compartment for paracrine GABA signaling

Gonzalo Manuel Sanchez[1]*, Tugce Ceren Incedal[1]*, Juan Prada[2], Paul O'Callaghan[1], Oleg Dyachok[1], Santiago Echeverry[1], Özge Dumral[1], Phuoc My Nguyen[1], Beichen Xie[1], Sebastian Barg[1], Johan Kreuger[1], Thomas Dandekar[2], and Olof Idevall-Hagren[1]

**The primary cilium is an organelle present in most adult mammalian cells that is considered as an antenna for sensing the local microenvironment. Here, we use intact mouse pancreatic islets of Langerhans to investigate signaling properties of the primary cilium in insulin-secreting β-cells. We find that GABA$_{B1}$ receptors are strongly enriched at the base of the cilium, but are mobilized to more distal locations upon agonist binding. Using cilia-targeted Ca²⁺ indicators, we find that activation of GABA$_{B1}$ receptors induces selective Ca²⁺ influx into primary cilia through a mechanism that requires voltage-dependent Ca²⁺ channel activation. Islet β-cells utilize cytosolic Ca²⁺ increases as the main trigger for insulin secretion, yet we find that increases in cytosolic Ca²⁺ fail to propagate into the cilium, and that this isolation is largely due to enhanced Ca²⁺ extrusion in the cilium. Our work reveals local GABA action on primary cilia that involves Ca²⁺ influx and depends on restricted Ca²⁺ diffusion between the cilium and cytosol.**

## Introduction

The primary cilium is a specialized signaling organelle interfacing the cell and its environment and with the capability of translating multimodal inputs and prompting appropriate cellular responses. The presence of specific ciliary receptors and effector proteins enables the generation of local ciliary signals and provides a platform for selective crosstalk between different pathways, with Ca²⁺ and cAMP being the most prominent second messengers in this organelle (Phua et al., 2015; Schou et al., 2015; Singla and Reiter, 2006; Wachten and Mick, 2021). Ca²⁺ in particular has been associated with different ciliary responses, but how ciliary Ca²⁺ signaling is protected from cytoplasmic interference and what events take place upstream of the second messenger are not well understood. The concentration of Ca²⁺ in the cilium has been reported to be higher than in the cytosol due to constitutive influx (Delling et al., 2013). However, such an arrangement is not readily reconcilable with traditional Ca²⁺ signaling downstream of cell surface receptors, and it would also greatly limit the number of Ca²⁺ effectors that can be used to decode the Ca²⁺ signals, since many have affinities for Ca²⁺ in the nanomolar range. Ca²⁺ has also been proposed to freely diffuse between the cilium and cytosol (Delling et al., 2013; Su et al., 2013), an arrangement that is difficult to reconcile with a specific role of Ca²⁺ in the cilium, since that would require a mechanism for distinguishing Ca²⁺ of cytosolic and ciliary origin. Many G protein–coupled receptors (GPCRs) localize to primary cilia,

and the current model for signal modulation postulates that GPCRs are removed from the cilia upon agonist activation for signal termination (Garcia et al., 2018). However, most of the details concerning ciliary signaling downstream of receptor activation remain elusive. In vertebrates, Hedgehog signaling relies on the cilium and, despite the complexity of the Hedgehog pathway, it is perhaps the best-known example of ciliary signal transduction. The Hedgehog receptor Patched is mainly localized to the ciliary membrane and its activation results in exit from the cilium, allowing entry of Smoothened and subsequent activation of the pathway (Ho and Stearns, 2021).

Pancreatic β-cells are the only source of insulin, a major anabolic hormone important for maintaining blood glucose homeostasis. Failure to produce or secrete insulin results in elevated blood glucose concentrations and contribute to the development of diabetes (Ashcroft and Rorsman, 2012). β-cells are located within islets of Langerhans, which are richly vascularized endocrine cell clusters dispersed throughout the pancreas. Each β-cell is equipped with a primary cilium that protrudes from the apical side facing away from the islet vasculature (Gan et al., 2017; Müller et al., 2021). β-cell-specific loss of primary cilia is associated with impaired cell function, but the signaling downstream of this organelle is still poorly understood (Gerdes et al., 2014; Hughes et al., 2020; Volta et al., 2019). In particular, a role of the β-cell cilia as an antenna sensing the islet microenvironment has not been demonstrated.

[1]Department of Medical Cell Biology, Uppsala University, Uppsala, Sweden;   [2]Department of Bioinformatics, University of Würzburg, Würzburg, Germany.

*G.M. Sanchez and T.C. Incedal contributed equally to this paper.   Correspondence to Olof Idevall-Hagren: olof.idevall@mcb.uu.se.

γ-Aminobutyric acid (GABA) is mainly regarded as the main inhibitory neurotransmitter in the brain even though it is an evolutionarily ancient signaling molecule and a variety of actions outside of the central nervous system have been documented (Gamlin et al., 2018). In the endocrine pancreas, for instance, GABA has both acute and long-term effects, including modulation of insulin secretion and maintenance of β-cell identity and function (Korol et al., 2018; Wang et al., 2019). β-cells produce and release GABA that acts locally within the pancreatic islets of Langerhans, though the mechanisms behind its signaling are still not well understood (Menegaz et al., 2019).

In the present work, we used intact endocrine pancreatic islets of Langerhans to measure ciliary $Ca^{2+}$ dynamics within these functional micro-organs. Contrary to the vast majority of studies on the primary cilium in immortalized cell lines, the current work enabled the study of cilia signaling in a preparation that preserves both tissue architecture and function. Using this model, we unveiled cilia-specific $Ca^{2+}$ activity driven, at least in part, by activation of metabotropic GABA receptors with a dependence on $Ca^{2+}$ extrusion, which isolates the cilium from contaminating cytosolic $Ca^{2+}$. We describe here for the first time a cilia-localized class C GPCR, the $GABA_{B1}$ receptor, and show that it exhibits a distinct distribution with enrichment toward the cilia base. We also show that agonist stimulation mobilizes receptors to more distal parts of the cilium and triggers ciliary $Ca^{2+}$ transients in a putative non-canonical fashion.

## Results

### $GABA_{B1}$ receptors localize to the primary cilium of β-cells and are mobilized upon agonist binding

β-cells express functional ionotropic ($GABA_A$) and metabotropic ($GABA_{B1/2}$) GABA receptors, but the localization of these receptors within β-cells of intact pancreatic islets of Langerhans has not been determined. Since GABA can reach the islets both through the circulation and through local release from β-cells, receptor segregation might be a mechanism by which the islet cells can control the response depending on the source of GABA. Immunostainings of mouse islets showed the expression of all three GABA receptors. Whereas $GABA_A$ and $GABA_{B2}$ receptors displayed diffuse localization with enrichment in cytoplasmic vesicles, $GABA_{B1}$ receptors instead localized to a single, small, elongated compartment with an average length of 2 μm and a width of <400 nm as determined by stimulated emission depletion (STED) imaging (Fig. 1, A and E). Co-immunostaining for markers of the ciliary axoneme (acetylated tubulin and Arl13b) and the centrosome (γ-tubulin) showed that $GABA_{B1}$ receptors were confined to the base of this structure and localized peri-axonemally, consistent with membrane localization (Fig. 1, B–E, and Fig. S1). The length of the $GABA_{B1}$ receptor positive segment did not strongly correlate with the length of the cilium, suggesting that it might correspond to a functional ciliary compartment (Fig. 1, F–I). The localization of $GABA_{B1}$ receptor to the cilia base was observed in islet β-cells positive for insulin, but also in other cell types of the islet. Clonal MIN6 β-cells, either grown as a monolayer or aggregated into islet-like clusters (pseudoislets), also presented with cilia enriched for $GABA_{B1}$

receptors, but in contrast to islets the receptors were not restricted to the cilium base but occupied the entire cilium membrane (Fig. 1, J and K). Human islet cells cilia were also enriched for $GABA_{B1}$ receptors (Fig. 1, J and K). We confirmed the localization of $GABA_{B1}$ receptors to the cilium using a different antibody targeting an extracellular epitope, and also the specificity of the antibodies by siRNA-mediated knockdown of $GABA_{B1}$ expression in MIN6 cells (Fig. S1). The localization of ciliary receptors is known to be modulated by agonist binding. We therefore determined the distribution of $GABA_{B1}$ receptors in the cilia of islet cells following 10 min exposure to 10 nM GABA or 10 nM of the $GABA_{B1}$ receptor-specific agonist baclofen. In both cases, $GABA_{B1}$ receptors were mobilized from the cilia base to more distal locations within the cilium while the length of the cilium was unaffected. STED imaging further revealed distinct clusters of $GABA_{B1}$ receptors all along the cilium following agonist addition (Fig. 1 L and Fig. S1). These results show that $GABA_{B1}$ receptors form a dynamic pool of ciliary receptors in β-cells. To test whether activation of $GABA_{B1}$ receptors might influence the distribution of other ciliary receptors, we determined the localization of the Hedgehog receptor Patched under resting conditions and following exposure to 100 nM GABA for 10 min. The addition of GABA caused an enrichment of Patched relative to unstimulated controls, indicating putative cross-talk between ciliary $GABA_{B1}$ receptors and Patched (Fig. 1 M).

### The primary cilium of intact islets of Langerhans

The primary cilium has recently been proposed to be an important structure in the regulation of islet cell function (Gerdes et al., 2014; Hughes et al., 2020; Kluth et al., 2019). To better understand the putative importance of ciliary $GABA_{B1}$ receptors, we decided to characterize the primary cilia in our isolated mouse islet preparations. We found that most islet cells were equipped with a single cilium, which averaged 8 μm in length (Fig. 2, A–C, and I). Similar observations were also made in pancreatic sections stained for the cilia marker acetylated tubulin and the islet hormones insulin (β-cell) and glucagon (α-cell; Fig. 2 D). The cilia displayed a non-random localization within the islets and were often found in clusters, sometimes even forming cilia–cilia contacts (Fig. 2, E–G). We occasionally observed cells with two primary cilia, and around 30% of all cilia presented with a bulging tip (Fig. 2, G–I). Cilia with bulging tips may represent cilia with active signaling, and consistent with this we observed mobilization of $GABA_{B1}$ receptors into more distal segments of these cilia even in the absence of an exogenous agonist (Fig. 2, G and I). Besides $GABA_{B1}$ receptors, the islet cell cilia contain the intraciliary transport protein IFT88, as well as the well-characterized ciliary receptors Smoothened, Patched, and Somatostatin Receptor 3 (Fig. 2 J). Notably, the distribution of these receptors was more homogenous than that of the $GABA_{B1}$ receptor, and they were found all along the cilium of isolated islet cells (Fig. 2 K). Acetylated tubulin immunostaining of human islets revealed that their cells also have primary cilia, although the fraction of ciliated cells was lower and the overall background from cytosolic acetylated tubulin was much more pronounced (Fig. 2 L).

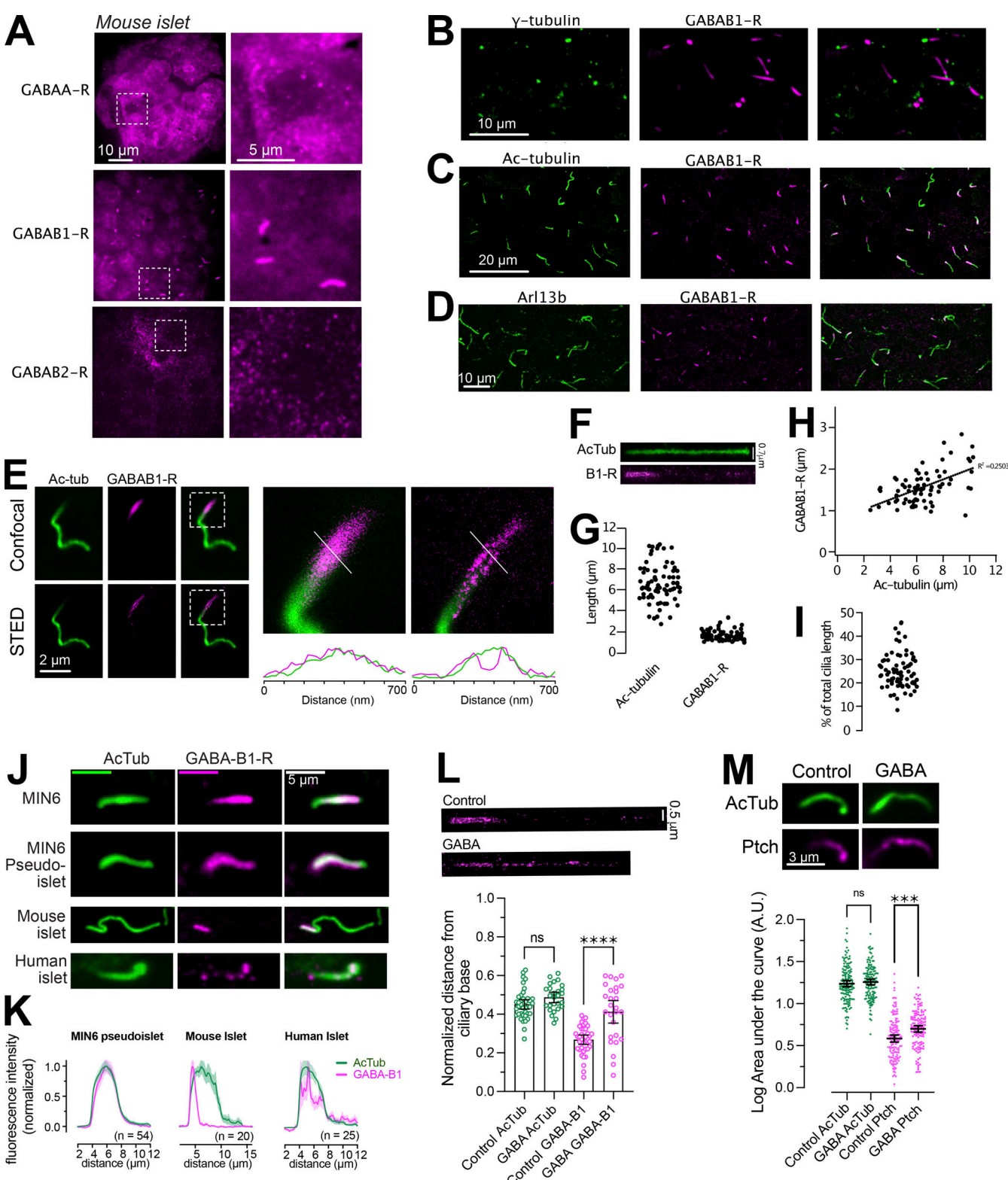

Figure 1. **GABA_B1 receptors localize to the primary cilium of islet cells and are mobilized by agonist binding. (A)** Immunofluorescence staining of a mouse islet showing the distribution of GABA_A, GABA_B1, and GABA_B2 receptors. **(B)** Immunofluorescence staining of a mouse islet showing γ-tubulin in green and GABA_B1 receptors in magenta. **(C)** Immunofluorescence staining of a mouse islet showing acetylated tubulin in green and GABA_B1 receptors in magenta. **(D)** Immunofluorescence staining of a mouse islet showing Arl13b in green and GABA_B1 receptors in magenta. **(E)** Confocal and STED images of a mouse islet cell cilia stained for acetylated tubulin (green) and GABA_B1 receptors (magenta). Boxed areas are magnified to the right with line profiles of the dotted lines shown below. Notice that the receptors localize peripheral to the microtubule core of the cilium. **(F)** Distribution of GABA_B1 receptors along a straightened primary cilium. **(G)** Distribution of whole cilia lengths (determined by acetylated tubulin) and the GABA_B1 receptor positive segment of the cilium in mouse islets ($n$ = 85 cilia from 4 islets). **(H)** Correlation between cilia length and the length of the GABA_B1 receptor positive segment. **(I)** Fraction of the cilium positive for

GABA$_{B1}$ receptors. **(J)** Representative images showing the cilia localization of GABA$_{B1}$ receptors (magenta) to the primary cilium (green) in MIN6 cells, MIN6 pseudo-islets, mouse islet cells, and human islet cells. **(K)** Quantifications of the cilia distribution of GABA$_{B1}$ receptors in the different cell preparations. **(L)** Center-of-mass analysis of mouse islet cilia stained for acetylated tubulin and GABA$_{B1}$ receptors under control conditions and following a 10-min incubation with 10 nM GABA. Notice that the addition of GABA results in a shift in the center of mass toward more distal regions of the cilium (means ± SEM; $n$ = 39, control and $n$ = 27, GABA cilia from at least 25 islets each, **** $P < 0.0001$, one-way ANOVA with Tukey's post hoc test). **(M)** Immunofluorescence images of cilia from mouse islet cells immunostained for acetylated tubulin (green) and Patched (magenta) under control conditions and following 10-min stimulation with 100 nM GABA. Quantifications below show the area under the curve (AUC) of line profiles drawn along the cilia of islet cells (means ± SEM; $n_{control}$ = 145 cilia from 15 islets from 3 animals; $n_{GABA}$ = 138 cilia from 15 islets from 3 animals). *** $P < 0.0001$, one-way ANOVA with Tukey's post hoc test.

## GABA$_{B1}$ receptor activation initiates cilia Ca$^{2+}$ signaling

Signal transduction downstream of GABA receptors typically act through cAMP- and Ca$^{2+}$-dependent pathways. Both cAMP and Ca$^{2+}$ are important second messengers for propagation of cilia signaling as well as for stimulation of insulin secretion. To measure cilia Ca$^{2+}$ concentration changes, we developed a cilia-targeted Ca$^{2+}$ biosensor based on the cilia-enriched receptor Smoothened, and delivered it to islets through adenoviral transduction (Fig. 3 A). More specifically, we fused the Ca$^{2+}$ indicator GCaMP5G and the reference fluorophore mCherry to the C-terminus of Smoothened (Smo-GCaMP5G-mCh). To evaluate the properties of this Ca$^{2+}$-indicator, we permeabilized clonal MIN6 β-cells expressing Smo-GCaMP5G-mCh and superfused the cells with Ca$^{2+}$-buffers while recording GCaMP5G and mCherry fluorescence using total internal reflection fluorescence (TIRF) microscopy. This technique enabled simple access to primary cilia protruding from the cell surface and also extended recordings of cilia Ca$^{2+}$ due to low photobleaching. We observed stepwise increases in ciliary Ca$^{2+}$ concentration, with half-maximal and maximal effect at 0.6 and 3 μM, respectively (Fig. 3, B and C). Although the majority of Smo-GCaMP5G-mCh was localized to the cilium, small amounts of the protein were also found in the plasma membrane, which enabled direct comparisons of GCaMP5G fluorescence change in the two compartments. Importantly, the indicator localization did not affect the response to Ca$^{2+}$, and the observed EC$_{50}$ was also similar to that reported previously for soluble GCaMP5G (Akerboom et al., 2012). The expression of Smo-GCaMP5G-mCh resulted in a 50% increase in cilia length and also increased the fraction of cells presenting with dilated tips (Fig. 3 D and Fig. S2), likely indicating exacerbated Hedgehog signaling when over-expressing Smoothened. Importantly, the overexpression Smo-GCaMP5G-mCh did not influence the distribution of endogenous GABA$_{B1}$ receptors (Fig. S2). Photolysis of caged Ca$^{2+}$ in the cytosol of individual MIN6 β-cells expressing Smo-GCaMP5G-mCh resulted in a Ca$^{2+}$ wave that propagated into the cilium, and we did not notice any correlation between indicator expression level and the kinetics of the Ca$^{2+}$ response in the cilium, suggesting that the expression of the indicator has negligible buffering effect (Fig. S3). We also performed photobleaching experiments to determine the mobility of the indicator in the ciliary and plasma membranes, and found that the indicator was essentially immobile in both compartments for the duration of the experiment (160 s; Fig. S3). Ca$^{2+}$ is considered an important second messenger in the cilium despite the fact that its actual role is still debated and poorly understood (Delling et al., 2016; Jin et al., 2014). One key aspect is how the resting concentration in the cilium compares to that

of the cytoplasm, and estimations based on electrophysiological and imaging approaches have shown that the ciliary Ca$^{2+}$ concentration is higher than that of the cytosol (Delling et al., 2013). We evaluated the relative concentrations in the two cellular compartments by comparing the average ratios of fluorescence line profiles drawn over the cilium and part of plasma membrane, as visualized by TIRF microscopy, in both mouse and human islet β-cells and in clonal MIN6 β-cells expressing Smo-GCaMP5G-mCh (Fig. 3, E–G). In contrast to previous studies, we found that the GCaMP5G-to-mCherry ratios in the cilia were either the same, or slightly lower, than those of the cytosol, indicating a lower resting Ca$^{2+}$ concentration in the cilia. Control experiments showed that this is likely not due to quenching of the pH-sensitive GCaMP5G, since a cilia-targeted pH indicator revealed that the cilia lumen is slightly more alkaline than the cytosol (Fig. S3).

Next, we tested if activation of GABA receptors could initiate Ca$^{2+}$ signaling in the primary cilium. The addition of GABA at low concentrations (1–10 nM) initiated ciliary Ca$^{2+}$ signaling in the form of distinct Ca$^{2+}$ spikes without accompanying changes in the cytosolic Ca$^{2+}$ concentration in MIN6 β-cells expressing Smo-GCaMP5G-mCh (Fig. 4, A–C). Baclofen, a selective agonist for metabotropic GABA$_B$ receptors, had a similar effect as GABA and siRNA-mediated knockdown of GABA$_{B1}$ receptors in MIN6 cell pseudoislets almost completely suppressed the response to baclofen, indicating that the evoked responses were triggered by activation of GABA$_B$ receptors (Fig. 4, D and E). We next tested the effect of GABAergic activation on cilia Ca$^{2+}$ signaling in mouse islets. To quantify ciliary Ca$^{2+}$ activity, we counted the number of events (defined as an increase in GCaMP5G fluorescence above 25% of resting levels) and calculated their density over defined periods of time (Fig. 4 E). Baclofen increased ciliary Ca$^{2+}$ activity and vigabatrin, which blocks GABA catabolism and promotes endogenous GABA accumulation and release, had a similar effect (Fig. 4, F–H).

## Cyclic nucleotide-gated ion channels and voltage-dependent Ca$^{2+}$ channels promote Ca$^{2+}$ influx in islet cell primary cilia

The cilium is a specialized compartment for cyclic nucleotide signaling, and changes in both cAMP and cGMP are known to modulate Ca$^{2+}$ signaling in β-cells. To test whether the cilia Ca$^{2+}$ flashes depend on cyclic nucleotides, we established conditions where we increase the concentration of either cAMP or cGMP, and determined the impact on cilia Ca$^{2+}$ signaling. To measure intracellular cAMP and cGMP, we expressed two recently developed Förster Resonance Energy Transfer (FRET) sensors based on Epac1 (Klarenbeek et al., 2015) and Protein kinase G (PKG; Calamera et al., 2019), respectively. The latter one (PfPKG)

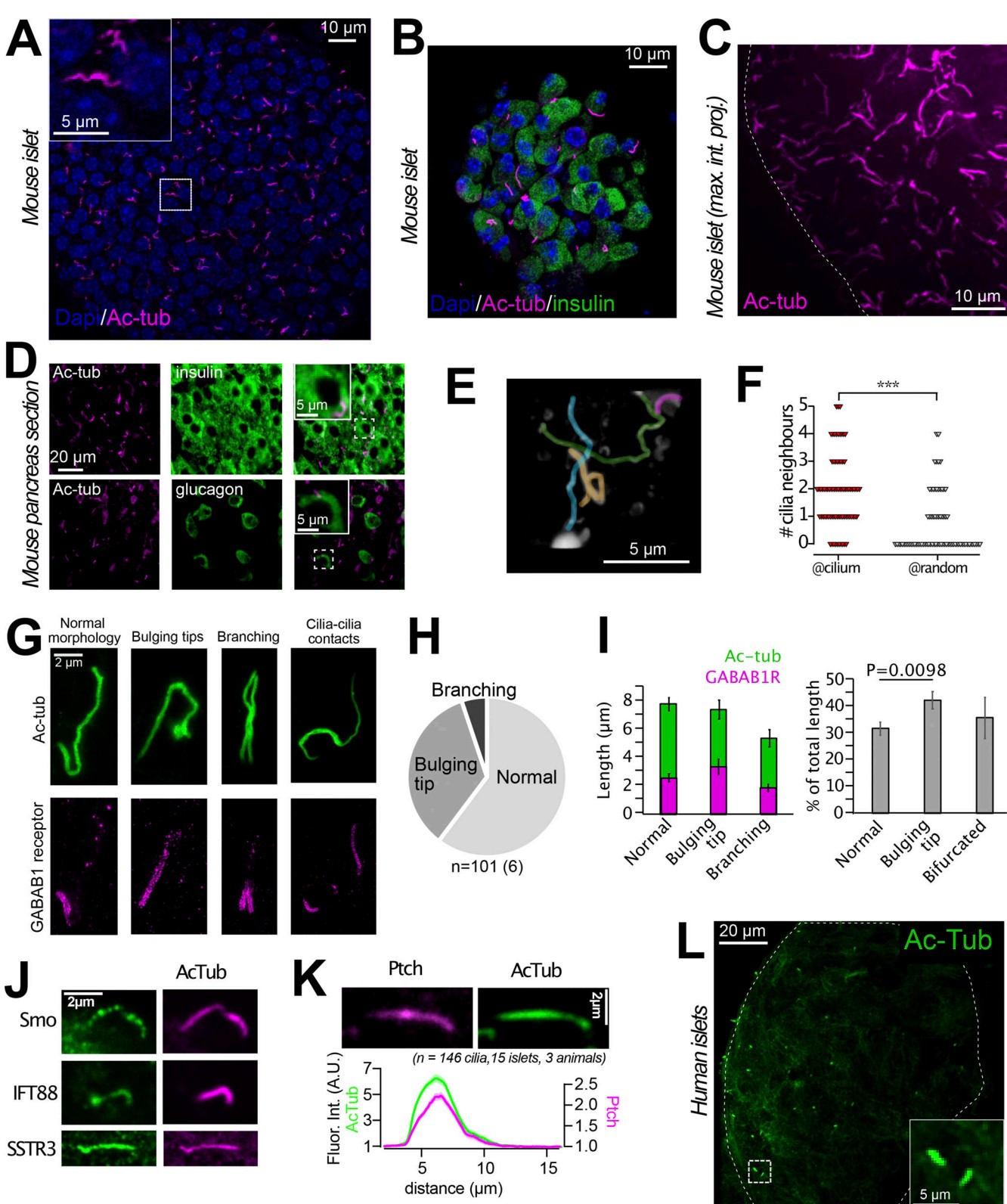

Figure 2. **The primary cilium of intact islets of Langerhans. (A)** Confocal microscopy image of a mouse islet where cilia are visualized by immunofluorescence staining against acetylated tubulin (magenta) and nuclei with Dapi staining (blue). **(B)** Confocal microscopy image of a small mouse islet showing insulin in green, acetylated tubulin in magenta, and nuclei in blue. **(C)** Maximum intensity projection of part of a mouse islet (outlined by dashed line) immunostained against acetylated tubulin. **(D)** Epifluorescence microscopy images of pancreas sections immunostained for acetylated tubulin (magenta), insulin (green, top panel), and glucagon (green, bottom panel). **(E)** 3D reconstruction of confocal micrographs of a section of a mouse islet that contains a cluster of primary cilia (pseudo-colored for clarity). **(F)** Quantification of the number of cilia found within a radius of 2.5 µm of another cilium compared to at random locations. *** P < 0.001 (two-tailed Student's unpaired *t* test). **(G)** Example STED images showing islet cell cilia morphologies (green) and the corresponding

distribution of GABA$_{B1}$ receptors (magenta). **(H)** Distribution of cilia morphologies in mouse islets ($n$ = 101 cilia from 6 islets). **(I)** Length of cilia and GABA$_{B1}$ receptor compartment in mouse islets separated based on cilia morphology (means ± SEM; $n$ = 101 cilia from 6 islets; two-tailed unpaired Student's $t$ test). **(J)** Immunofluorescence images showing the localization of Patched, Smoothened, IFT88, and SSTR3 (all green) to the primary cilium (acetylated tubulin; magenta) of mouse islet cells. **(K)** Immunofluorescence images (top) and quantification of fluorescence intensity (bottom) of acetylated tubulin (green) and Patched (magenta) in mouse islet cell cilia ($n$ = 146 cilia, 15 islets, 3 animals). **(L)** Confocal microscopy image of a large human islet showing acetylated tubulin in green.

also has affinity for cAMP, and to determine the crosstalk, we expressed it in MIN6 β-cells that were subsequently permeabilized on the stage of a TIRF microscope and exposed to intracellular-like buffers containing known concentrations of cAMP and cGMP. The sensor responded in a dose-dependent manner to cGMP in the 100–1,000 nM range, whereas 1,000 nM cAMP was required to elicit a response (Fig. 5, A and B). Consistently, MIN6 cells expressing PfPKG robustly responded to 500 nM atrial natriuretic peptide (ANP), which activate transmembrane guanyl cyclases in β-cells (Undank et al., 2017), while this stimulation was without effect in cells expressing the cAMP FRET sensor Epac-SH187 (Fig. 5, C and D). In mouse islets expressing the cilia-targeted Ca$^{2+}$ sensor 5HT$_6$-GEGO1, the

addition of 500 nM ANP caused an immediate increase in number and duration of cilia Ca$^{2+}$ flashes, which resembled those induced by GABA$_{B1}$ receptor activation (Fig. 5, E and F). Similar increase of cilia Ca$^{2+}$ activity was also seen when cGMP production was directly stimulated by a light-activated plasma membrane localized guanyl cyclase (RhGC; Scheib et al., 2018) but not when cAMP production was stimulated by a light-activated cytosolic adenylate cyclase (bPAC; Stierl et al., 2011; Fig. 5, G and H). Immunofluorescence staining revealed distinct localization of the cGMP-regulated cyclic nucleotide-gated Ca$^{2+}$ channel CNGA3 throughout the cilia of both MIN6 and mouse islet cells, where it also colocalized with GABA$_{B1}$ receptors (Fig. 5, I–K), and siRNA-mediated knockdown of CNGA3

Figure 3. **Measurements of cilia Ca$^{2+}$ in intact islets of Langerhans. (A)** Schematic illustration showing the principle of the ratiometric cilia-targeted Ca$^{2+}$ indicator Smo-GCaMP5G-mCh. **(B)** TIRF microscopy images of Smo-GCaMP5G-mCherry fluorescence from MIN6 cells. The cells were permeabilized with α-toxin and exposed to the indicated Ca$^{2+}$-buffer. Top row shows fluorescence change of GCaMP5G in response to the different Ca$^{2+}$ buffers, while the bottom row shows the corresponding change in mCherry fluorescence. **(C)** Dose-response curves for the GCaMP5G/mCherry fluorescence change in the cilium and cell body of permeabilized MIN6 cells exposed to the indicated Ca$^{2+}$ buffers ($n$ = 8 cells). **(D)** Principle of TIRF microscopy imaging of primary cilia in intact islets of Langerhans. An image of the footprint of a mouse islet with protruding cilia is shown to the right. **(E)** Quantifications of the resting GCaMP5G/mCherry fluorescence in the cilia and soma of mouse islet cells shows that the Ca$^{2+}$ concentration is lower in the cilium ($n$ = 39; **** $P$ < 0.0001; paired, two-tailed Student's $t$ test). **(F)** Quantifications of the resting GCaMP5G/mCherry fluorescence in the cilia and soma of MIN6 cells shows that there is no difference in Ca$^{2+}$ concentration between the two compartments ($n$ = 12; paired, two-tailed Student's $t$ test). **(G)** Quantifications of the resting GCaMP5G/mCherry fluorescence in the cilia and soma of human islet cells shows that there is no difference in Ca$^{2+}$ concentration between the two compartments ($n$ = 11; paired, two-tailed Student's $t$ test).

Figure 4. **GABA triggers Ca²⁺ entry into primary cilia via GABA$_{B1}$ receptors. (A)** TIRF microscopy recordings of GCaMP5G fluorescence in the soma and cilium of MIN6 cells exposed to 1 nM GABA. Notice that the addition of GABA induces a rise of cilia Ca²⁺ without affecting the Ca²⁺ concentration in the soma.

**(B)** TIRF microscopy recordings of GCaMP5G fluorescence in the soma and cilium of MIN6 cells exposed to 10 nM GABA. Notice that the addition of GABA induces a rise of cilia $Ca^{2+}$ without affecting the $Ca^{2+}$ concentration in the soma. **(C)** GABA (1 and 10 nM) and the $GABA_B$ receptor agonist Baclofen (100 nM) induces increases in the cilia $Ca^{2+}$ concentration in MIN6 cells ($n$ = 10 cells for 1 nM GABA; $n$ = 13 cells for 10 nM GABA; $n$ = 7 cells for Baclofen; * $P$ < 0.05; ** $P$ < 0.01; **** $P$ < 0.0001; Student's paired $t$ test). **(D)** Confocal microscopy immunofluorescence images of control (top) and $GABA_{B1}$ receptor KD (bottom) MIN6 cell pseudo-islets stained for acetylated tubulin (green) and $GABA_{B1}$ receptors (magenta). **(E)** Example TIRF microscopy image of a MIN6 pseudo-islet expressing $5HT_6$-GGECO (boxed area is magnified below). To the right are shown event counts of all cilia $Ca^{2+}$ flashes in control (black) and $GABA_{B1}$ receptor knockdown MIN6 pseudo-islets following addition of baclofen. Each row represents recordings from one control and one $GABA_{B1}$ receptor knockdown pseudo-islet. Quantifications are shown to the right ($n$ = 6 pseudo-islets for each condition). Statistical significance was assessed using Wilcoxon signed rank test. A TIRF microscopy image of a $5HT_6$-GGECO1-expressing MIN6 pseudoislet is shown to the left. **(F)** TIRF microscopy recordings of cilia $Ca^{2+}$ concentration changes in intact mouse islets expressing Smo-GCaMP5G-mCh. Picture on top shows the distribution of cilia within the islet, with cilia exhibiting at least one spontaneous $Ca^{2+}$ flash during the recording time highlighted in color. In the middle are shown $Ca^{2+}$ recordings from the individual cilia within the islet during addition of the $GABA_B$ receptor agonist Baclofen. The diagram at the bottom shows the overall cilia $Ca^{2+}$ activity of the whole islet. **(G)** Event count of all cilia $Ca^{2+}$ flashes following the addition of baclofen ($n$ = 7 islets). **(H)** Event count of all cilia $Ca^{2+}$ flashes following the addition of vigabatrin ($n$ = 13 islets). **(I)** Quantifications of the effect of baclofen and vigabatrin on cilia $Ca^{2+}$ flashes. Statistical significance was assessed using the Friedman test followed by multiple comparisons.

suppressed ANP-induced $Ca^{2+}$ increase in the cilium of MIN6 pseudoislets (Fig. 5 L and Fig. S4). CNGA3 typically form heteromers with CNGB1, however immunostainings of mouse islets did not show any enrichment of CNGB1 in cilia (Fig. S4). To test whether cyclic nucleotide gated (CNG) channel activation could be responsible for the GABA-induced $Ca^{2+}$ influx in cilia, we stimulated mouse islets with 100 nM of the $GABA_{B1}$ receptor agonist baclofen in the absence or presence of 100 µM of the CNG channel blocker L-cis-diltiazem. The addition of baclofen, as expected, caused an increase in cilia $Ca^{2+}$ activity that was unaffected by CNG channel inhibition (Fig. 5 M). $GABA_B$ receptors have been shown to interact with, and regulate, voltage-dependent $Ca^{2+}$ channels (VDCCs; Shen and Slaughter, 1999). β-cells express several functional VDCCs, including L-type channels, which are responsible for the $Ca^{2+}$-triggered release of insulin. Consistent with this, three structurally unrelated voltage-dependent $Ca^{2+}$ channel inhibitors (diltiazem, nifedipine, and verapamil) strongly inhibited depolarization-induced $Ca^{2+}$ influx in mouse islet β-cells (Fig. S4). Both verapamil (25 and 100 µM) and diltiazem (100 µM) also blocked the baclofen-induced $Ca^{2+}$ influx into cilia, while nifedipine (10 µM) was without effect (Fig. 5 M). Immunostaining of mouse islets revealed the presence, but not enrichment, of CaV1.2 L-type VDCC in primary cilia of some, but not all, islet cells, while CaV1.3 was absent from cilia (Fig. S4). In pertussis toxin-treated islets, where Gi-mediated signaling is inhibited (Fig. S4), baclofen retained the ability to induce $Ca^{2+}$ signaling in the primary cilium (Fig. 5 N). Together, these results show that cGMP elevations can trigger $Ca^{2+}$ influx in primary cilia of islet cells, but this mechanism is not involved in GABA action, which instead depends on local $Ca^{2+}$ influx through voltage-dependent channels through a mechanism that is independent of classical Gi-coupling.

### Spontaneous cilia $Ca^{2+}$ flashes in isolated pancreatic islets
The observation of GABA-induced $Ca^{2+}$signaling that was restricted to the primary cilium prompted us to ask whether the cilia of the islet might function as sensory antenna for locally released molecules. When performing prolonged recordings of cilia $Ca^{2+}$ concentrations in mouse islets kept in a low (3 mM) glucose-containing buffer, we observed the occurrence of spontaneous ciliary $Ca^{2+}$ flashes (Fig. 6 A). Similar flashes were also observed in human islet cells and in clonal MIN6 β-cells

(Fig. 6, B and C, and Fig. S4). Flashes were characterized by fast onset and offset transitions, and it was possible to identify the site of origin and to follow $Ca^{2+}$ spreading along the cilium (Fig. 6, D–I). Occasionally, it was even possible to detect a small rise of cytosolic $Ca^{2+}$ as the cilia $Ca^{2+}$ wave reached the base (see Fig. 6 A). The propagation of $Ca^{2+}$ in the cilium had a similar diffusion rate as that previously reported (Delling et al., 2013), and the average duration of these events was close to 1 min (Fig. 6, E–G). The $Ca^{2+}$ flashes were not caused by the overexpression of Smo-GCaMP5G-mCh, since similar responses were seen in islet cell cilia expressing a different $Ca^{2+}$ indicator ($5HT_6$-GGECO1; Fig. S5). Overexpression of cilia-localized proteins is known to affect cilia composition and function (May et al., 2021). Importantly, $5HT_6$-GGECO1 overexpression in either mouse islets or MIN6 pseudoislets did not influence Hedgehog signaling, determined as the exit of Patched from the cilium upon treatment with the Smoothened agonist SAG, nor prevent the SAG-induced lengthening of cilia (Fig. S5). Notably, most of the flashes started in the more distal regions of the cilium, including the tip. Observations of cilia $Ca^{2+}$ flashes throughout the cilia of an islet revealed a degree of temporal coordination, but not perfect synchronization, between the events across the islet, which may indicate that these fluctuations are caused by the action of a soluble molecule (Fig. 5 J). Flashes disappeared when extracellular $Ca^{2+}$ was removed and were suppressed when the glucose concentration was elevated or when islet cells were depolarized with KCl (Fig. 6, K–N and Fig. S5). GABA is locally produced and released from islet β-cells in a glucose-independent manner. To test whether the spontaneous cilia $Ca^{2+}$ flashes were caused by local GABA release and detection, we investigated the effect of GABA receptor antagonists. Application of a combination of $GABA_A$ (picorotoxin) and $GABA_B$ (CGP35348) receptor antagonists were without effect on the spontaneous cilia $Ca^{2+}$ signaling, but washout of the antagonists resulted in a pronounced increase in $Ca^{2+}$ activity (Fig. 6, O and Q). Moreover, long-term (5 h) inhibition of GABA breakdown with vigabatrin led to a reduction in spontaneous cilia $Ca^{2+}$ activity that did not reach statistical significance (Fig. 5, P and R). Together, these results show that primary cilia function as antennas for sensing the islet environment and that $Ca^{2+}$ is used as a messenger to propagate this response.

Figure 5. **cGMP, but not cAMP, triggers Ca²⁺ influx in islet cell cilia. (A)** TIRF microscopy recording of cytosolic PfPKG (cGMP sensor) FRET ratio in permeabilized MIN6 cells exposed to the indicated cAMP and cGMP concentrations. **(B)** Quantifications of the PfPKG FRET ratio change in response to the indicated cyclic nucleotide concentrations (*n* = 14 cells). **(C)** Normalized PfPKG FRET ratio in response to 500 nM ANP and 100 µM IBMX in combination with 500 nM ANP (means ± SEM; *n* = 15 MIN6 cells). Statistical significance was assessed with paired two-tailed Student's *t* test. **(D)** Normalized Epac-S^H187 FRET ratio in response to 500 nM ANP and 100 µM 3-isobutyl-1-methylxanthine (IBMX) in combination with 500 nM ANP (means ± SEM; *n* = 39 MIN6 cells).

Statistical significance was assessed with paired two-tailed Student's $t$ test. **(E)** Representative recording of cilium (black) and cytosolic (red) $Ca^{2+}$ concentration changes in a mouse islet cell expressing 5HT$_6$-G-GECO1 and exposed to 500 nM ANP. **(F)** Event count of all cilia $Ca^{2+}$ flashes following the addition of 500 nM ANP or solvent (water; $n$ = 5 islets for control; 8 islets for ANP). **(G)** Event count of all cilia $Ca^{2+}$ flashes following yellow light illumination of islets expressing 5HT$_6$-G-GECO1 and RhGC (gray) or empty vector (black; $n$ = 3 islets for both conditions). Red lines have been added to simplify comparison between the two test groups. **(H)** Event count of all cilia $Ca^{2+}$ flashes following blue light illumination of islets expressing 5HT$_6$-G-GECO1 and bPAC (gray) or empty vector (black; $n$ = 7 islets for both conditions). Red lines have been added to simplify comparison between the two test groups. **(I)** Immunofluorescence images of a mouse islet stained against acetylated tubulin (cilia; magenta) and cyclic nucleotide-gated $Ca^{2+}$ channel A3 (CNGA3; green). **(J)** STED images of a MIN6 cell cilium immunostained against acetylated tubulin (magenta) and CNGA3 (green). **(K)** Immunofluorescence images of a mouse islet cell cilia stained against GABA$_{B1}$ receptors (GABAB1R; magenta) and CNGA3 (green). **(L)** Quantifications of cilia $Ca^{2+}$ responses (5HT$_6$-GGECO1) from control (black) or CNGA3 KD (gray) MIN6 pseudoislets under basal conditions or following stimulation with 500 nM ANP (means ± SEM; $n$ = 5 [control] and 4 [CNGA3 KD] pseudoislets; * P < 0.05; Mann-Whitney $U$ test). **(M)** Quantifications of cilia $Ca^{2+}$ responses from mouse islet cells expressing 5HT$_6$-GGECO1 and imaged with TIRF microscopy. Individual cilia $Ca^{2+}$ events were counted under basal conditions (gray) and following addition of 100 nM baclofen alone (red) or in combination with 100 μM L-cis-diltiazem (dark blue), 100 μM diltiazem (light blue), 25 μM, verapamil (light purple), 100 μM verapamil (dark purple), or 10 μM nifedipine (green). Each data point represents the number of ciliary $Ca^{2+}$ events per islet for 9 islets (baclofen), 6 islets (L-cis-diltiazem), 10 islets (diltiazem), 6 islets (25 μM verapamil) 4 islets (100 μM verapamil), and 6 islets (nifedipine). Bars show means ± SEM and statistical significance was assessed using Wilcoxon signed-rank test. * P < 0.05. **(N)** Single cell recordings of 5HT6-GGECO1 fluorescence in cilia of mouse islet cells cultured under control conditions (black) or in the presence of pertussis toxin (PT; blue) and stimulated with 100 nM baclofen. Quantification of the response to baclofen in pertussis toxin-treated islets is shown to the right ($n$ = 45 cilia from 7 islets; * P < 0.05; statistical significance assessed with Wilcoxon signed rank test).

## The primary cilium is isolated against cytosolic $Ca^{2+}$ concentration changes

If the primary cilia of the islet cells utilize $Ca^{2+}$ as a signaling molecule, there has to be mechanisms in place that enable the cilium to distinguish $Ca^{2+}$ of ciliary origin from that diffusing in from the cytosol. This becomes particularly important in the case of β-cells, where the cytosolic $Ca^{2+}$ concentration undergoes regular changes that trigger insulin granule exocytosis and release to the circulation. We therefore decided to determine how changes in the cytosolic $Ca^{2+}$ concentration affect the $Ca^{2+}$ concentration in the primary cilium of cells within intact mouse islets. To this end, mouse islets expressing Smo-GCaMP5G-mCh were exposed to three stimuli that increase the cytosolic $Ca^{2+}$ concentration through distinct mechanisms; depolarization-induced entry from the extracellular space, release from intracellular stores, and photorelease of caged $Ca^{2+}$. The depolarization-induced opening of voltage-gated $Ca^{2+}$ channels in islet β-cells caused a robust increase in the cytosolic $Ca^{2+}$ concentration. We segmented the cilium into sub-compartments and observed a small rise of $Ca^{2+}$ at the base of the cilium which mirrored the change in cytosolic $Ca^{2+}$, while more distal compartments of the cilium were unaffected by the depolarization (Fig. 7, A–C). This indicates the existence of a $Ca^{2+}$ diffusion barrier and mechanisms operating at the base of cilium that prevent the penetration of $Ca^{2+}$ from the cytosol into the cilium. An alternative source of $Ca^{2+}$ is represented by intracellular stores. Release of $Ca^{2+}$ from the ER was triggered by the muscarinic receptor agonist carbachol while islets were kept in 20 mM glucose to keep the ER stores filled and diazoxide to prevent glucose-induced $Ca^{2+}$ influx. As expected, activation of muscarinic receptors produced a fast transient increase in cytosolic $Ca^{2+}$, which was accompanied by an even more pronounced ciliary $Ca^{2+}$ increase that, however, was restricted to a small compartment of around 3 μm in length at the base of the cilium (Fig. 7, D–G). The size of this compartment showed little variability between cilia and did not correlate strongly with the length of the cilium (Fig. 7, H and I). The rate of $Ca^{2+}$ extrusion was more rapid at the cilia base than in the cytosol of the same cell, and this difference was eliminated when extracellular $Na^+$ was removed, indicating

different modes of $Ca^{2+}$ regulation in the two compartments (Fig. S6). Similar results were also obtained using an alternative cilia-localized $Ca^{2+}$ sensor (Fig. S6). Next, we triggered acute elevations of cytosolic $Ca^{2+}$ by photorelease of caged $Ca^{2+}$ in discrete cytosolic regions at distance from the cilium. The photorelease generated a $Ca^{2+}$ wave that propagated into the cilium (Fig. 7, J and K). Interestingly, the spreading was strongly dampened within the cilium and was further reduced under conditions when the islets were depolarized by 30 mM KCl (Fig. 7, L and M). This effect of depolarization was abolished when 200 μM amiloride was included in the solution or reduced when extracellular $Na^+$ was removed (Fig. 7, L and M). The effects were most pronounced in the distal parts of the cilium (tip) and only small effects of the modulations were seen in the cytosol (Fig. 7, L and M). To determine whether this mechanism also restricts $Ca^{2+}$ diffusion when the elevation occurred in the cilium, we measured the rate of $Ca^{2+}$ extrusion along the length of islet cell cilia pixel-by-pixel following spontaneous $Ca^{2+}$ activity. We found that the rate of extrusion was highest at the cilia base and then gradually declined in more distal parts of the cilium (Fig. 7, O–R). The extrusion rate in the tip was more variable, but typically more similar to the base than the center of the cilium. Similar results were obtained using islets expressing both 5HT$_6$-GGECO1 and Smo-GCaMP5G-mCh, where the latter allowed easier determination of cilia orientation due to tip dilation (Fig. S6). Together, these results indicate that primary cilia of islet cells have enhanced $Ca^{2+}$ extrusion compared to the cell bodies and that this arrangement restricts $Ca^{2+}$ diffusion between the two compartments. In addition to extrusion, it is also possible that the cilium has a different $Ca^{2+}$ buffering capacity. To test whether enhanced $Ca^{2+}$ buffering can reduce $Ca^{2+}$ diffusion in the cilium, we targeted the $Ca^{2+}$-binding protein S100G to the cilium through fusion with Smo-mCherry. Photorelease of $Ca^{2+}$ in MIN6 cells expressing Smo-S100G-mCh resulted in reduced $Ca^{2+}$ spreading in the cilium compared to cells expressing Smo-mCherry (control), while the responses in the cell bodies were similar, showing that $Ca^{2+}$ buffering within the cilium may help to shape the $Ca^{2+}$ signals. Taken together, these results show that the cilium is isolated from cytosolic

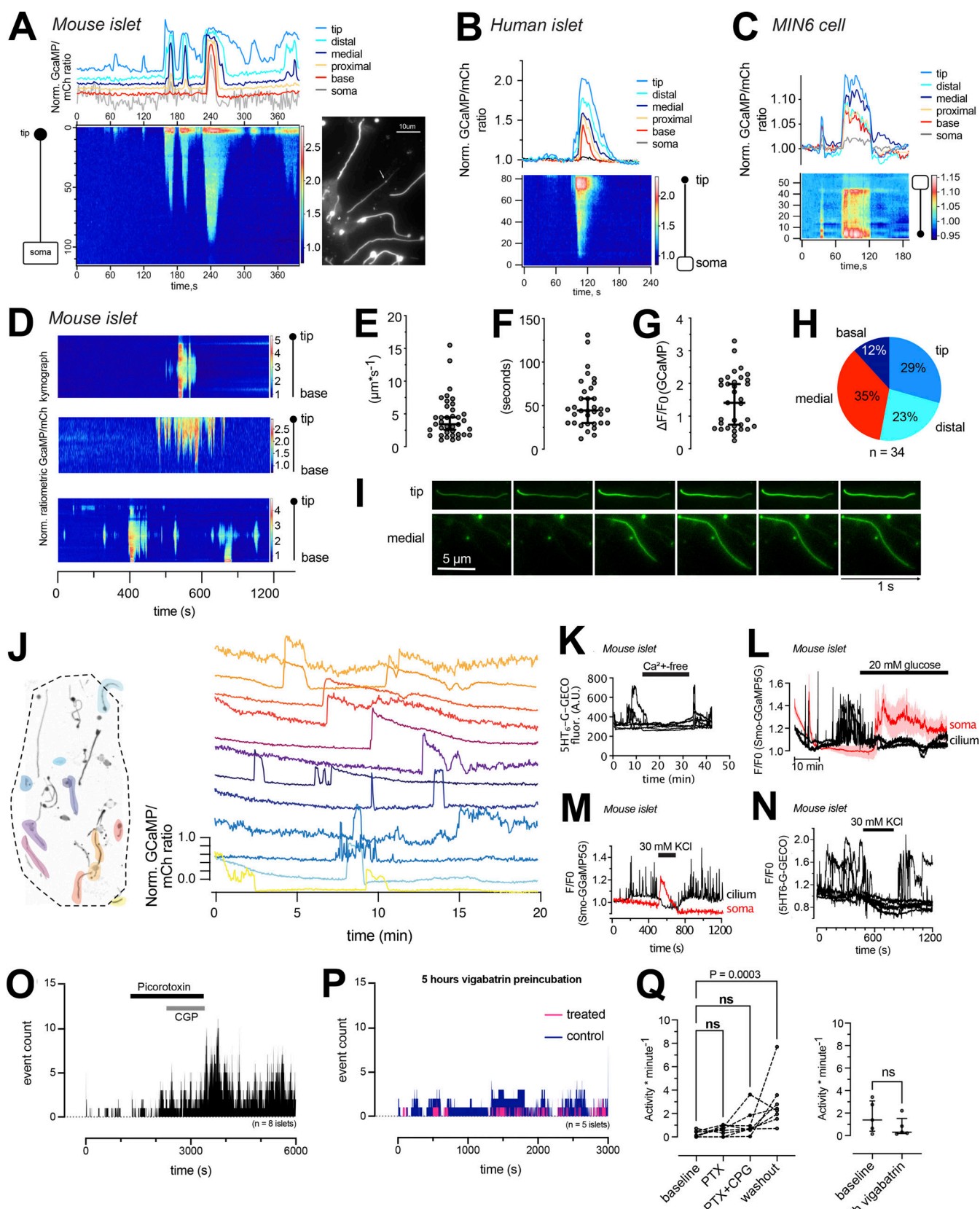

**Figure 6. Isolated cilia Ca²⁺ activity within intact islets of Langerhans. (A)** TIRF microscopy recording of CaMP5G fluorescence from the soma and cilium of a mouse islet cell during a spontaneous Ca²⁺ "flash" (white arrow points to the cilium). Traces show the Ca²⁺ concentration change at different segments of the cilium and in the soma (gray) and the kymograph shows the GCaMP5G fluorescence change along a line going from the soma to the tip of the cilium. **(B)** A spontaneous cilia Ca²⁺ flash recorded from a human islet β-cell expressing Smo-GCaMP5G-mCh. **(C)** A spontaneous cilia Ca²⁺ flash recorded from a MIN6 β-cell

expressing Smo-GCaMP5G-mCh. **(D)** Kymographs showing examples of spontaneous Ca²⁺ activity in three cilia-soma pairs within intact mouse islets. **(E)** Ca²⁺ diffusion rates in the primary cilium of mouse islet cells during spontaneous Ca²⁺ flashes ($n$ = 35 cilia from 30 islets). **(F)** Duration of spontaneous Ca²⁺ flashes in mouse islet primary cilia ($n$ = 35 cilia from 30 islets). **(G)** GCaMP5G fluorescence change during spontaneous Ca²⁺ flashes in mouse islet cells ($n$ = 35 cilia from 30 islets). **(H)** Site of Ca²⁺ flash in the primary cilium ($n$ = 34 cilia). **(I)** Examples showing the initiation and propagation of Ca²⁺ along the cilium of islet cells. **(J)** GCaMP5G fluorescence change in the primary cilia of a mouse islet kept in a basal buffer containing 3 mM glucose. Picture to the left shows the distribution of cilia within the islet (color represents individual cilia. Unmarked cilia did not exhibit spontaneous activity during the recording time). Traces to the right show the fluorescence change in individual cilia over time. Notice that there is certain coordination in responses between cilia of the islet. **(K)** TIRF microscopy recording of G-GECO1 fluorescence in the cilium of cells ($n$ = 7) in one islet cells during brief removal of extracellular Ca²⁺. **(L)** TIRF microscopy recording of GCaMP5G fluorescence in the cilia and cell bodies of cells in one islet exposed to 20 mM glucose. Notice that the rise of cytosolic Ca²⁺ is accompanied by the suppression of cilia Ca²⁺ flashes (means ± SEM for 10 cell bodies; three individual cilia within the same islet). **(M)** TIRF microscopy recording of GCaMP5G fluorescence in the cilium and cell body of an islet cell exposed to brief depolarization by 30 mM KCl. Notice that the rise of cytosolic Ca²⁺ is accompanied by the suppression of cilia Ca²⁺ flashes. **(N)** TIRF microscopy recording of GCaMP5G fluorescence in the cilia ($n$ = 7) of one islet exposed to brief depolarization by 30 mM KCl. **(O)** Event count of all cilia Ca²⁺ flashes from mouse islets following the addition and washout of picrotoxin and CGP-35348 ($n$ = 8 islets). **(P)** Event count of all cilia Ca²⁺ flashes following preincubation with DMSO (control) and vigabatrin for 5 h ($n$ = 5 islets for both conditions). **(Q)** Quantifications of the effect of picrotoxin + CGP-35348 and long-term vigabatrin treatment on cilia Ca²⁺ flashes. Statistical significance was assessed using the Friedman test followed by multiple comparisons.

Ca²⁺, and that this at least in part is due to an enhanced Ca²⁺ extrusion, which is Na⁺ dependent and sensitive to amiloride.

### Cilia Ca²⁺ dynamics in glucose-stimulated islet cells

To determine the cilia Ca²⁺ activity in a more physiological context, we exposed Smo-GCaMP5G-mCh–expressing islets to a buffer containing 11 mM glucose, a concentration at which insulin secretion is strongly stimulated. This resulted in an immediate and very pronounced increase in GCaMP5G fluorescence in the islet cell cilia (Fig. 8 A and Fig. S7). Less pronounced increases were also seen in the cell bodies. This rise occurred before depolarization-induced Ca²⁺ influx and was not seen with soluble GCaMP (Chen et al., 2016; Fig. S7), making us question whether it reflects a genuine Ca²⁺ concentration change. One possibility is that ATP generated by glucose metabolism affects the fluorescent properties of GCaMP5G (Willemse et al., 2007). Experiments performed in permeabilized MIN6 cells expressing Smo-GCaMP5G-mCh or a plasma membrane–anchored red Ca²⁺ indicator (Lyn₁₁-R-GECO) did however not reveal any direct impact of ATP on either GCaMP5G or R-GECO fluorescence (Fig. S7). Measurements of pH using 5HT₆-venus-CFP (Su et al., 2013) showed that glucose stimulation alkalized both cytosol and cilia. Since GCaMP5G fluorescence is strongly influenced by pH (Cho et al., 2017), it is reasonable to assume that the initial glucose-induced increase in GCaMP5G fluorescence is caused by a rise in pH (Fig. S7). This glucose-induced rise of GCaMP5G fluorescence was followed by the appearance of cytosolic Ca²⁺ oscillations with a period of 2–5 min (Fig. 8 A). We observed similar periodic fluctuations in the cilia Ca²⁺ concentration, and when we analyzed the low-frequency component of the oscillations in pairs of connected cilia and cell bodies, we found a stable phase shift of $-\pi/2$ between the two signals (Fig. 8, A and B). The anti-phase locking of the cilia and cytosolic Ca²⁺ oscillations may reflect the enhancement of Ca²⁺ extrusion due to depolarization taking place at supra-threshold membrane potentials that marks the peak of cytoplasmic [Ca²⁺] and the nadir of the ciliary [Ca²⁺]. Inspection of the change in cilia Ca²⁺ concentration along the length of the cilium revealed a pattern reminiscent of that observed following depolarization, where the cytosol-proximal region exhibited

changes similar to the cytosol (i.e., oscillations in-phase), whereas more distal regions were phase-shifted (Fig. 8 C). There was a positive correlation between the duration of the cytosolic [Ca²⁺] increase and the decrease in cilia [Ca²⁺] (Fig. 8 D), and pulsatile application of depolarizing KCl concentrations mimicked the effect of glucose (Fig. 8 E). Addition of the K$_{ATP}$-channel opener diazoxide or the L-type voltage-dependent Ca²⁺ channel inhibitor verapamil inhibited the glucose-induced Ca²⁺ oscillations in both compartments (Fig. 8, F and G), while the addition of the cAMP-elevating agent forskolin transformed the slow cytosolic Ca²⁺ oscillations to rapid spike-like fluctuations and was accompanied by the disappearance of cilia Ca²⁺ oscillations (Fig. 8 H). These results indicate that the glucose-induced Ca²⁺ concentration changes in the cilium and soma are driven by a common underlying mechanism. The cilium is devoid of mitochondria but has been shown to have intrinsic glycolysis and could therefore potentially generate local ATP from glucose that might drive Ca²⁺ influx or extrusion (Villar et al., 2020). To test whether a cilia-derived metabolite might be responsible for the observed anti-synchronicity, we exposed islets to the mitochondrial substrate α-ketoisocaproic acid (5 mM). This resulted in similar slow Ca²⁺ oscillations in the β-cell cilia that were out-of-phase with those of the cell bodies (Fig. 5 I). Taken together, these results confirm our previous observations that the islet cell cilia are effectively isolated against changes in the cytosolic Ca²⁺ concentration and demonstrate that this mechanism operates in a physiological context.

### Mechanical stimulation does not trigger Ca²⁺ signaling in islet cell primary cilia

The primary cilium is a mechanosensory organelle in many cell types, although to what extent Ca²⁺ is a relevant signaling molecule downstream of mechanical stimulation in cilia is controversial (Delling et al., 2016; Jin et al., 2014). We therefore exposed islet cells to two forms of mechanical stimulation, mild compression and fluid flow, while recording Ca²⁺ concentration changes in the primary cilia by confocal microscopy. Compression was accomplished using the cell press (O'Callaghan et al., 2022), a piezomotor-controlled manipulator that linearly translates a flexible polydimethylsiloxane (PDMS) compression pillar, which was fitted to the stage of the microscope. Using this

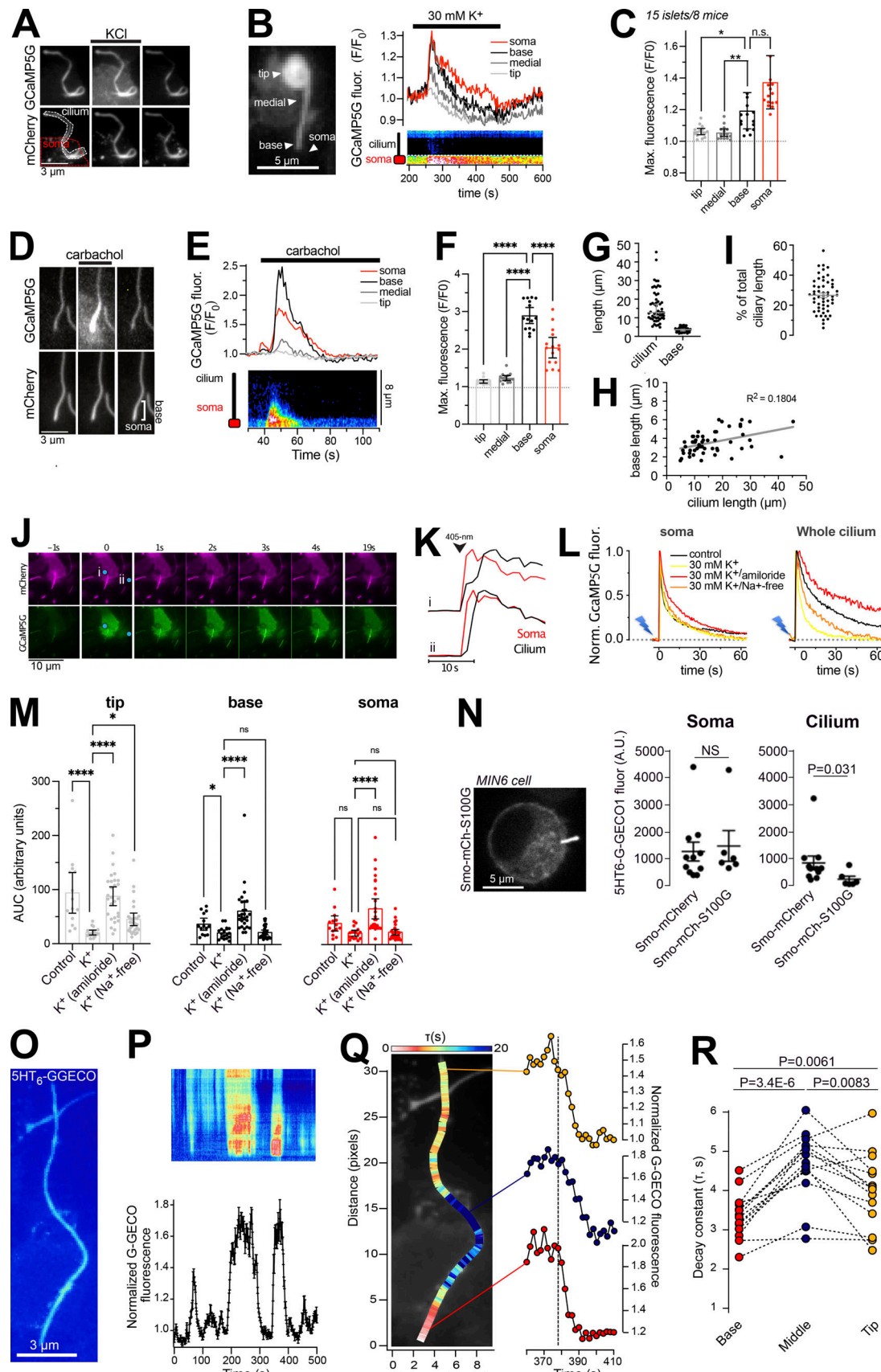

Figure 7. **Restricted entry of cytosolic Ca²⁺ into the primary cilium of islet cells. (A)** TIRF microscopy images of Smo-GCaMP5G-mCh fluorescence change during KCl-depolarization. Notice the lack of Ca²⁺ increase in the more distal parts of the cilium. **(B)** TIRF microscopy recordings of GCaMP5G fluorescence from

mouse islet cells transduced with an adenovirus encoding Smo-GCaMP5G-mCh. The islet was exposed to a brief depolarization (30 mM KCl) while the $Ca^{2+}$ concentration was recorded in the soma (red) and cilium (grayscale). **(C)** Quantifications of the GCaMP5G fluorescence increase in response to depolarization in the cell body (red) and in three cilia segments (grayscale; means ± SEM; $n$ = 14 cells; Friedman test followed by Dunn's Multiple Comparisons). **(D)** TIRF microscopy images of Smo-GCaMP5G-mCh fluorescence change during stimulation with 10 µM carbachol. Notice the strong response at the cilia base and the lack of $Ca^{2+}$ increase in the more distal parts of the cilium. **(E)** TIRF microscopy recordings of GCaMP5G fluorescence from mouse islet cells transduced with an adenovirus encoding Smo-GCaMP5G-mCh. The islet was exposed to 100 µM carbachol while the $Ca^{2+}$ concentration was recorded in the soma (red) and cilium (grayscale). **(F)** Quantifications of the GCaMP5G fluorescence increase in response to carbachol in the cell body (red) and in three cilia segments (grayscale; means ± SEM; $n$ = 14 cells; repeated measures one-way ANOVA followed by multiple comparisons). **** $P < 0.0001$. **(G)** Quantifications of the cilia length and the length of the compartment at the cilia base to which the carbachol response was restricted in mouse islet cells expressing Smo-GCaMP5G-mCh ($n$ = 54 cilia, 12 islets, 3 animals). **(H)** Correlation between cilia length and length of the basal compartment to which the carbachol response was restricted. **(I)** Fraction of total cilia length affected by the carbachol stimulation. **(J)** TIRF microscopy images of Smo-GCaMP5G-mCh fluorescence change following $Ca^{2+}$-uncaging in two islet cells (blue dot indicate site of photolysis). **(K)** $Ca^{2+}$ concentration change in soma (red) and cilium (black) following the uncaging in J. **(L)** Recordings of GCaMP5G fluorescence change in the soma and cilia of mouse islet cells following $Ca^{2+}$ uncaging under control conditions (black), in the presence of 30 mM KCl (yellow), in the presence of 30 mM KCl in combination with Amiloride (200 µM, red), or in the presence of 30 mM KCl and no extracellular $Na^+$ (replaced with $Li^+$). Data presented as means for 20–40 cells. **(M)** Quantification of the $Ca^{2+}$ concentration change in the cell body (red), cilium base (black), and cilium tip (gray) following $Ca^{2+}$ uncaging under the indicated conditions. (means ± SD; $n$ = 14, 17, 28, 24, 29, respectively, 4–6 islets, 1 animal each, Kruskal–Wallis test followed by Dunn's Multiple Comparisons). * $P < 0.05$; **** $P < 0.0001$. **(N)** Picture to the left shows a confocal microscopy image of a MIN6 cell expressing a cilia-targeted $Ca^{2+}$ chelator (Smo-S100G-mCh). Scatterplot to the right shows the change in 5HT6-GGECO1 fluorescence in the soma and cilia of MIN6 cells expressing Smo-mCherry ($n$ = 11) or Smo-S100G-mCherry ($n$ = 6) following photo-release of $Ca^{2+}$. Bars show means ± SEM and statistical significance was assessed with the two-tailed unpaired Student's $t$ test. **(O)** TIRF micrograph of an islet cell cilium expressing 5HT6-GGECO1. **(P)** Normalized kymograph from (top), and average 5HT6-GGECO1 fluorescence change in (bottom), the cilium in O during a spontaneous $Ca^{2+}$ flash. **(Q)** The same cilium as in O that has been pseudo-colored to illustrate the rate of $Ca^{2+}$ extrusion during the declining phase of the spontaneous $Ca^{2+}$ flash. The colors show the tau values (rates) for each pixel-wide segment of the entire cilia length, where white/red represents the fastest rate and blue represents the slowest rate. Individual intensiometric traces of 5HT6-GGECO1 fluorescence change from three distinct segments of the cilium (base, middle, and tip) are shown to the right. Notice that the rate of extrusion is fastest at the cilia base. **(R)** Rate of $Ca^{2+}$ decay in three cilia segments (base, middle [40 pixels from base], and tip). Each point corresponds to one cilium and lines indicate individual cilium ($n$ = 14 cilia, two-tailed paired Student's $t$ test).

device, we compressed islets expressing the cilia-targeted $Ca^{2+}$ indicator. Mild compression, which did not cause noticeable change in islet morphology, was typically without effect on cilia $Ca^{2+}$, whereas strong compression, which caused visible islet deformation, triggered a small $Ca^{2+}$ response in 10% of the cilia (Fig. 9, B and C). Since compression affects both cilia and soma of the islet cells, it is possible that the observed effect is secondary to the compression of the soma. To directly stimulate cilia without affecting the cell bodies, we exposed MIN6 β-cell cilia to fluid flow delivered through a pulled glass pipette placed next to a cilium using a micromanipulator (Fig. 9 E). Local "puffing" of the bath solution caused noticeable movement of the cilium but did not result in any change in cilia $Ca^{2+}$ concentration (Fig. 9, F and G). Together, these results suggest that β-cell cilia do not respond directly to mechanical stimulations with increased $Ca^{2+}$ concentrations.

## Discussion

Studies of mammalian primary cilia $Ca^{2+}$ signaling have largely been carried out in cell lines, with important exception being groundbreaking work on isolated primary olfactory neurons (Leinders-Zufall et al., 1997; Leinders-Zufall et al., 1998), and more recently also on embryonic tissues (Mizuno et al., 2020). Here we employed the islet of Langerhans as a model system that preserves organ like features and allows exploration of ciliary signaling by imaging approaches with high spatiotemporal resolution. Two features of this preparation are of crucial relevance in this regard; the cells have an extremely low rate of proliferation (which means that their cilia develop a specialized phenotype) and a variety of endogenously produced signaling molecules are released from, and act locally in, the isolated islet. This is exemplified by the fact that both endogenous and

overexpressed Smoothened shows a constitutive ciliary localization in islet cells, likely due to the endogenous production and release of Hedgehog (Cigliola et al., 2018; Landsman et al., 2011). Using this model, we find that primary cilia of islet cells are efficiently isolated against changes in the cytosolic $Ca^{2+}$ concentration, and that this isolation enables the cilia to function as antennas that detect GABA through cilia-localized $GABA_{B1}$ receptors that are coupled to local $Ca^{2+}$ influx.

A main finding in this study is that the primary cilium of islet cells is isolated against changes in cytosolic $Ca^{2+}$. This is different from previous observations in epithelial cells, where $Ca^{2+}$ was shown to freely diffuse between the cytosol and cilium (Delling et al., 2013; Su et al., 2013). The reason for this discrepancy is not clear, but one possibility is that efficient extrusion is more easily accomplished in the very long cilia of islet cells. It is also possible that this mechanism is more important in excitable cells, where regular changes in the cytosolic $Ca^{2+}$ concentration are an essential part of normal cell function. Such an isolation is therefore a fundamental requirement if $Ca^{2+}$ is to function as a cilia-intrinsic second messenger generated downstream of cilia-localized receptors and ion channels. Seclusion of the cilium is achieved by potent $Ca^{2+}$ extrusion concentrated at the base of the cilium. The finding that the addition of amiloride or removal of $Na^+$ slows down clearance kinetics of both physiologically generated $Ca^{2+}$ signals and elevations induced by photorelease of caged $Ca^{2+}$ points to the involvement of a $Na^+$-dependent transport process. In addition to extrusion, local $Ca^{2+}$ buffering likely contributes to shaping cilia $Ca^{2+}$ signals. The ciliary proteome contains many $Ca^{2+}$-binding proteins that may serve this function (Mick et al., 2015), and increasing the $Ca^{2+}$ buffering capacity by overexpression of $Ca^{2+}$-binding proteins reduced ciliary $Ca^{2+}$ diffusion (Yuan et al., 2015; see Fig. 7 N).

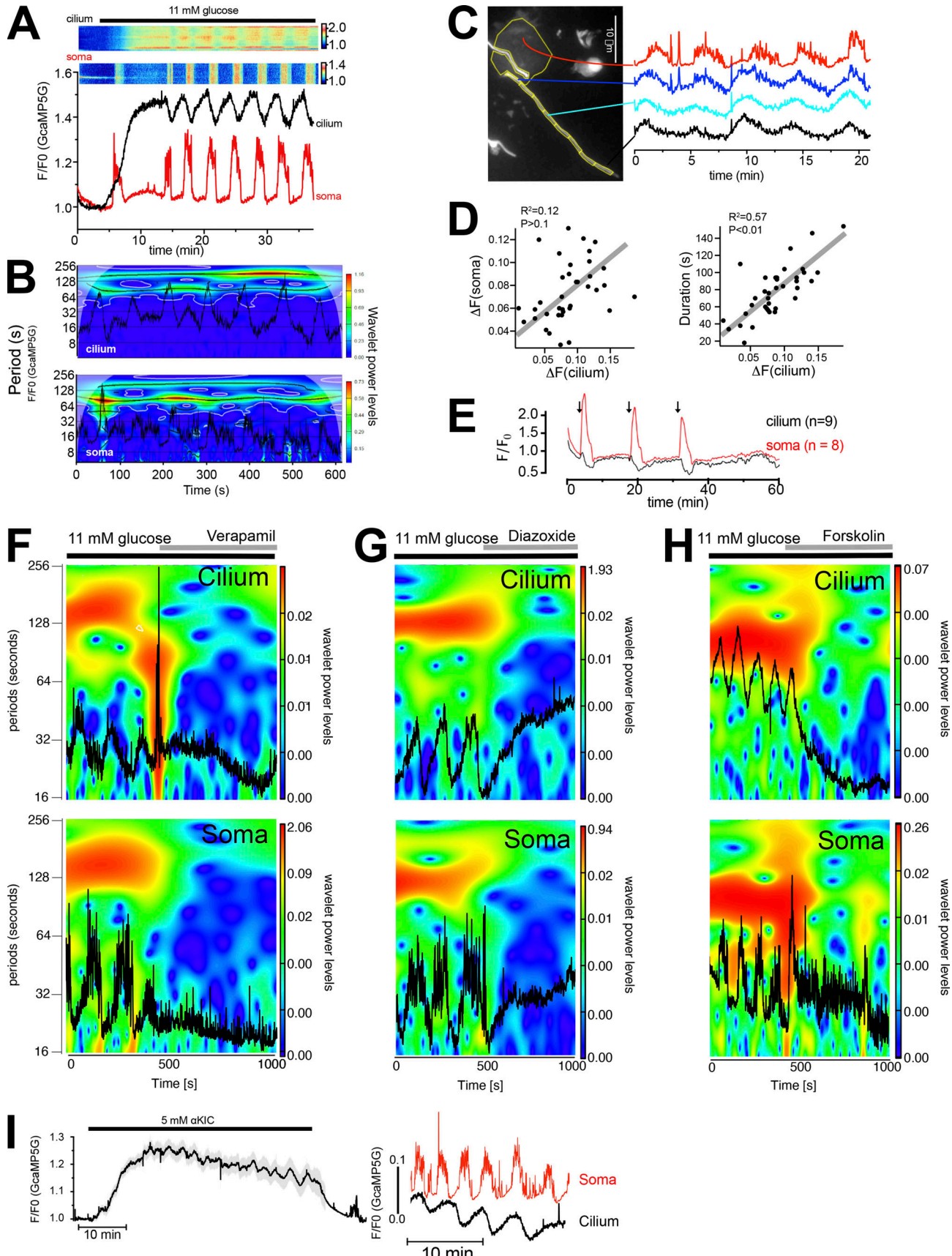

Figure 8. **Cilia and cytosolic Ca²⁺ are regulated by a common mechanism. (A)** TIRF microscopy recording of Smo-GCaMP5G-mCh fluorescence from a β-cell within an intact mouse islet during an increase in the glucose concentration from 3 to 11 mM. Notice the appearance of regular oscillations in both cilium

and soma. Kymograph above shows GCaMP5G/mCh ratio changes in a small part of the soma and along the cilium. **(B)** Color coded spectrogram of ciliary (top) and somatic (bottom) frequency components with superimposed traces (black) of the whole compartments $Ca^{2+}$ activity recorded from one cell during 11 mM glucose exposure. Note the prevalence of a slow component with a period of about 100 s in both cilium and soma. **(C)** GCaMP5G fluorescence change from a mouse islet cell stimulated with 11 mM glucose. The cilium has been segmented and the fluorescence change within the segments is shown to the right. Notice that there is a gradual change of the oscillatory pattern as the segments becomes more distal to the cell body. **(D)** Scatter plots showing a positive correlation between the relative $Ca^{2+}$ concentration changes in the cilia and soma (left) and between the duration of the cytosolic $Ca^{2+}$ increase and the change in cilia $Ca^{2+}$ concentration (right). **(E)** TIRF microscopy recording of GCaMP5G fluorescence change in the cilium (black) and soma (red) of islets cells following exposure to three 3-min KCl depolarizations (arrows). Data are means ± SEM for 8 cells from 1 islet and representative of 15 islets. **(F–H)** Color coded power spectra of ciliary (top) and somatic (bottom) frequency components with superimposed traces (black) of the whole compartments $Ca^{2+}$ activity recorded from individual cells exposed to different treatments (see horizontal lines above plots). **(I)** TIRF microscopy recording of GCaMP5G fluorescence from 12 cilia from 1 mouse islet (means ± SEM) following addition of 5 mM α-KIC. Shown to the right are recordings of cilia and cytosolic $Ca^{2+}$ from a matched cilia–soma pair within the islet. Representative of eight islets.

Glucose is the main physiological stimulus for insulin secretion from β-cells, and elevation of the glucose concentration causes regular cytosolic $Ca^{2+}$ oscillations driven by voltage-dependent $Ca^{2+}$ influx. We now find that these oscillations are mirrored by corresponding phase-locked, antiparallel oscillations of cilia $Ca^{2+}$, and this coupling is lost whenever the mechanisms generating the cytosolic activity are perturbed, implying a role of the membrane potential in ciliary $Ca^{2+}$ extrusion. A similar mechanism of isolation has been observed in olfactory neurons, which are specialized, multi-ciliated cells (Leinders-Zufall et al., 1997). Another relevant comparison is between the primary cilium and the dendritic spine, which is a highly specialized structure with a unique $Ca^{2+}$ signature and dimensions that resemble the cilium (1 μm long, 100 nm in diameter). $Ca^{2+}$ changes in dendritic spines do not propagate into the dendrite, largely due to efficient extrusion, and this mechanism is required for normal synaptic function (Müller and Connor, 1992; Sabatini et al., 2002). The isolation of the primary cilium of islet β-cells would be similarly important for the initiation of local $Ca^{2+}$ signaling. In addition to the isolation from cytosolic $Ca^{2+}$, we often observe compartmentalized $Ca^{2+}$ signaling within the cilium in the form of stable microdomains (see Fig. 6 D). It will be important to determine whether these microdomains are related to the recently described actin corrals that constitute functional subunits of the cilium (Lee et al., 2018) or if they reflect the distinct localization of ion channels or $Ca^{2+}$-binding proteins that might prevent unrestricted spreading of $Ca^{2+}$ along the cilium.

Activation of muscarinic receptors caused a dramatic increase of $Ca^{2+}$ at the base of the cilium, raising the possibility of a close apposition with the ER, which is the main intracellular $Ca^{2+}$ store. Although proximity between the ER and the cilia base has not been demonstrated, the ER plays a fundamental role in generating similar $Ca^{2+}$ microdomains at the immunological synapse, a structure that bears many similarities to the primary cilium (Cassioli and Baldari, 2019; Quintana et al., 2011). An alternative possibility is that $IP_3$ generated downstream of the muscarinic receptors instead mobilize $Ca^{2+}$ from the Golgi apparatus (Pinton et al., 1998), an organelle involved in both ciliogenesis and delivery of cargo to the mature cilium (Nachury and Mick, 2019). The apposition between the cilia base and an intracellular $Ca^{2+}$ store may also permit information flow in the opposite direction, where cilia-generated $Ca^{2+}$ signals could be amplified in the cytosol through a $Ca^{2+}$-induced $Ca^{2+}$ release

mechanism. Indeed, we do occasionally observe a small, spatially restricted, rise in cytosolic $Ca^{2+}$ as a $Ca^{2+}$ wave propagates through the cilium and reaches the base (see, e.g., Fig. 6 A). Interestingly, similar bidirectional communication has been proposed to occur between the cilium and cytosol in olfactory neurons through a mechanism involving cilia-proximal mitochondria (Zufall, 2012).

GABA is released from pancreatic β-cells in a glucose-independent manner that likely involves plasma membrane localized transporters (Menegaz et al., 2019). GABA exerts multiple effects on β-cells, including both acute modulation of insulin secretion and more long-term effects on proliferation and cell-fate maintenance (Korol et al., 2018; Wang et al., 2019). Most of these effects are thought to emanate downstream of ionotropic $GABA_A$ receptors, which seem to be the dominating GABA receptor in the cell body (Korol et al., 2018). We now show that islet β-cells also express $GABA_{B1}$ receptors, that the receptor expression is restricted to the primary cilium and that activation of the receptors triggers local $Ca^{2+}$ increases in the cilium. This very distinct localization to, and action in, primary cilia could easily have been overlooked in earlier studies. The $GABA_{B1}$ receptors are enriched at the base of the cilium, which is at odds with the fact that the GABA-induced $Ca^{2+}$ signaling typically originated from the medial to distal regions of the cilium. Perhaps, these $GABA_{B1}$ receptors represent a population leaving the organelle after activation, as has been described for other cilia receptors (Nachury and Mick, 2019). Alternatively, that segment may be a functional representation of the inversin compartment (Bennett et al., 2020), where the receptors are trapped in an inactive state. Agonist binding could relieve the inhibition, enabling diffusion to more distal segments where they could interact with signaling partners. Consistent with this notion, we observe receptor mobilization in response to short-term incubation with GABA. How GABA binding to $GABA_{B1}$ receptors on the cilia surface leads to an increase in cilia $Ca^{2+}$ is not entirely clear, but most likely involves a non-canonical pathway that does not require the $GABA_{B2}$ receptor subunit. This is supported by our failure to detect ciliary $GABA_{B2}$ by immunostaining, by the lack of effect of the $GABA_{B1/2}$ antagonist CGP-35348 on cilia $Ca^{2+}$ flashes (Wood et al., 2000), by the lack of $GABA_{B2}$ receptor expression in human islets (Rachdi et al., 2020), and by the insensitivity to pertussis toxin. While GABA is primarily thought of as a neurotransmitter and immune modulator, it has an evolutionary origin that predates

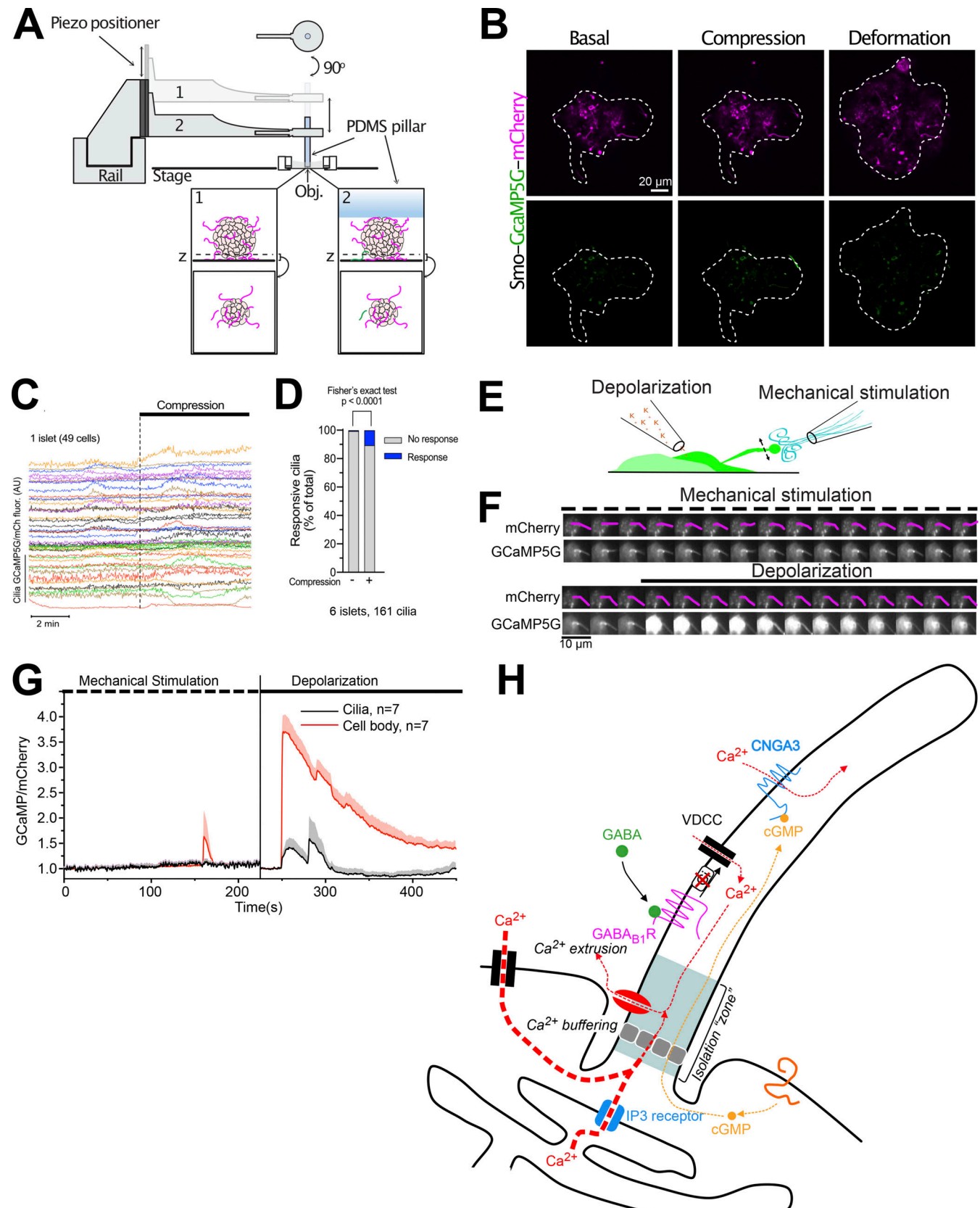

Figure 9. **Islet cell cilia do not respond with Ca²⁺ signaling following mechanical stimulation. (A)** Schematic drawing of the cell press used to compress islets on the stage of a confocal microscope. **(B)** Confocal micrographs of a whole islet expressing Smo-GCaMP5G-mCh. Upper row shows mCherry fluorescence and bottom row shows GCaMP5G fluorescence. Notice how both mild compression and deformation is without effect on both cilia and cytosolic Ca²⁺.

(C) Changes in cilia GCaMP5G/mCh fluorescence ratio during mild compression of a mouse islet ($n$ = 49 cells). (D) GCaMP5G/mCh ratio change following strong compression with islet deformation for the indicated number of islets and islet cells. (E) Schematic drawing of the principle of mechanical stimulation using a glass pipette. (F) Confocal microscopy images of a MIN6 cell expressing Smo-GCaMP5G-mCh following exposure to mechanical stimulation (top) and depolarization (bottom). (G) Means ± SEM ($n$ = 7) for the cilia and cell body GCaMP5G fluorescence change in response to mechanical stimulation and depolarization. The increase in cilia fluorescence upon depolarization is likely a contamination from the cytoplasm, which partially overlaps with the cilia in these recordings. (H) Proposed model for cilia-dependent $Ca^{2+}$ signaling in β-cells. The primary cilium is isolated against cytosolic $Ca^{2+}$ changes due to efficient $Ca^{2+}$ extrusion in the cilium. This enables the cilium to utilize $Ca^{2+}$ as an intrinsic signaling molecule. Activation of cilia-localized $GABA_{B1}$ receptors results in receptor mobilization into the cilium where it triggers local $Ca^{2+}$ influx in a process that depends on L-type $Ca^{2+}$ channel activation.

both the nervous and immune systems, and its appearance coincides with that of a $GABA_{B1}$-like receptor (Gou et al., 2012). The $GABA_{B1}$ receptor is ubiquitously expressed in human tissues, while $GABA_{B2}$ expression is largely restricted to the central nervous system (Human Protein Atlas; search terms GABBR1 and GABBR2). This implies that $GABA_{B1}$ receptors have important functions independent of the G-protein-coupled $GABA_{B2}$ receptor, and that these functions may depend on its ciliary localization. In some neurons, $GABA_B$ receptors are functionally coupled to voltage-dependent $Ca^{2+}$ channels, and activation of the receptors have both stimulatory and inhibitory effects depending on the type of cell and channel (Booker et al., 2018; Carter and Mynlieff, 2004; Laviv et al., 2011; Shen and Slaughter, 1999). L-type voltage-dependent $Ca^{2+}$ channels, which are the dominating type in β-cells, have also been found in the primary cilium of other cell types (Jin et al., 2014). Although L-type (CaV1.2) channels appear to be present in some islet cell cilia, the selective inhibition of these with the very potent blocker nifedipine failed to completely suppress the GABA-induced $Ca^{2+}$ influx. The influx was, however, sensitive to the structurally unrelated VDCC blockers verapamil and diltiazem. Although both of these drugs are potent L-type VDCC inhibitors, they function by blocking the pore of ion channels and also inhibit other types of VDCC (Laryushkin et al., 2021; Zhao et al., 2019). Our conclusion is therefore that $GABA_{B1}$-receptor activation is coupled to the activation of non-L-type VDCCs in the primary cilium of β-cells. Although the identity of this ion channel is currently unknown, both R-type and P/Q-type VDCCs are expressed in mouse islets cells (Rorsman et al., 2012). The mechanism of VDCC activation may be similar to that in certain neurons, where GABA facilitates voltage-dependent opening by lowering the activation threshold, a mechanism that also appears to be independent of $GABA_{B2}$ receptors (Shen and Slaughter, 1999). The primary cilium has a more positive membrane potential than the bulk plasma membrane (Delling et al., 2013), which indicates that it is insulated from changes in membrane potential. Consistent with this, direct depolarization caused $Ca^{2+}$ influx in the soma, but not in the primary cilia, of islet cells. It is therefore possible that activation of ciliary VDCCs does not involve changes in cilia membrane potential but rather changes in how voltage-dependent $Ca^{2+}$ channels sense membrane potential, such as alterations in the local lipid environment or posttranslational modification of channel subunits.

The biological significance of the GABA-mediated ciliary $Ca^{2+}$ signaling remains to be determined. GABA is released in a glucose-independent manner from islet cells (Menegaz et al., 2019); however, GABA-induced cilia $Ca^{2+}$ signaling is suppressed under conditions that stimulate insulin secretion, such as high glucose concentrations and direct depolarization. This implies that GABA effects on the cilium are temporally uncoupled from insulin secretion and most prominent under resting conditions. GABA has previously been demonstrated to affect cell fate properties in islet cells and also shown to stimulate proliferation and differentiation of mature β-cells (Ben-Othman et al., 2017), although these findings have been questioned in more recent studies (Ackermann et al., 2018; van der Meulen et al., 2018). Interestingly, the cilium is ascribed similar properties in many cell types (Irigoin and Badano, 2011). It is therefore tempting to speculate that the ciliary GABA pathway may crosstalk with other cilia signaling pathways, such as Wnt and Hedgehog, either indirectly through second messenger generation or through sensitization of cilia receptors, similar to the role of $GABA_B$ receptors in neurons and astrocytes (Boyer et al., 2009; Hirono et al., 2001; Nilsson et al., 1993). Our observation that a brief GABA stimulation leads to enrichment of the Hedgehog receptor Patched in the cilium is consistent with such crosstalk, although it remains to be proven that this is coupled to changes in downstream signaling. The mechanism of GABA action could also involve control of the intraflagellar transport machinery, which is regulated by ciliary $Ca^{2+}$ in flagellated unicellular organisms (Collingridge et al., 2013), or receptor extrusion through ectocytosis. Elucidating the mechanism and function of GABA action on primary cilia represents an important future research direction.

## Materials and methods

### Plasmids, adenoviruses, and reagents
The following plasmids were used: $Lyn_{11}$-R-GECO (Gandasi et al., 2017), $5HT_6$-G-GECO (Addgene plasmid 47499; gift from Takanari Inoue (Johns Hopkins University, Baltimore, MD) [Su et al., 2013]), 5HT6-Venus-CFP (Addgene plasmid 47501; gift from Takanari Inoue (Johns Hopkins University, Baltimore, MD) [Su et al., 2013]), ss-pHluorin-Smo (gift from Derek Toomre, Yale Univeristy, New Haven, CT [Kukic et al., 2016]), Epac-SH187 (gift from Kees Jalink, Netherlands Cancer Institute, Amsterdam, Netherlands [Klarenbeek et al., 2015]), pGEM-HE-h_bPAC_cmyc (Addgene plasmid 28134, gift from Peter Hegemann, Humboldt University, Berlin, Germany [Stierl et al., 2011]), RhGC(BE)-pGEM (Addgene plasmid 85469, gift from Peter Hegemann, Humboldt University, Berlin, Germany [Scheib et al., 2018]) and PfPKG (kind gift from Kjetil Wessel Andressen, Oslo University, Oslo, Norway [Calamera et al., 2019]). Smo-GCaMP5G-mCherry was generated by PCR-amplification of GCaMP5G (Addgene plasmid 31788, gift from Loren Looger,

University of Califonia, San Diego, CA [Akerboom et al., 2012]) using the following primers: GCaMP-EcoRI-Fwd (5′-TATATA GAATTCTAATGGGTTCTCATCATCA-3′) and GCaMP-SacII-Rev (5′-CTCTCTCCGCGGCTTCGCTGTCATCATT-3′) followed by digestion of the PCR product with EcoRI and SacII and ligation in between Smoothened and mCherry in mCherry-Smo (Addgene plasmid 55134, kind gift from Michael Davidson). S100G was PCR amplified from a human lung cDNA library and ligated to Smo-mCherry using SalI and SacII to make Smo-S100G-mCherry. Adenoviral particles (E5 serotype) carrying 5HT$_6$-G-GECO, Smo-GCaMP5G-mCh, bPAC, RhGC, Epac-S$^{H187}$, and PfPKG under the control of CMV-promoters were produced by Vector Biolabs. All salts, Mg-ATP, and α-toxin were from Sigma-Aldrich/Merck. GABA, diazoxide, methoxyverapamil, α-ketoisocaproic acid, nifedipine, clonidine, GLP-1, forskolin, baclofen, CGP35348, picorotoxin, Diltiazem, L-cis-diltiazem, pertussis toxin, Atrial natriuretic peptide, amiloride, ATP, carbachol, and vigabatrin were from Tocris. The following antibodies were used in the study: acetylated tubulin (T745, host: mouse, 1: 500; Sigma-Aldrich), IFT88 (F41236, host: rabbit, 1:200; NSJ Bioreagents), SSTR3 (E-AB-1607, host: rabbit, 1:200; Elabscience), Patched (ab53715, host: rabbit, 1:200; Abcam), Smoothened (ab113438, host: rabbit, 1:200; Abcam), GABA-B1 receptor (NBP1-52389, host: goat, 1:200; Novus Biologicals and AGB-001, host: rabbit, 1:200; Alamone Labs), GABA-B2 receptor (ab230136, host: rabbit, 1:200; Abcam), GABA-A receptor (224-103, host: rabbit, 1:200; Synaptic Systems), CNGA3 (ab2519323002150000, host: rabbit, 1:200; Abcam), γ-tubulin (T5326, host: mouse, 1:200; Sigma-Aldrich), Arl13b (ab136648, host: mouse, 1:300; Abcam), CaV1.2 (ACC-022, host: rabbit, 1:100; Alamone labs), CaV1.3 (ACC-005, host: rabbit, 1:100; Alamone labs), and CNGB1 (bs-12093R, host: rabbit, 1:100; Bioss antibodies). Secondary antibodies for confocal microscopy were anti-mouse 568 nm (A11004; Invitrogen) and anti-mouse 488 nm (A28175; Invitrogen), anti-rabbit 488 nm (A11034; Invitrogen), anti-goat 555 nm (A21432; Invitrogen); all secondary antibodies were diluted to 1:500. For super-resolution microscopy, the following secondary antibodies were used: anti-mouse Abberior Star 580 and anti-rabbit Abberior Star Red (Abberior GmbH). All secondary antibodies were diluted 1:300.

## Cell and islet culture and transfection
### MIN6 cell culture and transfection with plasmids and siRNA
The mouse β-cell line MIN6 (passages 18–30; Miyazaki et al., 1990) was cultured in DMEM (Life Technologies) supplemented with 25 mmol/liter glucose, 15% FBS, 2 mmol/l L-glutamine, 50 µmol/l 2-mercaptoethanol, 100 U/ml penicillin, and 100 µg/ ml streptomycin. The cells were kept at 37°C and 5% CO$_2$ in a humidified incubator. These cells were used as a complement to the primary mouse islets to confirm that observations made in islets cells corresponded to responses from β-cells. Prior to imaging, 0.2 million cells were resuspended in 100 µl Opti-MEM-I medium (Life technologies) with 0.2 µg plasmid (total) and 0.5 µl lipofectamine 2000 (Life technologies) and seeded in the center of a 25-mm poly-L-lysine–coated coverslip. The transfection reaction was terminated after 4–6 h by the

addition of 2 ml complete culture medium and cells were imaged 18–24 h later. For knockdown experiments, MIN6 cells were resuspended with 25 nM siRNA (Dharmacon, siGENOME) premixed with 1.6 µl/ml Lipofectamine RNAiMAX (Life technologies) in Opti-MEM I medium. After 3 h, the Opti-MEM I medium was replaced by DMEM growth medium containing 25 nM siRNA and 1.6 µl/ml Lipofectamine 3000 (Life technologies) and incubated overnight. For knockdown in MIN6 pseudoislets, MIN6 cell suspensions were mixed with siRNA/liposome complexes and allowed to spontaneously form pseudoislets by seeding into non-adherent plastic 12-well plates (see below). The following siRNA was used: GABBR1 (L-057519-02-0005) and CNGA3 (L-043122-00-0005). To inactivate Gi-signaling, islets were incubated for 18 h with 200 ng/ml pertussis toxin prior to imaging experiments.

### MIN6 pseudoislet formation and culture
Detached MIN6 cells (3 to 5 million/ml) were transferred to a non-adherent Petri dish (Sarstedt) with 5-ml culture medium and kept in culture for 5–7 d at 37°C in a humidified atmosphere with 5% CO$_2$ until they spontaneously formed cellular aggregates (pseudoislets).

### Mouse islet isolation and culture
Adult C57Bl6J mice (>8 mo) were sacrificed by CO$_2$ asphyxiation and decollation, and the pancreas was removed and put on ice prior to digestion with collagenase P and mechanical disaggregation at 37°C. The digestion was terminated by 1 ml BSA 0.1 g/ml, and islets were then separated from exocrine tissue by handpicking with a 10-µl pipette under a stereo microscope. Islets were cultured in RPMI 1640 medium with 5.5 mM glucose, supplemented with 10% fetal calf serum, 100 units/ml penicillin, 100 µg/ ml streptomycin for 2–5 d at 37°C in a humidified atmosphere with 5% CO$_2$. All procedures for animal handling and islet isolation were approved by the Uppsala animal ethics committee.

### Human islet culture
Human islets from normoglycemic cadaveric organ donors were kindly provided through the Nordic Network for Clinical Islet Transplantation. All experiments with human islets were approved by the Uppsala human ethics committee. Islets were cultured for up to 5 d in CMRL 1066 medium containing 5.5 mM glucose and supplemented with 10% fetal calf serum, 100 units/ ml penicillin, and 100 µg/ml streptomycin and kept at 37°C in an atmosphere of 5% CO$_2$ in humidified air.

### Viral transduction of islets and pseudoislets
Islets were transferred to a Petri dish containing a 200 µl drop of culture medium and 10 µl of high titration virus (>10$^{12}$–10$^{13}$ vp/ ml; Vector Biolabs). After 3 h, islets were transferred to a new Petri dish filled with 5 ml of fresh medium and kept in culture for 1–3 d to allow for biosensor expression before performing experiments.

### α-Toxin permeabilization and preparation of intracellular-like media
Intracellular-like media with buffered pH, [Ca$^{2+}$], and [Mg$^{2+}$] used in α-toxin permeabilization experiments contained: 6 mM

Na$^+$, 140 mM K$^+$, 1 mM (free) Mg$^{2+}$, 0–100 μM (free) Ca$^{2+}$, 1 mM Mg-ATP, 10 mM Hepes, 2 mM (total) EGTA, and 2 mM (total) nitrilotriacetic acid with pH adjusted to 7.00 at 22°C with 2 M KOH. The total concentration of Ca$^{2+}$ and Mg$^{2+}$ was calculated using the online version of MaxChelator (https://somapp.ucdmc.ucdavis.edu/pharmacology/bers/maxchelator/webmaxc/webmaxcE.htm). Media were made fresh on the day of experiment and kept on ice. For permeabilization, 25-mm poly-L-lysine-coated glass coverslips with transfected, adherent MIN6 cells were used as exchangeable bottoms in a modified Sykes-Moore open superfusion chamber that was mounted on the stage of a TIRF microscope (described below) and connected to a peristaltic pump that allowed rapid change of medium. Following change from normal, extracellular-like, medium (125 mM NaCl, 4.9 mM KCl, 1.3 mM MgCl$_2$, 1.2 mM CaCl$_2$, 25 mM Hepes1 mg/ml BSA with pH set to 7.4) to an intracellular-like medium (see below), the superfusion was interrupted and α-toxin was added directly to the chamber (final concentration ≈50 μg/ml). Permeabilization typically took 2–5 min, after which superfusion was started again, and the cells were exposed to intracellular-like buffers containing calibrated Ca$^{2+}$ concentrations or different concentrations of cyclic nucleotides while fluorescence from GCaMP5G/mCherry, Epac-S$^{H187}$, or PfPKG was recorded. These experiments were performed at ambient temperature (21–23°C).

## Fluorescence microscopy
### Confocal microscopy
A spinning-disk confocal microscope unit (Yokogawa CSU-10) mounted on an Eclipse Ti2 body (Nikon) was used to acquire pictures of immuno-labeled samples. The microscope was equipped with a CFI Apochromat TIRF 100×, 1.49 NA oil immersion objective from Nikon, and excitation light was provided by 491- and 561-nm diode-pumped solid state (DPSS) lasers from Cobolt (Hübner Photonics). Lasers were merged with dichroic mirrors, homogenized using a rotating, light-shaping diffusor and delivered to the CSU. Excitation light source was selected by electronic shutters (SmartShutter, Sutter Instruments) and emission light was separated using the following filters fitted into a filter wheel controlled by a Lambda 10-3 unit (Sutter Instruments): GFP/Alexa488 (530/50, Semrock), mCherry/Alexa561 (593LP; Semrock). Images were captured with a back-illuminated electron-multiplying charge-coupled device (EMCCD) camera (DU-888; Andor technology) using MetaFluor software (Molecular Devices). For the mechanical stimulation experiments shown in Fig. 3, alternative confocal microscope setups were used. For details, see section "Mechanical stimulation of islets and cells" below.

### TIRF microscopy
TIRF imaging was performed on a Nikon TiE microscope equipped with an iLAS2 TIRF illuminator for multi-angle patterned illumination (Gataca systems) and a 100×, 1.49-NA Apo-TIRF objective. Excitation light for GFP and mCherry was delivered by 488- and 561-nm diode-pumped solid-state lasers with built-in acousto-optical modulators, and light for bleaching or photorelease of Ca$^{2+}$ was delivered by a 405-nm DPSS laser (all from Coherent, Inc). Fluorescence was detected with a back-illuminated EMCCD camera (DU-897; Andor Technology) controlled by MetaMorph (Molecular Devices). Emission wavelengths were selected with filters (527/27 nm for GFP and 590 nm long-pass for mCherry) mounted in a filter wheel (Sutter Instruments). Islets and cells were mounted in an open perfusion chamber with temperature held at 37°C. FRET-based detection of cAMP (using Epac-S$^{H187}$) and cGMP (using PfPKG) was achieved using 445-nm excitation light (Cobolt Twist DPSS, 50 mW) and with emission detected at 483/32 and 542/27 nm (interference filters from Semrock). For fluorescence recovery after photobleaching analysis of indicator mobility, a 3 × 3–μm area within the cells was exposed to a 100 ms 405-nm light pulse to bleach the fluorophores, followed by continued acquisition for 120 s at 4 fps. Calculations of the mobile fraction ($Mf$) of each fluorescent protein were performed using the following formula:

$$Mf = 100 * \frac{(Fpc - Fbg)}{(F\infty c - Fbg)} * \frac{((F\infty - Fbg) - (F0 - Fbg))}{((Fp - Fbg) - (F0 - Fbg))}.$$

Fpc (whole-cell pre-bleach intensity), Fp (bleach region of interest [ROI] pre-bleach intensity), F∞c (asymptote of fluorescence recovery of the whole cell), Fbg (mean background intensity), F∞ (asymptote of the bleach ROI), and F0 (bleach ROI post-bleach intensity). The same approach was used for photorelease of caged Ca$^{2+}$ in islet cells that had been preincubated for 30 min in imaging buffer supplemented with 5 μM of the AM-ester-form of o-nitrophenyl-EGTA (Thermo Fisher Scientific).

### STED microscopy
Sub-diffraction microscopy was performed with a Stedycon STED super-resolution instrument (Abberior Instruments) with an Olympus BX63F upright motorized microscope body. The objective was a 100× / 1.56 NA (Olympus) and the STED depletion laser had a wavelength of 775 nm. Emission filters were 575–625 and 650–700 nm corresponding to Star 580 and Star Red fluorophores, respectively.

### Optogenetic production of cAMP and cGMP
Islets were transduced with adenoviruses carrying the blue-light-activated adenylate cyclase (bPAC) or the yellow-light-activated guanyl cyclase (RhGC) under the control of cytomegalovirus (CMV) promoters. Activation of cAMP production was achieved by 491-nm light exposure delivered through evanescent wave excitation on the stage of a TIRF microscope (see below for details). Activation light was delivered in pulses (100 ms every 2 s). Because of spectral overlap between activation of bPAC and excitation of the cilia-targeted Ca$^{2+}$ indicator 5HT$_6$-GGECO1, we were unable to record pre-stimulatory activity in islets expressing bPAC. This was instead assessed in parallel control experiments performed on islets lacking bPAC expression. Activation of cGMP production was achieved by 561-nm light exposure delivered through evanescent wave excitation on the stage of a TIRF microscope (see below for details). Activation light was delivered in pulses (100 ms every 2 s) and the impact on Ca$^{2+}$ signaling was determined using the co-expressed cilia-targeted Ca$^{2+}$ indicator 5HT$_6$-GGECO1.

## Mechanical stimulation of islets and cells

### Islet compression

Islet compression experiments were performed with a custom-built manipulator referred to as the cell press (O'Callaghan et al., 2022). The device consists of 3D-printed parts, a linear piezoelectric positioner (SLC-1730, SmarAct GmbH), which is operated through a MCS2 manual controller (SmarAct), and a PDMS compression pillar, which was cast from a 3D-printed mold. The cell press was secured directly to the microscope behind the translation stage of an LSM700 confocal microscope (Zeiss). A relative zero position for the PDMS pillar was first established in an islet-free dish by bringing the surface of the pillar in contact with the bottom of the dish and recording the position displayed on the MCS2 controller. The pillar was retracted and a dish containing islets expressing Smo-GCaMP5G-mCherry was placed on the stage. Individual islets were imaged using a Plan-Apochromat 63×/1.4 (Zeiss) objective with the pinhole set to produce a 7.5-μm-thick optical z-slice. Islets were imaged at their point of contact with the cover-glass, and image acquisition was performed using Zen software (Zeiss). Islets were imaged prior to the initiation of compression experiments to establish a baseline for the GCamP5G and mCherry fluorescence signals. Compression experiments were performed by initially lowering the pillar to within ~200–300 μm from the zero position, after which the pillar was lowered in 50-μm steps until initial contact was made with the upper surface of the islet, after which compression continued in 5-μm steps. Compression continued until regions of the islet that were previously not visible (as they were above the imaged z-slice height) were pushed into the field of view; this point onward was referred to as deformation. This occasionally resulted in additional cilia being relocated into the field of view; however, image analysis was limited to cilia for which baseline GCamP5G and mCherry fluorescence signals had been established. Islets were compressed in the presence of 1.3 mM $Ca^{2+}$ and were indicated and depolarized with 30 mM KCl.

### Fluid flow stimulation

For mechanical stimulation of MIN6 cell cilia, we used borosilicate pipettes of 0.5–1μM diameter to puff extracellular solution locally (mM composition: NaCl 138, $CaCl_2$ 2.6, $MgCl_2$ 1.2, KCl 5.6, Hepes 10, and glucose 3; pH 7.4, 300 mOsm). The pipette solution was puffed toward the cilia by gently applying positive pressure with the mouth. A second borosilicated pipette with the same resistance was used to induce plasma membrane depolarization by puffing a solution containing (in mM): NaCl 63, $CaCl_2$ 2.6, $MgCl_2$ 1.2, KCl 8.6, Hepes 10, and glucose 3; pH 7.4, 300 mOsm. The mechanical stimulation pipette was located >5 μm from the cilia head and the high potassium pipette 10–15 μM from the cell body at the opposite side of the mechanical stimulation pipette. The mechanical stimulation was intermittently applied every 1 s during 225 s. During the recording, the cilia were allowed to rest from the stimulation 1–3 times to see possible changes at the cilia fluorescence baseline. Then, cells were let to rest 25 s without stimulation before inducing membrane depolarization for 200 s. The high potassium solution was puffed by applying constant positive pressure (3.7 psi) controlled by

a Pico Spritzer (ValveLink8.2, Automate Scientific). The experiments were carried out at 32°C. The samples were observed with a 40× Zeiss objective with a 1.2 NA. A Prime 95B CMOS camera (Teledyne Photometrics) was used to acquire the images, and GCaMP and mCherry fluorophores were excited by 473- and 561-nm lasers (Cobolt). Red and green channel images were acquired simultaneously at a 2 Hz frequency, and the emission light was separated onto the two halves of the camera chip using an image splitter (Optical Insights) with a cutoff at 565 nm (565dcxr, Chroma) and emission filters (FF01-523/610, Semrock; and ET525/50 m and 600EFLP, both from Chroma). Cilia and cell body intensities over time were analyzed by homemade Macros in Fiji (ImageJ). Results are reported as the Δ(GCaMP/mCherry) intensity ratio.

## Immunofluorescence

Culture medium was removed and samples were washed with Dulbecco's PBS at 37°C and fixed with 4% PFA in PBS for 5 min, then permeabilized with 0.5% Triton X-100 in PBS for 10 min and blocked with 2% BSA in PBS for 1 h. Primary antibodies were diluted in blocking solution and samples incubated for 2 h at RT (cells grown on coverslips were placed on a 100-μl drop while 20–50 isolated islets were put into 200 μl). Next, a wash with 2% BSA in PBS for 5 min was followed by 1 h incubation with secondary antibodies (same way as primary). Subsequently, samples were washed with 2% BSA in PBS for 5 min and mounted using ProLong Gold Antifade Mountant with DAPI (Thermo Fisher Scientific). To preserve islets structural integrity, they were embedded into a 2% agar in PBS gel after being put into mounting medium.

## Image analysis

All images were analyzed using the Fiji version of the ImageJ Software (Schindelin et al., 2012) or IgorPro (Wavemetrics). Fluorescence line profiles were produced by drawing a segmented line over the cilium and part of cytoplasm for obtaining intensity values that were then scaled to the cytoplasmic or background levels (ratiometric data was obtained in a similar way by dividing profiles corresponding to different channels). Line width was set to match cilium diameter (~3–5 pixels for confocal and TIRF, 7–9 pixels STED); for transverse profiles (Fig. 1 F), line width was expanded to include the whole basal compartment (117 pixels). Analysis of live cell $Ca^{2+}$ imaging data (TIRF) was performed by drawing ROIs containing the cilium from where mean values of pixel's intensities were extracted from each time frame. Kymographs were created using the corresponding function in Fiji; for ratiometric ones, the GCaMP kymograph was divided by the corresponding mCherry kymograph. For the analysis presented in Fig. 7, O–R, kymographs and quantification of extrusion kinetics were performed using built-in functions in IgorPro. Center of mass calculation was performed on straightened cilia (Fiji function and analysis tool); the length coordinate was then normalized to total length of each individual cilium. The spontaneous $Ca^{2+}$ activity in the cilium was analyzed using the *Neural Activity*[3] (Prada et al., 2018). This tool consists of a peak-detection algorithm based on the continuous wavelet transform. The video is first divided in a grid

that can be as dense as a single pixel per square; each zone in the grid is analyzed as an independent $Ca^{2+}$ trace. Each trace is transformed using the Mexican-hat wavelet and the spectrum ridges are detected and grouped as a tree formation. Then, those ridges are filtered in order to discard those of minor importance and keep the only the ones that contain possible activity peaks. Lastly, the peaks in each of the remaining ridges are discarded based on the signal-to-noise ratio, leaving only those with a power over certain threshold. The phase relationship between cilia and soma $Ca^{2+}$ was determined using the *WaveletComp* package in R (Roesch and Schmidbauer, 2018). A representative $Ca^{2+}$ recording is selected for each structure, one for the cilia and one for the soma. Each of those traces is transformed with the continuous wavelet transform. This transformation results in a complex spectrum that contains information of magnitude and phase of the signals, which means that it is then possible to compare the phase spectrums of both signals and establish whether there exists a sustained phase relationship between both of them. That relationship is then presented in radians; see the package vignette for a thorough explanation.

### Statistical analysis

All data were processed using GraphPad Prism version 9.1.2 for Mac (GraphPad Software, https://www.graphpad.com). Samples were tested for normal distribution using Kolmogorov-Smirnov test to fulfill the criteria for performing Student's *t* test (two-tailed; paired or unpaired), and logarithmic transformations were performed in cases of non-normal distributions. Where suitable, non-parametric Wilcoxon signed-rank tests were performed instead. For small data sets, non-parametric Mann–Whitney *U*-tests were performed instead. For multiple comparisons where data distribution was normal, one-way ANOVA was used followed by Tukey's or Dunnett's post hoc tests. When data distribution was non-normal, Friedman test or Kruskal–Wallis test with Dunn's multiple comparisons was used instead. Fisher's exact test was used for categorical data. Graphs show mean values ± SEM unless otherwise stated. We did not perform a power analysis to predetermine sample sizes. The experiments (with the exception of those shown in Fig. 4, G–I, and Fig. 6, O–Q) were not randomized and the investigators were not blinded. For more details, see figure legends.

### Online supplemental material

Fig. S1 shows experiments related to Fig. 1 characterizing the GABA-B1 receptor distribution to primary cilia and the redistribution in response to agonist binding. Fig. S2 is related to Fig. 2 and shows calculations of biosensor enrichment in the cilium and effects of biosensor expression on cilia morphology. Fig. S3 is related to Fig. 3 and shows characterizations of the cilia-localized $Ca^{2+}$ indicator and of cilia pH. Fig. S4 is related to Fig. 5 and has control experiments validating the antibody against CNGA3 and showing the cellular distribution of CNGB1. It also contains control experiments showing the effect of $Ca^{2+}$ channel blockers on cytosolic $Ca^{2+}$ in mouse islets and the effect of pertussis toxin treatment on Gi-induced cAMP lowering and immunostaining of L-type voltage-dependent $Ca^{2+}$ channels in mouse islets. Fig. S5 contains experiments related to Fig. 6

showing characterization of spontaneous cilia $Ca^{2+}$ signaling using an alternative $Ca^{2+}$ indicator (5HT6-GGECO1) and control experiments showing that overexpression of this sensor does not interfere with Hedgehog-dependent exit of Patched from the cilium or with Hedgehog-mediated cilia lengthening. Fig. S6 relates to Fig. 7 and shows characterization of $Ca^{2+}$ clearance mechanisms in the cilium and confirmatory experiments using an alternative cilia-localized $Ca^{2+}$ indicator. Fig. S7 is related to Fig. 8 and characterizes $Ca^{2+}$-independent effects of glucose on $Ca^{2+}$-indicator behavior in the primary cilium.

### Data availability

Data that support the findings in this study as well as all plasmids and reagents generated in this study are available from the corresponding author upon request.

## Acknowledgments

We are grateful to Bryndis Birnir, Anders Tengholm, Erik Gylfe, and all the members of the O. Idevall-Hagren lab for important input and suggestions on this work. We also acknowledge the excellent help with STED imaging provided by the staff at the BioVis facility at Uppsala University. 3D printing was performed at U-PRINT: Uppsala University's 3D-printing facility at the Disciplinary Domain of Medicine and Pharmacy.

This work was supported by grants from the Swedish Research Council, the Novo-Nordisk Foundation, the Swedish Diabetes Foundation, the Family Ernfors Foundation, European Foundation for the Study of Diabetes/Lilly, and Exodiab to O. Idevall-Hagren, and the Swedish Cancer Society (grant 20 1285 PjF) and by Sweden's Innovation Agency VINNOVA (grant number 2019-00029) to J. Kreuger.

The authors declare no competing financial interests.

Author contributions: Conceptualization: G.M. Sanchez and O. Idevall-Hagren. Methodology: G.M. Sanchez, J. Prada, P. O'Callaghan, O. Dyachok, S. Echeverry, B. Xie, S. Barg, J. Kreuger, T. Dandekar, O. Idevall-Hagren. Investigation: G.M. Sanchez, T.C. Incedal, J. Prada, P. O'Callaghan, O. Dyachok, S. Echeverry, Ö. Dumral, P.M. Nguyen, B. Xie, O. Idevall-Hagren. Visualization: G.M. Sanchez, T.C. Incedal, J. Prada, P. O'Callaghan, O. Dyachok, S. Echeverry, Ö. Dumral, P.M. Nguyen, B. Xie, O. Idevall-Hagren. Funding acquisition: J. Kreuger, O. Idevall-Hagren. Writing—Original draft: G.M. Sanchez, O. Idevall-Hagren. Writing—review and editing: G.M. Sanchez, T.C. Incedal, P. O'Callaghan, J. Kreuger, O. Idevall-Hagren.

Submitted: 19 August 2021

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

# Supplemental material

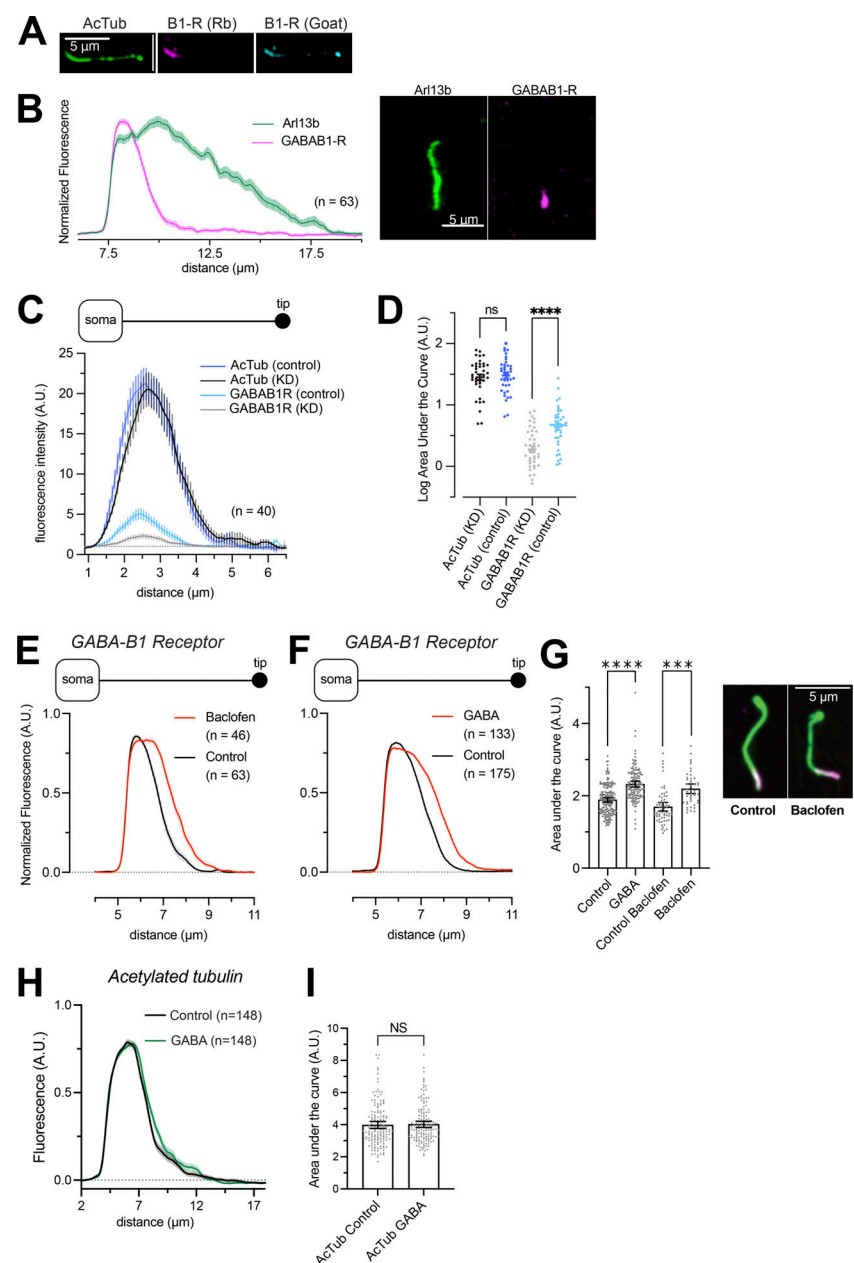

Figure S1.  **GABA_B1 receptor distribution in the cilium after stimulation. (A)** Immnofluorescence staining of a mouse islet cell cilium against acetylated tubulin (green), GABA_B1 receptors (Rabbit monoclonal antibody, magenta), and GABA_B1 receptors (Goat polyclonal antibody, blue). **(B)** Quantifications of line profiles drawn along cilia of mouse islet cells immunostained for Arl13b (green) and GABA_B1 receptors (magenta). An example cilium is shown to the right. **(C)** Quantifications of line profiles drawn along cilia of MIN6 cells immunostained for acetylated tubulin and GABA_B1 receptors. The two lines are form cells treated for 48 h with control siRNA or siRNA against the GABA_B1 receptor (KD). Notice how the GABA_B1 immunoreactivity is reduced in cells treated with GABA_B1 R siRNA ($n$ = 40 cells from three experiments). **(D)** Cilia fluorescence intensity from control and GABA_B1 receptor KD MIN6 cells immunostained for acetylated tubulin (left) or GABA_B1 receptors (right). One-way ANOVA with Šidák's multiple comparison test (**** $P < 0.0001$; $n$ = 40). **(E)** Quantifications of GABA_B1 receptor fluorescence intensity and distribution along primary cilia of mouse islet cells under control conditions (black) and following 10-min stimulation with 100 nM of the GABA receptor agonist baclofen (red). Data presented as means ± SEM for the indicated number of cells. **(F)** Quantifications of GABA_B1 receptor fluorescence intensity and distribution along primary cilia of mouse islet cells under control conditions (black) and following 10-min stimulation with 100 nM GABA (red). Data presented as means ± SEM for the indicated number of cells. **(G)** GABA_B1 receptor distribution in mouse islet cilia under control conditions and following stimulation for 10 min with 100 nM GABA or 100 nM Baclofen. Data presented are calculations of the AUC of line profiles drawn along cilia of mouse islet cells immunostained for acetylated tubulin and GABA_B1 receptors. Representative images of cilia from a control cell and a cell exposed to baclofen are shown to the right (green, acetylated tubulin; magenta, GABA_B1 receptor). Bars show means ± SEM and statistical significance was assessed with ANOVA with multiple comparisons; $n$ = 63 (control) and $n$ = 46 (baclofen); $n$ = 175 (control) and $n$ = 133 (GABA). *** $P < 0.001$, **** $P < 0.0001$. **(H)** Quantifications of acetylated tubulin fluorescence intensity distribution in mouse islet cells under control conditions (black) and following 10-min stimulation with 100 nM GABA (green). Data presented as means ± SEM for 148 cilia from 15 islets and 3 animals. **(I)** Length distribution of acetylated tubulin in mouse islet cells under control conditions and following stimulation for 10 min with 100 nM GABA. Data presented are calculations of the AUC of line profiles drawn along the acetylated tubulin immunoreactivity (means ± SEM). Two-tailed Student's unpaired $t$ test; $n$ = 148 (control) and $n$ = 148 (GABA).

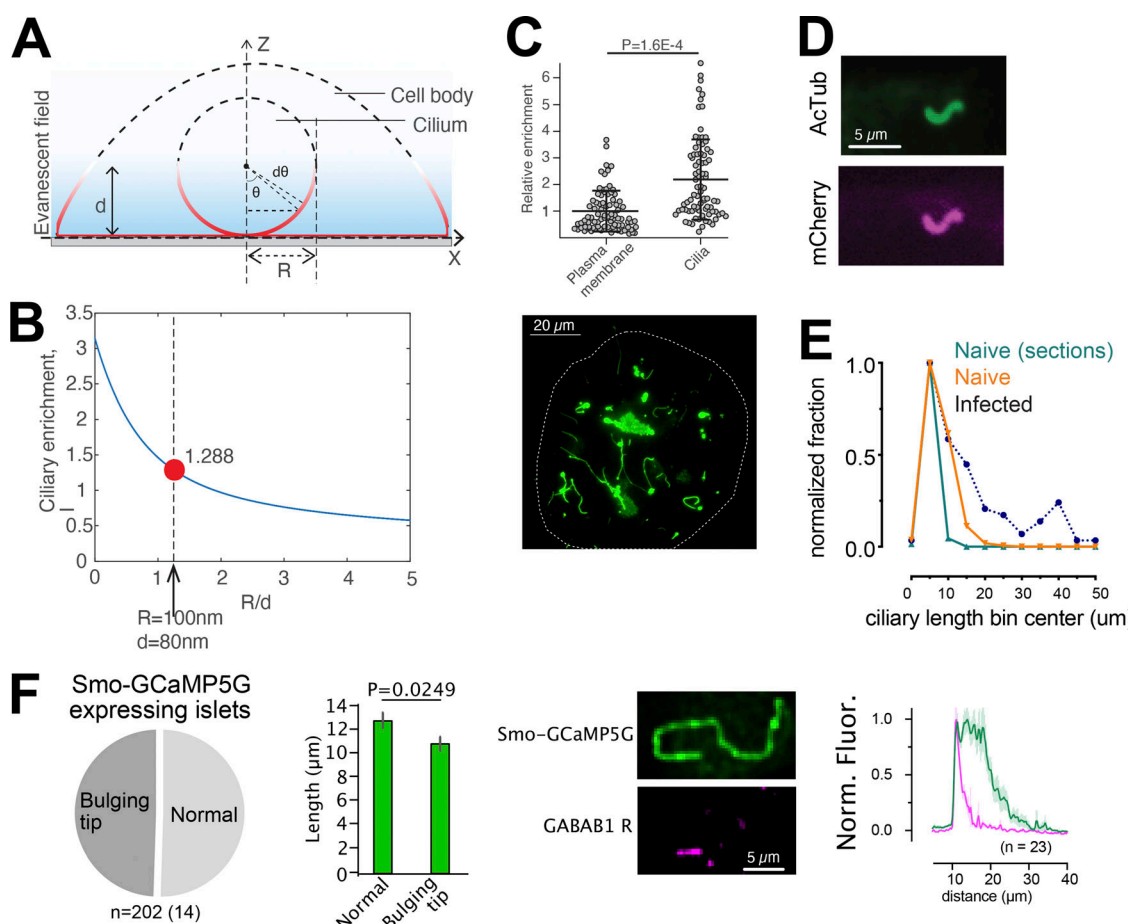

Figure S2. **Characterization of a cilia-targeted Ca²⁺ sensor. (A and B)** Principle for determining the enrichment of Smo-GCaMP5G-mCh within the primary cilium of islet cells imaged by TIRF microscopy. With a radius of the cilium of 100 nm and an evanescent wave penetration depth (d) of 80 nm, we estimate that a ratio of cilia-to–plasma membrane fluorescence >1.288 indicate cilia enrichment. **(C)** Quantifications of fluorescence intensities of Smo-GCaMP5G-mCh from the plasma membrane and cilia of islet cells. The average cilia-to–plasma membrane fluorescence ratio is 2, which indicate strong cilia enrichment (means ± SEM; n = 113 soma and 95 cilia; Student's two-tailed unpaired t test). **(D)** Immunofluorescence staining of a MIN6 cells expressing Smo-GCaMP5G-mCherry using antibodies against acetylated tubulin (green) and mCherry (magenta). **(E)** Measurements of cilia length in islets cells from mouse pancreatic sections (green), isolated mouse islets (red), and mouse islets transduced with Smo-GCaMP5G-mCh for 48 h (black). **(F)** Quantification of cilia morphologies in islet cells expressing Smo-GCaMP5G-mCh (means ± SEM; n = 202 cilia from 14 islets; two-tailed unpaired Student's t test). Images show an example of the distribution of GABA_{B1} receptors in a cilium expressing Smo-GCaMP5G-mCh and quantification of the distribution along the cilia of 23 islet cells.

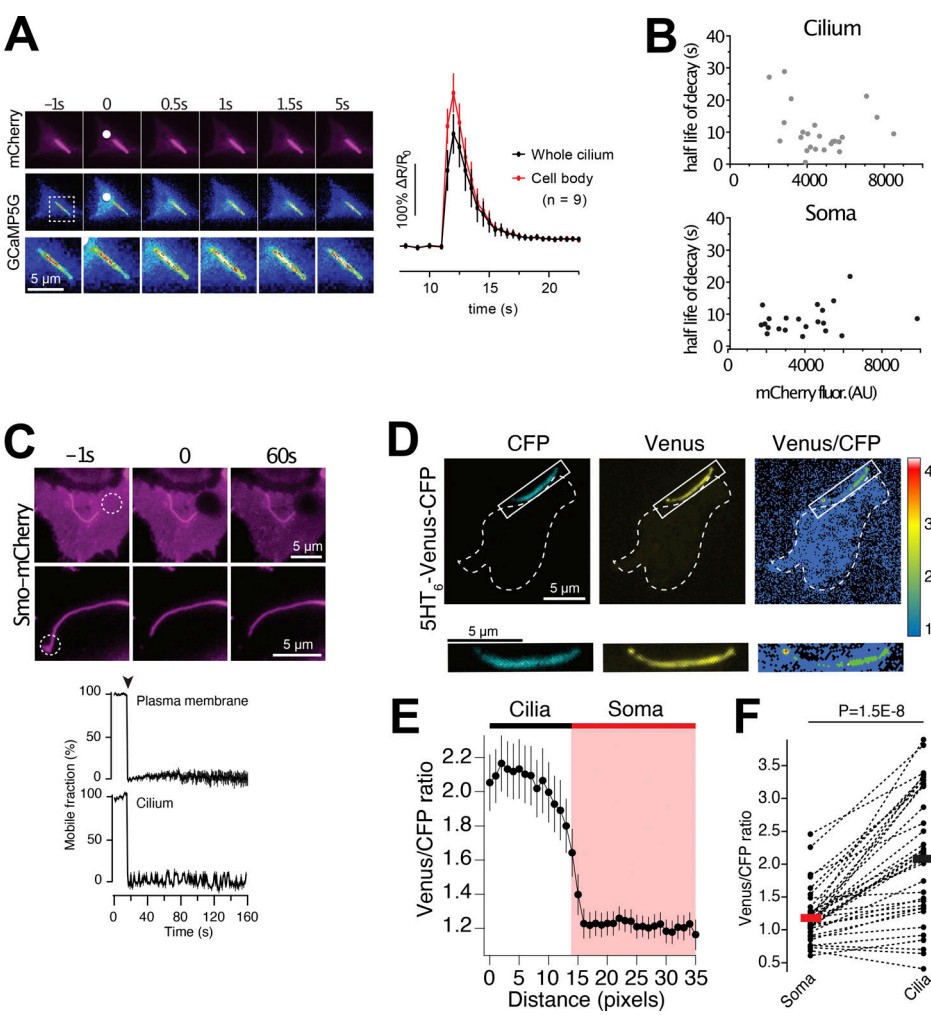

Figure S3. **Characterization of a cilia-targeted Ca²⁺ sensor. (A)** TIRF microscopy images of a MIN6 cell expressing Smo-GCaMP5G-mCh and loaded with caged Ca²⁺. White dot marks site of local UV-induced photolysis of the caged Ca²⁺. Graph below shows the GCaMP5G fluorescence increase in the cell body (red) and cilium (black) following uncaging (means ± SEM, $n$ = 9 cells). **(B)** Scatter plots showing the lack of correlation between the expression level of Smo-GCaMP5G-mCh and half-life of Ca²⁺ decay in the cilium (top) and cell bodies (bottom) of MIN6 cells following photolysis of caged Ca²⁺ ($n$ = 24 cells). **(C)** TIRF microscopy images showing lack of recovery of mCherry fluorescence following local photobleaching (dashed white circle) of MIN6 cells expressing Smo-GCaMP5G-mCh). Graphs show the recovery of fluorescence after photobleaching in the plasma membrane (top) and cilium (bottom; means ± SEM; $n$ = 14 cells). **(D)** TIRF microscopy images of a MIN6 cell expressing the cilia-targeted pH-sensor 5HT₆-Venus-CFP under resting conditions. Notice that the venus/CFP ratio is higher in the cilium. **(E)** Means ± SEM ($n$ = 30 cells) of line profiles drawn from cilia to cell body across the base of the cilium in TIRF microscopy images of MIN6 cells expressing 5HT₆-Venus-CFP. Notice that the venus/CFP ratio is higher in the cilium, indicating more alkaline pH. **(F)** Cilia and soma venus/CFP fluorescence ratios in MIN6 cells expressing 5HT₆-Venus-CFP ($n$ = 30 cells, Student's two-tailed paired $t$ test).

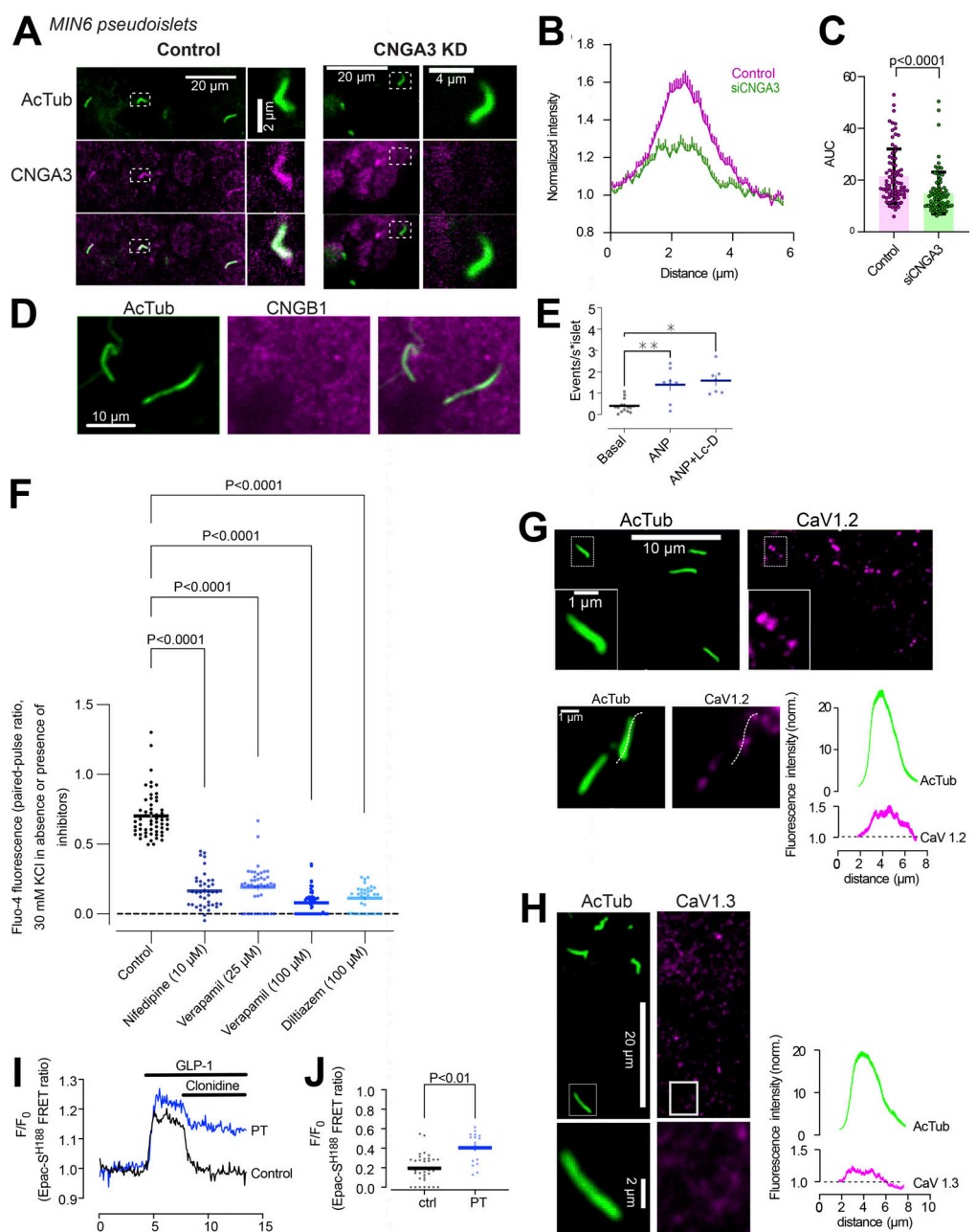

Figure S4. **Characterization of inhibitors used in the study. (A)** Confocal microscopy images of MIN6 cell pseudoislets transfected with control (left) or CNGA3 (right) siRNA and immunostained for acetylated tubulin (green) and CNGA3 (magenta). **(B)** Quantifications of the ciliary CNGA3 intensity (line profile) from control (magenta) and CNGA3 KD (green) MIN6 cell pseudoislets ($n$ = 78 and 81 cilia from two individual experiments). **(C)** Ciliary CNGA3 fluorescence intensity in control and CNGA3 KD MIN6 cell pseudoislets (means ± SEM for 78 and 81 cilia; two-tailed unpaired Student's $t$ test). **(D)** Confocal microscopy images of a mouse islet immunostained for acetylated tubulin (green) and CNGB1 (magenta). **(E)** Quantification of ciliary $Ca^{2+}$ activity in mouse islets expressing 5HT6-GGECO1 following exposure to 100 nM ANP alone or in combination with 100 µM L-cis-diltiazem (Lc-D; means ± SEM; $n$ = 215, 122, and 93 cilia from 7–15 islets; * P < 0.05, ** P < 0.01. Statistical significance assessed with Wilcoxon signed-rank test). **(F)** Mouse islets loaded with the $Ca^{2+}$ indicator Fluo-4 were imaged on the stage of an epifluorescence microscope while being exposed to two consecutive 3-min pulses of 30 mM KCl. The second pulse was performed in the presence of the indicated L-type $Ca^{2+}$ channel inhibitor. The data presented are the ratios between the Fluo-4 fluorescence increase during the second and first pulse. Each data point represents one cell (means ± SEM; $n$ = 3 islets per condition; ANOVA with Dunnett's post hoc test). **(G)** Confocal microscopy images of a mouse islets immunostained for acetylated tubulin (green) and CaV1.2 (magenta). Quantification of line profiles drawn along the cilium as shown in the lower figure are shown at the bottom right ($n$ = 75 cilia; $n$ = 2). **(H)** Confocal microscopy images of a mouse islets immunostained for acetylated tubulin (green) and CaV1.3 (magenta). Quantification of line profiles drawn along the cilium as shown in the lower figure are shown at the bottom right ($n$ = 80 cilia; $n$ = 2). **(I)** TIRF microscopy recordings of the FRET ratio of the cAMP sensor Epac1[S188] expressed in two mouse islet β-cells during stimulation of cAMP production with 100 nM GLP-1 and inhibition via clonidine-induced activation of inhibitory G-proteins. Traces are from one control cell (black) and one cell treated with pertussis toxin (blue). **(J)** Quantification of the reduction in GLP-1–induced cAMP formation following addition of clonidine in control islet cells and islet cells treated with pertussis toxin. Notice that the inhibitory effect of clonidine is reduced following pertussis toxin treatment (means; $n$ = 33 and 18 cells from 5 experiments; two-tailed unpaired Student's $t$ test).

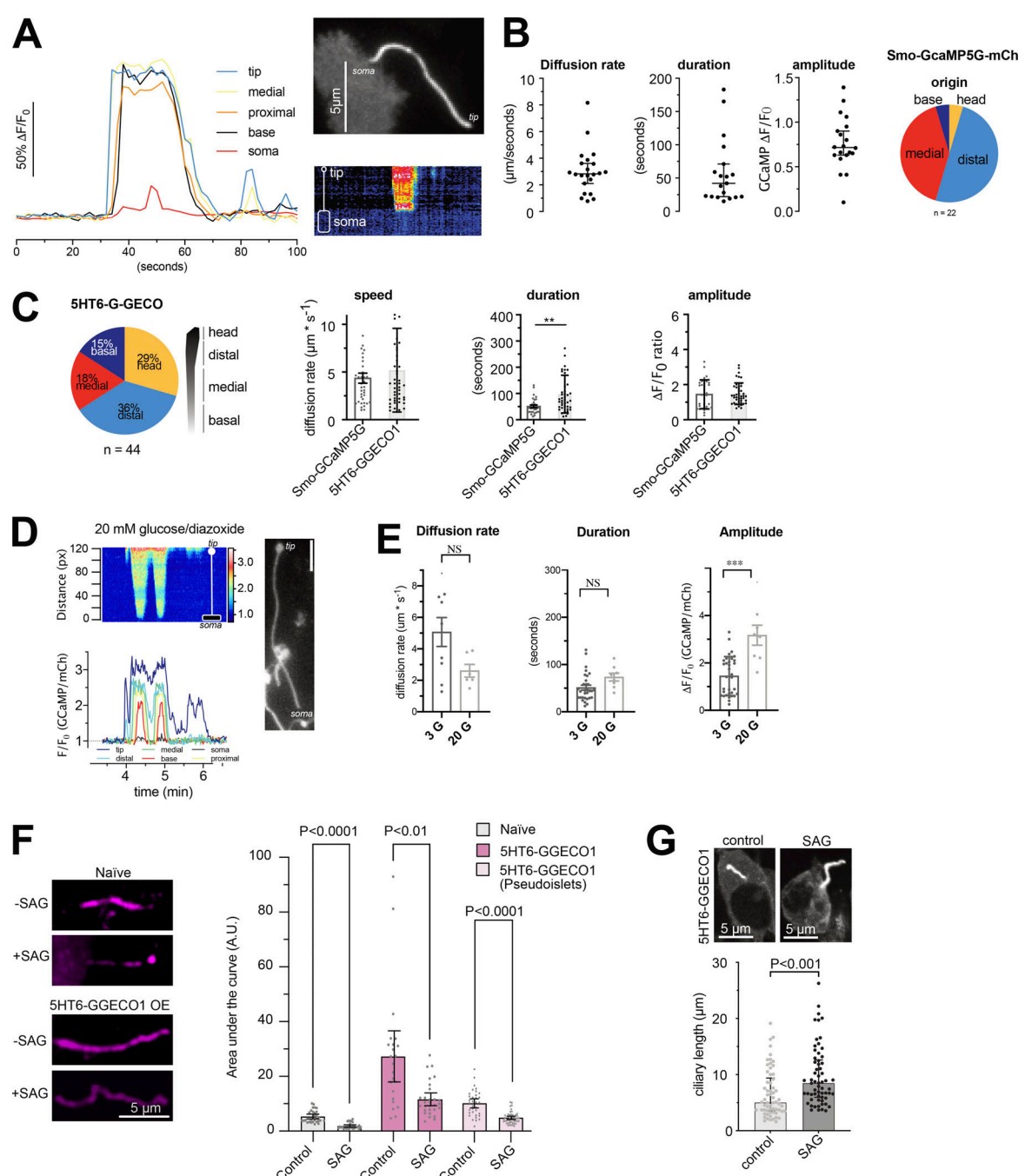

Figure S5. **Characterization of spontaneous Ca²⁺ flashes in MIN6 cells and mouse islet cells. (A)** Spontaneous Ca²⁺ flash in the primary cilium of a MIN6 cells expressing Smo-GCaMP5G-mCherry. Notice how the flash originates in the tip of the cilium and propagates toward the base and how this coincides with a small local rise of Ca²⁺ in the cilia-adjacent cytosol. **(B)** Characteristics of cilia Ca²⁺ flash diffusion rate, duration, amplitude, and site of origin in MIN6 cells expressing Smo-GCaMP5G-mCh (means ± SEM; $n$ = 22 cilia). **(C)** Characteristics of cilia Ca²⁺ flashes detected with the Ca²⁺ sensor 5HT₆-G-GECO1. Pie chart shows the origin of Ca²⁺ influx and the bar graphs show the diffusion rate, duration of the events, and amplitude of the response compared to Smo-GCaMP5G-mCh (means ± SEM; $n$ = 24–42 cells). Statistical significance was determined using a two-tailed unpaired Student's $t$ test (** $P$ < 0.01). **(D)** A spontaneous Ca²⁺ flash in a mouse islet cell expressing Smo-GCaMP5G-mCh and kept in a buffer containing 20 mM glucose and 250 µM diazoxide. Notice how the wave propagates from tip to base and how the strength of the flash is diminished as it approaches the base. **(E)** Characteristics of cilia Ca²⁺ flash diffusion rate, duration, and amplitude in mouse islet cells expressing Smo-GCaMP5G-mCh and kept in 3 mM glucose or 20 mM glucose supplemented with diazoxide. Statistical significance was determined using a two-tailed unpaired Student's $t$ test (*** $P$ < 0.001). Diffusion rate: 3G $n$ = 37 cells from 30 islets; 20G $n$ = 6 cells from 4 islets. Duration: 3G $n$ = 31 cells from 30 islets; 20G $n$ = 8 cells from 4 islets. Amplitude: 3G $n$ = 33 cells from 30 islets; 20G $n$ = 8 cells from 4 islets (means ± SEM). **(F)** Quantification of immunofluorescence images of mouse islets and MIN6 pseudoislets stained for the cilia receptor Patched under control conditions and following 24 h stimulation with SAG (100 nM). Bar graphs in gray represent naive islets and in magenta islets or pseudoislets transduced with a viral vector encoding the cilia localized Ca²⁺ sensor 5HT₆-GGECO1. Statistical significance was assessed with Brown-Forsythe and Welch one way ANOVA (means ± SEM; $n$ = 29, 27, 23, 29, 34, and 48). **(G)** Measurements of cilia length in MIN6 cells expressing 5HT6-GGECO treated for 24 h with solvent (control) or 1 µM SAG (means ± SEM; $n$ = 61 cilia for control and 64 cilia for SAG; statistical significance assessed with Kolmogorov-Smirnov test).

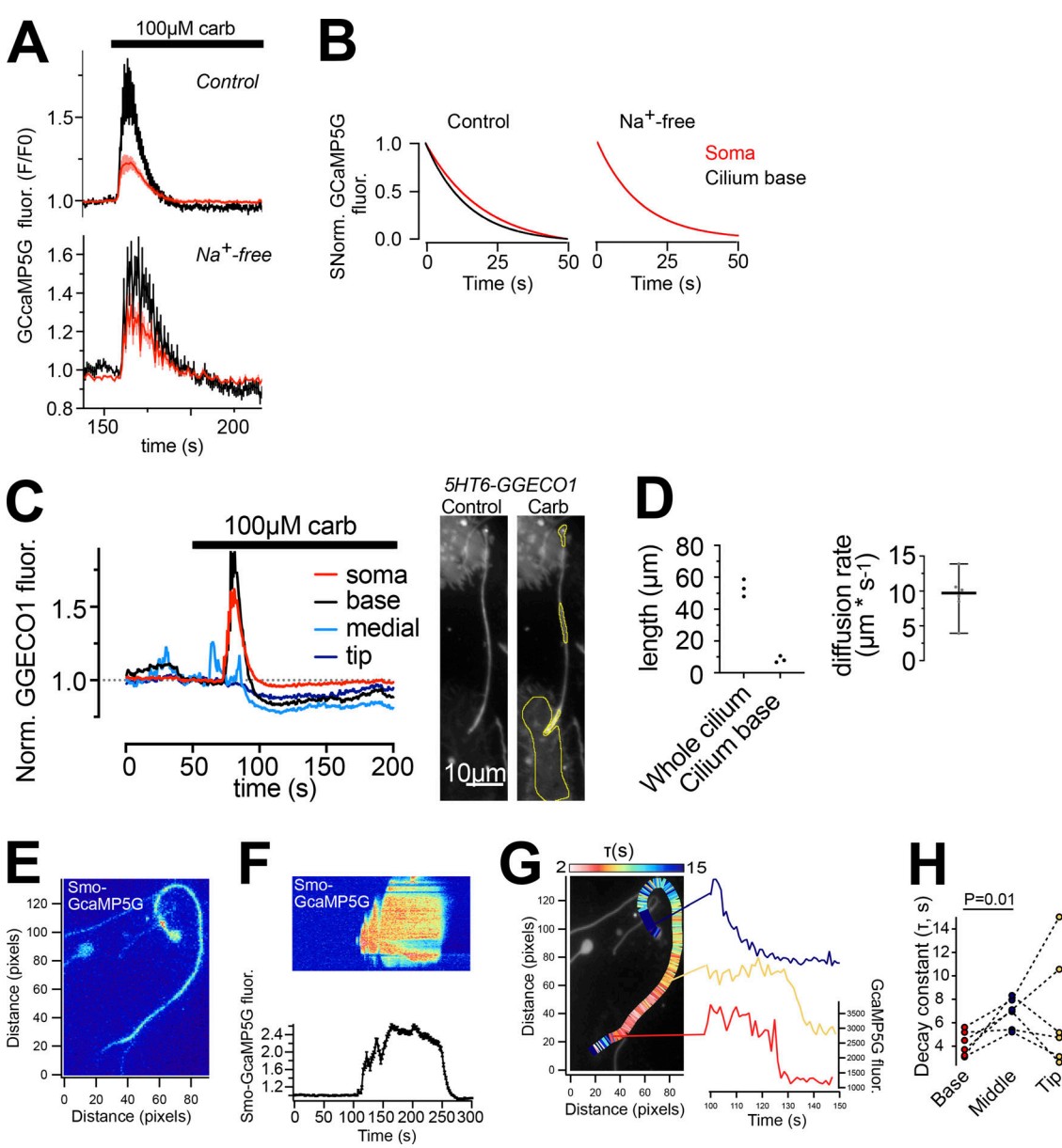

Figure S6. **Carbachol-induced changes in cilia Ca²⁺.** **(A)** Ca²⁺ recordings from mouse islet cell cilia (black) and soma (red) expressing Smo-GCaMP5G-mCh following exposure to 100 µM carbachol in normal or Na⁺-free extracellular medium (means ± SEM for 18 cells from 9 islets from 5 animals, control; 12 cells, 4 islets, 2 animals). **(B)** Curve fittings of the Ca²⁺ decay in soma and cilia under normal and Na⁺-free conditions. **(C)** Ca²⁺ recordings from a mouse islet cilium and soma expressing 5HT₆-G-GECO1 during stimulation with 100 µM carbachol. Images to the right show the G-GECO1 fluorescence before and during application of carbachol. **(D)** Quantifications (means ± SD) of cilia length and the length of the ciliary compartment where carbachol elicited a rise of Ca²⁺. To the right is Ca²⁺ diffusion rates determined in cilia expressing 5HT₆-G-GECO1. **(E)** TIRF micrograph of an islet cell cilium expressing Smo-GCaMP5G-mCherry. **(F)** Normalized kymograph from (top), and average Smo-GCaMP5G-mCherry fluorescence change in (bottom), the cilium in E during a spontaneous Ca²⁺ flash. **(G)** The same cilium as in E that has been pseudo-colored to illustrate the rate of Ca²⁺ extrusion during the declining phase of the spontaneous Ca²⁺ flash. The colors show the tau values (rates) for each pixel-wide segment of the entire cilia length, where white/red represents the fastest rate and blue represents the slowest rate. Individual intensiometric traces of 5HT6-GGECO1 fluorescence change from three distinct segments of the cilium (base, middle, and tip) are shown to the left. Notice that the rate of extrusion is fastest at the cilia base. **(H)** Rate of Ca²⁺ decay in three cilia segments (base, middle [40 pixels from base], and tip). Each point corresponds to one cilium and lines indicate individual cilium.(n = 6 cilia, two-tailed paired Student's t test.

Figure S7. **Ca²⁺-independent changes in GCaMP5G fluorescence induced by 20 mM glucose. (A)** Ratiometric recording (black) of GCaMP5G (green) and mCherry (magenta) fluorescence from a mouse islet cell cilium expressing Smo-GCaMP5G-mCh. Notice that glucose causes an immediate increase in GCaMP5G fluorescence but is without effect on mCherry fluorescence. **(B)** Ratiometric (GCaMP5G/mCherry) recording from the cilium of a mouse islet cell exposed to 11 mM glucose followed by the addition of hyperpolarizing diazoxide (250 µM) and carbachol (10 µM). Notice that the addition of diazoxide suppresses the glucose-induced Ca²⁺ oscillations but does not bring the GCaMP5G/mCh ration back to resting levels. **(C)** TIRF microscopy recording of GCaMP6 fluorescence from a transgenic mouse islet β-cell expressing GCaMP6 under control of the insulin promoter. The islet was exposed to a step increase in the surrounding glucose concentration, from 3 to 20 mM, followed by addition of the hyperpolarizing agent diazoxide. Notice that in contrast to Smo-GCaMP5G-mCh (see Fig. S2 A), glucose does not cause an immediate increase in GCaMP fluorescence, and the glucose-induced increase in GCaMP fluorescence is suppressed to resting levels in the presence of diazoxide. **(D)** Fluorescence changes from α-toxin–permeabilized MIN6 cells expressing Smo-GCaMP5G-mCh (green/dashed black) or plasma membrane-anchored R-GECO (Lyn₁₁-R-GECO; magenta) following exposure to intracellular-like buffers containing the indicated Ca²⁺ and ATP concentrations. **(E)** Quantifications of GCaMP5G and Lyn₁₁-R-GECO fluorescence changes in permeabilized cells exposed to the indicated intracellular buffers (n = 13–16 cells, ** P < 0.01, two-tailed paired Student's t test). **(F)** Venus/CFP ratio changes in the cell body (red) and cilia (black) of MIN6 cells expressing cilia-targeted 5HT₆-Venus-CFP and exposed to an increase in the surrounding glucose concentration from 3 to 20 mM (means ± SEM of 21 cells). **(G)** Venus/CFP ratio changes in the cell body (red) and cilia (black) of MIN6 cells expressing cilia-targeted 5HT₆-Venus-CFP and exposed to 20 mM NH₄Cl, which causes strong alkalinization of the cytosol (means ± SEM of 21 cells).

