## [Peer Review File · The Journal of Cell Biology]

The β -cell primary cilium is an autonomous Ca^{2+} compartment for paracrine GABA signalling

Gonzalo Sanchez, Tugce Incedal, Juan Prada, Paul O'Callaghan, Oleg Dyachok, Santiago Echeverry, Özge Dumral, Phuoc My Nguyen, Beichen Xie, Sebastian Barg, Johan Kreuger, Thomas Dandekar, and Olof Idevall-Hagren

Corresponding Author(s): Olof Idevall-Hagren, Uppsala University

Review Timeline:

Submission Date:	2021-08-19
Editorial Decision:	2021-10-05
Revision Received:	2022-04-11
Editorial Decision:	2022-06-13
Revision Received:	2022-09-01
Editorial Decision:	2022-10-03
Revision Received:	2022-10-07

Monitoring Editor: Maxence Nachury

Scientific Editor: Lucia Morgado-Palacin

Transaction Report:

DOI: <https://doi.org/10.1083/jcb.202108101>

October 5, 2021

Re: JCB manuscript #202108101

Dr. Olof Idevall-Hagren
Uppsala University
Medical Cell Biology
Husargatan 3
BMC, Box 571
Uppsala 75123
Sweden

Dear Dr. Idevall-Hagren,

Thank you for submitting your manuscript entitled "The β -cell primary cilium is an autonomous Ca^{2+} compartment for paracrine GABA signalling". The manuscript has been evaluated by expert reviewers, whose reports are appended below. We thank you for your patience while we were assessing the reviews. Unfortunately, after an assessment of the reviewer feedback, our editorial decision is against publication in JCB.

You will see that, although all three reviewers express interest in your premise that ciliary Ca^{2+} waves are physically restricted to the cilium via GABA signaling, they raise several major concerns that precludes from publication at JCB, including the lack of direct evidence supporting the role of Na/Ca^{2+} exchangers and GABA-B1 receptor in ciliary Ca^{2+} signals, as noted by reviewer #1, and of appropriate physiological models to demonstrate the relevance of your findings, which questions the presented model as voiced by reviewers #2 and #3. We feel that the points raised by the reviewers are more substantial than can be addressed in a typical revision period. If you wish to expedite publication of the current data, it may be best to pursue publication at another journal.

Given interest in the topic, we would be open to resubmission to JCB of a significantly revised and extended manuscript that fully addresses the reviewers' concerns and is subject to further peer-review. If you would like to resubmit this work to JCB, please contact the journal office to discuss an appeal of this decision or you may submit an appeal directly through our manuscript submission system. You are also welcome to submit a revision plan so we can give you feedback on its suitability, but please note that this will be treated as an appeal. Please note that priority and novelty would be reassessed at resubmission.

Regardless of how you choose to proceed, we hope that the comments below will prove constructive as your work progresses. We would be happy to discuss the reviewer comments further once you've had a chance to consider the points raised in this letter. You can contact the journal office with any questions, cellbio@rockefeller.edu.

Thank you for thinking of JCB as an appropriate place to publish your work.

Sincerely,

Maxence Nachury
Monitoring Editor
Journal of Cell Biology

Lucia Morgado-Palacin, PhD
Scientific Editor
Journal of Cell Biology

Reviewer #1 (Comments to the Authors (Required)):

This manuscript studies calcium signaling in primary cilia of beta cells in isolated pancreatic islets and in primary cilia of Min6 cells. The authors find that the primary cilium is a compartmentalized signaling organelle, which is in agreement with previously published data. However, contrary to recent findings the authors show that cytoplasmic Ca^{2+} does not back-propagate into the ciliary compartment in this specific cell type. Lastly the authors find that GABA generates Cilia specific Ca^{2+} signals, likely by GABA-B1 receptors via an unknown mechanism.

This manuscript is very interesting and novel since it investigates for the first time the Ca^{2+} signaling within primary cilia of beta cells in detail. Beta cells have been reported to have long primary cilia and diabetes is one hallmark of ciliopathies. The function

of primary cilia on beta cells is only poorly understood, hence this study is highly relevant for JCB.

While I really like the first part of the paper I am a bit dissatisfied with the second part. Using the non-specific blocker Amiloride the authors conclude that Na/Ca²⁺ exchangers are modulating the Ca²⁺ oscillations. Amiloride is a nonspecific ion channel blocker, so it is difficult to assign the Amiloride effects to exchangers. Also, so far no Na/Ca²⁺ exchangers have been reported in primary cilia. Olfactory cilia are definitely a different class of cilia. Thus more data is needed to claim that Na/Ca²⁺ exchangers are more widely expressed in primary cilia.

Second, while GABA-dependent ciliary signaling is very interesting, the manuscript would definitely benefit from a more detailed analysis of the origin of ciliary Ca²⁺ signals. I am not sure I follow the GABA-B1 receptor arguments. The localization does not seem to fit the origin of Ca²⁺ influx (distal vs proximal?). Are there any ionotropic GABA receptors in the cilium that may generate the Ca²⁺ signal directly?

If we follow the GABA-B1 receptor argument the authors offer a few possibilities in the discussion how the receptor may couple to downstream ion channels (CNG channels). These possibilities could be addressed experimentally.

Overall I like the study but feel that more precise conclusions would make this manuscript much stronger.

Reviewer #2 (Comments to the Authors (Required)):

In this manuscript, Sanchez and colleagues use genetically encoded fluorescent reporters to assess the calcium dynamics in primary cilia of pancreatic beta cells from isolated pancreatic islets. They report that:

- 1) In contrast to previous reports in other cell lines (Delling et al., 2013; DeCaen et al., 2013), primary cilia of pancreatic beta cells contain lower resting calcium concentrations than the beta cell cytoplasm.
- 2) Again in contrast to previous reports (Su et al., 2013; Delling et al., 2013), the primary cilium calcium environment in beta cells is functionally separated from the cytoplasm by calcium extrusion mechanisms, such that calcium signals from the cytoplasm are restricted to the ciliary base.
- 3) In agreement with this observation, spontaneous calcium spikes in beta cell primary cilia do not propagate to the cytoplasm.
- 4) These calcium spikes are mediated via GABA-B1 receptors.

The topic is clearly of great interest to the bigger community of cell biologists interested in calcium signaling, beta cell and cilia biology. As far as I can judge the presented experiments, they appear very difficult and have been performed rigorously. However, even though the study is performed in large parts in isolated islets *ex vivo*, I have strong reason to believe that the authors have been studying highly artificial "cilia" due to high overexpression of the fluorescent reporter transgene. Therefore, I am questioning the overall design of the study, the relevance of their findings, and the proposed mechanistic insight. As I consider this an absolutely fundamental flaw of the presented study, I do not recommend publication in *The Journal of Cell Biology*.

Major points:

- Supplemental Figure 1H shows that the consequence of transducing pancreatic islets with their calcium sensor Smo-GCaMP5G-mCherry are massively long "primary cilia" (sometimes branched, see Figs. 2, S3) that are far from the physiological situation. Such unphysiological cilia are known to result from overexpression of cilia-localized membrane proteins, such as specific GPCRs, including Smo. Importantly, the high overexpression leads not only to extreme lengthening (as presented here) but most importantly also to an alteration of the ciliary protein composition, such that it is unclear whether the studied cilia actually contain the known or suggested proteins that may explain observed effects (ion channels and transporters). Therefore, all data resulting from the analysis of such artificially long cilia may be leading to false interpretations. This includes (although not all micrographs contain scale bars): Figs. 1H, 2C, 2F, 2H, 3B, 4C, 5A, 5I, 5J, 6D, 7C, 7D, 7F, 7G, 7I, S1C, S1L, S1K, S3C, S3D. Moreover, as the original micrographs are not presented, it is not unlikely that the following figures were also derived from artificially long and therefore non-physiological cilia: Figs. 1J, 1K, 2A, 2B, 2D, 2E, 2G, 2I, 3C, 3D, 3F, 3G, 4A, 4B, 4D, 4E, 4F, 4G, 4H, 4I, 4J, 5B, 5C, 5D, 5E-H, 6A-C, 6, 7E, 7H. While abnormally long cilia might indeed be irrelevant to assess some of the basic properties of the used reagents, presented in Figs. 1F, 1G, all other results should be reevaluated.
- Supplemental Figure 1K showcases the difficulties with the experimental approach, as the calcium sensor fused to Smo does not show recovery after photobleaching, which is well documented for virtually all cilia membrane proteins that are not part of ciliary subcompartments, such as the transition zone. This includes ciliary GPCRs, such as Smo or SSTR3 (Ye et al., 2013).
- Since the presented data are conflicting with a number of previous studies (see above) it is not sufficient to base all interpretations on a single technique that investigates extremely long cilia, which represent artifacts of the experimental approach.
- The comparisons of cilia with their respective cell bodies appears extremely difficult in the light of the immensely long cilia, such that from the presented data it is unclear whether one can trace back an individual cilium to the cell body it emerges from (see Figs. 1I-K, 2A,B,D,E,G,I, Figs. 5 and 6). This criticism is showcased in Fig 4C, where the extremely long cilium seems next to a cell body, but whether they are connected is uncertain. Along the same line, an assessment of tip vs. base localization seems extremely difficult without appropriate markers (see Figs. 2, 4C, 5, S3).
- Changes observed in cilia subdomains are very hard to interpret since the "base" area is not well specified and often appears as long as a cilium of physiological length, for example see Fig. 2F.
- Some changes in fluorescent signal intensities are somewhat ambiguous, as in presented micrographs often very long cilia are located in part above a cell body (or other structure). Often case, increases in cilia intensities are only visible directly above

other structures that also show a change in fluorescence intensity (see Figs. 2C, 2F top middle, 2Hb).

- There is only circumstantial evidence for a Ca^{2+} extrusion mechanism.

Minor points:

- Fig. 2D: unlike stated in the text, the effect on cytosolic Ca^{2+} seems more pronounced than the effect on ciliary Ca^{2+} .
- Figs S4 shows quantifications of fluorescence signals relative to the soma in μm , however, as cilia will have different lengths, a relative position (relative to the full length, as in Figs. 7E and 7I) seems more appropriate than a mere distance.
- Fig. S1D: unclear whether the structure shown is part of a cell, a larger area should be shown.
- The topology of Smo is nicely confirmed, but the reasoning is unclear. Smo topology is well known. Here, Smo topology has been determined for a transgene that shows a very different domain arrangement than the calcium sensor used (Fig. S1).
- The $\text{Na}^{+}/\text{Ca}^{2+}$ exchanger is not a direct target of amiloride. Amiloride targets sodium channels (and at higher concentration $\text{Na}^{+}/\text{H}^{+}$ exchanger). If this is known to affect the $\text{Na}^{+}/\text{Ca}^{2+}$ exchanger in beta cells, the original publication should be referenced.
- The finding that only GABA-B1 could be identified in cilia is exciting and surprising, as the functional receptor is believed to consist of a heterodimer of B1 and B2 subunits. If a functional B1 receptor homomer was reported in literature, please give the reference.
- All micrographs should contain scale bars.

Reviewer #3 (Comments to the Authors (Required)):

Sanchez et al. measure calcium ions inside primary cilia of beta-cells of isolated pancreatic islets. They report that calcium ions are regulated in these primary cilia independently from the contiguous cytosol of the same cells. They also identify GABA-B1 receptor to be exclusively expressed in these primary cilia and contribute to calcium increase also exclusive to the primary cilia. This unique regulation is achieved with the help of $\text{Na}^{+}/\text{Ca}^{2+}$ exchanger, revealed by pharmacological inhibitor experiments. The manuscript is generally written well and experimental designs are mostly reasonable. There are interesting observations of potential importance, along with some technical advances. However, this reviewer could not be excited about the findings for the following reasons:

1. Weak rationale: "Functional studies of the mammalian primary cilium have primarily been carried out in cell lines, with the exception of pioneering work conducted with isolated olfactory neurons". This sentence describes their motivation to perform calcium imaging using "primary" preparations rather than cell lines. However, there are several points that do not make sense. First, the authors may have forgotten citing a work by Mizuno et al. (PMID: 32743070), in which they not only performed calcium imaging in isolated mouse tissues, but also elegantly demonstrated the physiological significance of such cilia calcium. Second, there seems to be no discussion of whether and how different sample preparations (primary cultures vs. cell lines vs. intact animal) affect calcium signaling inside primary cilia. It is also puzzling to learn that the authors used MIN6 cell lines to validate the findings from islet experiments. Third, it remains unclear whether the present isolated islets is sufficiently physiologically relevant for functional studies of beta cells. Thus, the claimed significance of using primary culture does not seem to be adequately justified.
2. Lack of functional characterization: While there are interesting observations, it is not clear how any of these are linked to physiological functions of islets such as insulin secretion. Therefore, this reviewer could not judge physiological significance of these observations.
3. Potential pitfall: As the above Mizuno et al. clearly demonstrated, confusion in the cilia calcium field originates at least partially from trivializing the importance of binding affinity of biosensors against calcium ions. Little to no detectable calcium signal does not mean that there is no calcium increase. It is simply non-detectable with the sensor used. A biosensor with a smaller K_d (i.e., higher affinity) could detect the otherwise pseudo-negative signal. The authors tested only one sensor, namely GCaMP5G, and discussed obtained data qualitatively, which is prone to misinterpretation.

Minor: Main texts describing Fig. 4D-I are apparently misaligned with figure numbers.

Point-by-point answer to reviewers' comments

Reviewer #1 (Comments to the Authors (Required)):

This manuscript studies calcium signaling in primary cilia of beta cells in isolated pancreatic islets and in primary cilia of Min6 cells. The authors find that the primary cilium is a compartmentalized signaling organelle, which is in agreement with previously published data. However, contrary to recent findings the authors show that cytoplasmic Ca²⁺ does not back-propagate into the ciliary compartment in this specific cell type. Lastly the authors find that GABA generates Cilia specific Ca²⁺ signals, likely by GABA-B1 receptors via an unknown mechanism.

This manuscript is very interesting and novel since it investigates for the first time the Ca²⁺ signaling within primary cilia of beta cells in detail. Beta cells have been reported to have long primary cilia and diabetes is one hallmark of ciliopathies. The function of primary cilia on beta cells is only poorly understood, hence this study is highly relevant for JCB.

While I really like the first part of the paper I am a bit dissatisfied with the second part.

Using the non-specific blocker Amiloride the authors conclude that Na/Ca²⁺ exchangers are modulating the Ca²⁺ oscillations. Amiloride is a nonspecific ion channel blocker, so it is difficult to assign the Amiloride effects to exchangers. Also, so far no Na/Ca²⁺ exchangers have been reported in primary cilia. Olfactory cilia are definitely a different class of cilia. Thus more data is needed to claim that Na/Ca²⁺ exchangers are more widely expressed in primary cilia.

Our original conclusion that the extrusion at the cilia base were dependent on the NCX relies on the findings that extrusion is accelerated when the cell is depolarized and that it is impaired in the presence of amiloride. We agree with the reviewer that amiloride is a very non-specific inhibitor, and that the data presented was insufficient to draw such a conclusion. We have now performed additional experiments where we show that that Na⁺ replacement with choline chloride results in slower Ca²⁺ clearance from the cilium, indicating that the process is Na⁺-dependent (Fig. 7L,M and Suppl. Fig. 4). Because this is still not proof that the extrusion is mediated by NCX, we have removed this statement in the manuscript and instead discuss it in a broader context of Na⁺-dependent extrusion. We have also performed more in-depth analysis of Ca²⁺ dynamics in the cilium (Fig. 7O-Q and Suppl. Fig. 4) and find that the extrusion rate varies along the length of the cilium and is fastest close to the cilia base, indicating that enhanced extrusion contributes to generate a unique Ca²⁺ signature in the cilium. In addition, we also believe that buffering may play a role in isolating the cilium from the cytosol. We now show in Fig. 7N that expression of a cilia-localized Ca²⁺ binding protein reduces the ability of Ca²⁺ to equilibrate between cytosol and cilium.

Second, while GABA-dependent ciliary signaling is very interesting, the manuscript would definitely benefit from a more detailed analysis of the origin of ciliary Ca²⁺ signals. I am not sure I follow the GABA-B1 receptor arguments. The localization does not seem to fit the origin of Ca²⁺ influx (distal vs proximal?). Are there any ionotropic GABA receptors in the cilium that may generate the Ca²⁺ signal directly?

If we follow the GABA-B1 receptor argument the authors offer a few possibilities in the discussion how the receptor may couple to downstream ion channels (CNG channels). These possibilities could be addressed experimentally.

The reviewer raises a very good point. As we show in the manuscript, we detect neither the GABAB2 receptor or GABAA receptors in the cilium, but only GABAB1 receptors (Fig. 1A).

This implies that the receptor in the cilium functions in a non-canonical fashion. Interestingly, looking at the human tissue and cell line expression of GABAB receptor subunits in the human protein atlas we find that whereas most tissues express the GABAB1 receptor, only some express its bona fide partner GABAB2 receptors, suggesting unique roles of GABAB1. Perhaps part of this function could be related to cilia Ca²⁺ signaling. We have now further explored the connection between GABAB1 receptors and cilia Ca²⁺ signaling and found that we can initiate Ca²⁺ flashes in the cilium resembling those induced by GABA when elevating cGMP levels (either receptor-triggered or by a light-activated guanyl cyclase) but not cAMP levels (by light-activate adenylate cyclase). Moreover, we have found that CNG3 channels are enriched in the primary cilium in a pattern that well matches the observed Ca²⁺ responses in the cilium. However, these channels do not seem to be involved in the response to GABA, which is instead mediated by L-type voltage-dependent Ca²⁺ channels. These results are presented in the completely new figure 5. In addition, we have also included experiments showing that siRNA-mediated knockdown of the GABAB1 receptor abolishes the GABA-induced cilia Ca²⁺ signaling (Fig. 4C).

As pointed out by the reviewer, the mismatch between GABAB1 localization (basal cilium) and the site of origin of the calcium transients (medial to distal) in mouse islets is puzzling. On the contrary, GABAB1 in MIN6 cells (either grown as monolayers or aggregates) is distributed all along the cilium, matching the initiation points of Ca²⁺ transients (Fig. 1I).

When stimulated with agonists, MIN6 ciliary responses are immediate (Fig. 4A,B,E) while in mouse islets there is a delay preceding the increase in ciliary calcium activity (Fig. 4F). We believe that this dichotomy reflects functional aspects of ciliary signaling which may involve distinct subciliary compartments where the receptor is accumulated but precluded from interacting with downstream partners (proximal) as opposed to distal segments of the cilium where GABAB1 diffuses upon activation and encounters its partners.

Overall I like the study but feel that more precise conclusions would make this manuscript much stronger.

We thank the reviewer for these encouraging words. We have completely reworked the manuscript, including changing the order of how results are presented and rewriting of the discussion.

Reviewer #2 (Comments to the Authors (Required)):

In this manuscript, Sanchez and colleagues use genetically encoded fluorescent reporters to assess the calcium dynamics in primary cilia of pancreatic beta cells from isolated pancreatic islets. They report that:

- 1) In contrast to previous reports in other cell lines (Delling et al., 2013; DeCaen et al., 2013), primary cilia of pancreatic beta cells contain lower resting calcium concentrations than the beta cell cytoplasm.
- 2) Again in contrast to previous reports (Su et al., 2013; Delling et al., 2013), the primary cilium calcium environment in beta cells is functionally separated from the cytoplasm by calcium extrusion mechanisms, such that calcium signals from the cytoplasm are restricted to the ciliary base.
- 3) In agreement with this observation, spontaneous calcium spikes in beta cell primary cilia do not propagate to the cytoplasm.
- 4) These calcium spikes are mediated via GABA-B1 receptors.

The topic is clearly of great interest to the bigger community of cell biologists interested in calcium signaling, beta cell and cilia biology. As far as I can judge the presented experiments, they appear very difficult and have been performed rigorously. However, even

though the study is performed in large parts in isolated islets ex vivo, I have strong reason to believe that the authors have been studying highly artificial "cilia" due to high overexpression of the fluorescent reporter transgene. Therefore, I am questioning the overall design of the study, the relevance of their findings, and the proposed mechanistic insight. As I consider this an absolutely fundamental flaw of the presented study, I do not recommend publication in The Journal of Cell Biology.

Major points:

• Supplemental Figure 1H shows that the consequence of transducing pancreatic islets with their calcium sensor Smo-GCaMP5G-mCherry are massively long "primary cilia" (sometimes branched, see Figs. 2, S3) that are far from the physiological situation. Such unphysiological cilia are known to result from overexpression of cilia-localized membrane proteins, such as specific GPCRs, including Smo. Importantly, the high overexpression leads not only to extreme lengthening (as presented here) but most importantly also to an alteration of the ciliary protein composition, such that it is unclear whether the studied cilia actually contain the known or suggested proteins that may explain observed effects (ion channels and transporters). Therefore, all data resulting from the analysis of such artificially long cilia may be leading to false interpretations. This includes (although not all micrographs contain scale bars): Figs. 1H, 2C, 2F, 2H, 3B, 4C, 5A, 5I, 5J, 6D, 7C, 7D, 7F, 7G, 7I, S1C, S1L, S1K, S3C, S3D. Moreover, as the original micrographs are not presented, it is not unlikely that the following figures were also derived from artificially long and therefore non-physiological cilia: Figs. 1J, 1K, 2A, 2B, 2D, 2E, 2G, 2I, 3C, 3D, 3F, 3G, 4A, 4B, 4D, 4E, 4F, 4G, 4H, 4I, 4J, 5B, 5C, 5D, 5E-H, 6A-C, 6, 7E, 7H. While abnormally long cilia might indeed be irrelevant to assess some of the basic properties of the used reagents, presented in Figs. 1F, 1G, all other results should be reevaluated.

We agree that the overexpression of the Ca²⁺ indicator causes as lengthening of the primary cilia, something that has been noted in previous publications using similar tools. We have now also quantitatively evaluated this effect and it shows that biosensor expression induces a 50% increase in cilia length which is also accompanied by a slight change in overall morphology, with a larger fraction of cilia presenting with dilated or swollen tips (Suppl. Fig. 1E, F). Importantly, the overexpression did not alter the distribution of GABAB1 receptors, which were still confined to an approximately 3 μ m long compartment at the cilia base. What was not so clear in the previous version of the manuscript is that naïve (non-transduced) islet cells have cilia that are both long and display a range of morphologies. We have now quantified this and presented the data in Fig. 2G-I. This data (all immunostainings of naïve islets) shows that there are cells that have two cilia, cells that have cilia with dilated or swollen tips and also examples of cilia-cilia contacts. We therefore believe that the overexpression of the biosensor amplifies an already existing heterogeneity in islet cell cilia morphology. To further investigate to what extent biosensor expression may interfere with cilia signaling, we have correlated the cilia Ca²⁺ response to biosensor expression and cilia length and do not find evidence that the sensor expression or length of cilium has an impact (Fig. 7G-I and Suppl. Fig. 2H). In addition, we have also repeated some experiments using an unrelated biosensor (G-GECO1 targeted to the cilium via 5HT6-receptor from Su et al, 2013) without observing any difference from the results obtained with Smo-GCaMP5G (Suppl. Fig. 3; Fig. 5E-G; Fig 6K,N; Fig 7O-R). We also thank the reviewer for pointing out the missing scale bars, which have now been added to all micrographs.

• Supplemental Figure 1K showcases the difficulties with the experimental approach, as the calcium sensor fused to Smo does not show recovery after photobleaching, which is well documented for virtually all cilia membrane proteins that are not part of ciliary

subcompartments, such as the transition zone. This includes ciliary GPCRs, such as Smo or SSTR3 (Ye et al., 2013).

We agree that this is contrasting previous studies. We do not know the reason for this (there was no correlation between mobility and expression level in our system). Please note that the mobility of plasma membrane-localized Smo was restricted to the same extent as that of cilia-localized Smo (shown in suppl. Fig. 2). We are performing the experiments using TIRF microscopy, so the cell bodies, and to some extent cilia, that we image are in direct contact with the glass coverslip. It is possible that the physical tethering of cells and cilia to the glass coverslip has an influence on the mobility of the Smo-based Ca²⁺ indicator. We believe that this works to our advantages, as this immobilization greatly facilitate the detection of local Ca²⁺ concentrations changes. Cilia FRAP analysis was also performed at the cilia tip, which is a special compartment where Smo mobility is lower (<https://www.ncbi.nlm.nih.gov/pmc/articles/PMC4500289/>) and this may also have contributed to the observed restricted diffusion. It is also important to note that the overall movement of receptors is not precluded in our experimental model system, since we see clear mobilization of GABAB1 receptors in response to agonist addition (Fig. 1K).

• Since the presented data are conflicting with a number of previous studies (see above) it is not sufficient to base all interpretations on a single technique that investigates extremely long cilia, which represent artifacts of the experimental approach.

We do not have reason to believe that the lengthening of the cilia caused by receptor overexpression influences the Ca²⁺ response in the cilia (see answers above). For example, we find that the size of both the GABAB1 compartment at the cilia base (Fig. 1E-G) and the size of the carbachol-induced Ca²⁺ microdomain at the cilia base are quite uniform and show only weak correlation to cilia length. We are also not aware of any other technique for measuring Ca²⁺ inside cilia. Traditional Ca²⁺ dyes or soluble genetically encoded sensors do not accumulate inside cilia, so targeting by some means is required. We have repeated some experiments using an alternative cilia-targeting sequence and Ca²⁺ indicator (5HT6-G-GECO1; Su et al, 2013), and could confirm observations made with SMO-GCaMP5G-mCh.

• The comparisons of cilia with their respective cell bodies appears extremely difficult in the light of the immensely long cilia, such that from the presented data it is unclear whether one can trace back an individual cilium to the cell body it emerges from (see Figs. 1I-K, 2A,B,D,E,G,I, Figs. 5 and 6). This criticism is showcased in Fig 4C, where the extremely long cilium seems next to a cell body, but whether they are connected is uncertain. Along the same line, an assessment of tip vs. base localization seems extremely difficult without appropriate markers (see Figs. 2, 4C, 5, S3).

The reviewer is correct; it is very difficult to accurately trace a cilium to the corresponding cell body. We show in the manuscript examples where we are certain that the cilia and cell body are connected. This was particularly important for the cross-correlation analysis performed in Fig. 8. However, in most cases we analyze the average of the Ca²⁺ response in all cilia and all cell bodies within the islet and do not attempt to find matching pairs. For uncaging experiments shown in Fig. 7J-M, it was fairly easy to connect cell bodies and cilia, as the photo-release of Ca²⁺ in one cell body only propagated into the cilium to which it was connected. As for the tip vs base distribution, we have now performed immunostainings of GABAB1 receptors and gamma-tubulin (centrosomal marker) to show that the GABAB1 receptors always accumulate at the cilia base (Fig. 1B).

• Changes observed in cilia subdomains are very hard to interpret since the "base" area is not well specified and often appears as long as a cilium of physiological length, for example see Fig. 2F.

We have now quantified the base region in our Ca²⁺ recordings based on the cilia response to carbachol (which occurred selectively at the cilia base). We find that the size of the cilia base (i.e. the region of local Ca²⁺ increase) is around 2-3 μm (i.e. shorter than most mammalian cilia), and the size of this compartment is not strongly influenced by the length of the cilium (Fig. 7G-H).

• Some changes in fluorescent signal intensities are somewhat ambiguous, as in presented micrographs often very long cilia are located in part above a cell body (or other structure). Often case, increases in cilia intensities are only visible directly above other structures that also show a change in fluorescence intensity (see Figs. 2C, 2F top middle, 2Hb).

This is also a valid point. Since recordings are done with TIRF microscopy, which in our case has an evanescent wave penetration depth of around 65 nm, a 200 nm diameter cilium would essentially prevent much of the signal from underlying cells to interfere with the signal from the cilium. Therefore, we believe that most of the signal we detect at regions where the cilia overlap with cell bodies emanate from the cilia. This is also supported by the calculations in Suppl. Fig. 2.

• There is only circumstantial evidence for a Ca²⁺ extrusion mechanism.

Our conclusions that the extrusion at the cilia base was dependent on the NCX relies on the findings that extrusion is accelerated when the cell is depolarized and that it is impaired in the presence of amiloride. To give further support to the idea of enhanced extrusion in primary cilia, we performed experiments in Na⁺-free conditions. Removal of Na⁺ resulted in reduced Ca²⁺ extrusion at the cilia base following stimulation with carbachol (Suppl. Fig. 4A, B). Na⁺-removal also counteracted the accelerated Ca²⁺ extrusion observed in depolarized cells (Fig. 7L,M). In addition, we have now performed in-depth analysis of Ca²⁺ kinetics during spontaneously generated Ca²⁺ flashes (Fig. 7O-Q and Suppl. Fig. 4) and found that the rate of Ca²⁺ extrusion is different along a cilium and highest at the cilia base. Because this is still not proof that the extrusion is mediated by NCX, we have removed this statement in the manuscript and instead discuss it in a broader context of Na⁺-dependent extrusion. In addition, we also believe that buffering may play a role in isolating the cilium from the cytosol. We now show in Fig. 7N that expression of a cilia-localized Ca²⁺ binding protein reduces the ability of Ca²⁺ to equilibrate between cytosol and cilium.

Minor points:

• Fig. 2D: unlike stated in the text, the effect on cytosolic Ca²⁺ seems more pronounced than the effect on ciliary Ca²⁺.

The response in the entire cilia (which is what we show) is smaller than that of the cell body, but the response at the cilia base is larger (see figure to the left). We have now made changes to the figure to make this clearer (Fig. 7).

• Figs S4 shows quantifications of fluorescence signals relative to the soma in μm , however, as cilia will have different lengths, a relative position (relative to the full length, as in Figs. 7E and 7I) seems more appropriate than a mere distance.

We agree that this is a more appropriate way to analyze the data. In this particular data set we had not co-immunostained for a general cilia marker, making it difficult to perform such analysis. We therefore now provide separate analysis of cilia length in islets immunostained for acetylated tubulin and treated in the same way (10 minutes with 100 nM GABA). As can be seen from the analysis (Suppl. Fig. 1H,I) this treatment is without effect on cilia length but has a clear effect on the distribution of GABAB1 receptors (Suppl. Fig. E-G).

• Fig. S1D: unclear whether the structure shown is part of a cell, a larger area should be shown.

This picture has been replaced with one showing an entire islet (new Suppl. Fig. 2C).

• The topology of Smo is nicely confirmed, but the reasoning is unclear. Smo topology is well known. Here, Smo topology has been determined for a transgene that shows a very different domain arrangement than the calcium sensor used (Fig. S1).

This is a good point. We have removed this data from the manuscript.

• The $\text{Na}^+/\text{Ca}^{2+}$ exchanger is not a direct target of amiloride. Amiloride targets sodium channels (and at higher concentration Na^+/H^+ exchanger). If this is known to affect the $\text{Na}^+/\text{Ca}^{2+}$ exchanger in beta cells, the original publication should be referenced.

See also comment above. We use a high concentration of amiloride that likely also partially blocks NCX, and now also show that removal of extracellular Na^+ has a similar effect as addition of amiloride (Fig. 7L,M). We have not been able to provide further support for the involvement of NCX and have removed such statements from the manuscript. We instead conclude that extrusion is Na^+ -dependent, sensitive to membrane potential and to amiloride. We have also included a more detailed analysis of Ca^{2+} decay kinetics in cilia that exhibit spontaneous Ca^{2+} activity that shows that extrusion is fastest close to the cilia base (Fig. 7O-R and suppl. Fig. 4)

• The finding that only GABA-B1 could be identified in cilia is exciting and surprising, as the functional receptor is believed to consist of a heterodimer of B1 and B2 subunits. If a functional B1 receptor homomer was reported in literature, please give the reference.

There are examples where GABAB1 receptors function independent of GABAB2 receptors. For example, GABAB1 receptors have been found to desensitize TRPV1 channels in a GABA-dependent, GPCR-independent manner (PMID: 25679765). It is also worth noting that the GABAB1 receptor is much more ubiquitously expressed in human tissues than the GABAB2 receptor (human protein atlas, search terms GABBR1; GABBR2) and the same is true in pancreatic islets (PMID: 32778664), where GABAB2 receptor expression is barely detectable while GABAB1 receptors are abundantly expressed. From an evolutionary stand point, the GABAB1 receptor also emerged before GABAB2 (PMID: 23266985). This is now discussed in the paper.

- All micrographs should contain scale bars.

We agree! This has been fixed.

Reviewer #3 (Comments to the Authors (Required)):

Sanchez et al. measure calcium ions inside primary cilia of beta-cells of isolated pancreatic islets. They report that calcium ions are regulated in these primary cilia independently from the contiguous cytosol of the same cells. They also identify GABA-B1 receptor to be exclusively expressed in these primary cilia and contribute to calcium increase also exclusive to the primary cilia. This unique regulation is achieved with the help of Na/Ca exchanger, revealed by pharmacological inhibitor experiments. The manuscript is generally written well and experimental designs are mostly reasonable. There are interesting observations of potential importance, along with some technical advances. However, this reviewer could not be excited about the findings for the following reasons:

1. Weak rationale: "Functional studies of the mammalian primary cilium have primarily been carried out in cell lines, with the exception of pioneering work conducted with isolated olfactory neurons". This sentence describes their motivation to perform calcium imaging using "primary" preparations rather than cell lines. However, there are several points that do not make sense. First, the authors may have forgotten citing a work by Mizuno et al. (PMID: 32743070), in which they not only performed calcium imaging in isolated mouse tissues, but also elegantly demonstrated the physiological significance of such cilia calcium.

While most studies on cilia Ca²⁺ has been done in cell lines, there are studies on primary cell and tissue preparations other than the ones we cited (for example Mizuno et al). We have now rephrased this sentence to: "Studies of mammalian primary cilia Ca²⁺ signaling have largely been carried out in cell lines, with important exception being groundbreaking work on isolated primary olfactory neurons^{38,39}, and more recently also on embryonic tissues⁴⁰."

Second, there seems to be no discussion of whether and how different sample preparations (primary cultures vs. cell lines vs. intact animal) affect calcium signaling inside primary cilia. It is also puzzling to learn that the authors used MIN6 cell lines to validate the findings from islet experiments. Third, it remains unclear whether the present isolated islets is sufficiently physiologically relevant for functional studies of beta cells. Thus, the claimed significance of using primary culture does not seem to be adequately justified.

The isolated islets are the gold standard for studying beta cell function and has been so for many decades. It preserves much of the cytoarchitecture, although islets become devascularized and de-innervated in culture. The functionality of the islets in culture can be exemplified by the fact that human islets kept in culture can be successfully transplanted to patients with type-2 diabetes to treat the disease (the islets become rapidly revascularized and start releasing insulin to the circulation). An alternative would have been to disperse the islets into single cells, but this requires mechanical disruption which in our hands damages the cilia. The MIN6 cells were used to confirm that what we observe in islets were responses from beta cells and not from other islet cell types. We have now added a clarifying sentence in the Methods section "These cells were used as a complement to the primary mouse islets to confirm that observations made in islets cells corresponded to responses from β -cells."

2. Lack of functional characterization: While there are interesting observations, it is not clear how any of these are linked to physiological functions of islets such as insulin secretion. Therefore, this reviewer could not judge physiological significance of these observations.

We agree that the manuscript to some extent is descriptive and that we do not directly provide a role of the described ciliary GABA signaling pathway. However, we think that several of our observations are of physiological relevance that extends beyond that of the pancreatic beta cell:

- 1) The identification of a diffusion barrier for Ca²⁺ at the base of the cilium in excitable cells have important implications for the understanding of cilia signaling and is a prerequisite for the generation of cilia-intrinsic Ca²⁺ signaling. This show that the cilium, similar to other small structures like the dendritic spines, are equipped with a machinery that enables physical insulation without a true physical barrier.
- 2) The finding that GABAB1 receptors localize to the cilium where they participate in non-canonical signaling (GABAB2 receptor-independent) that involves the initiation of local Ca²⁺ signaling. Given the ubiquitous expression of GABAB1 receptors (much broader than GABAB2), the action of GABA locally in the cilium may represent a previously unknown, conserved route of action for this versatile messenger.
- 3) In the revised version of the manuscript, we now also show an interplay between cyclic nucleotides and cilia Ca²⁺ signaling (Fig. 5). Using light-activated adenylyl and guanylyl cyclases to selectively produce cAMP and cGMP, respectively, we find that the former is without effect on cilia Ca²⁺ whereas cGMP can induce Ca²⁺ signaling in the cilium but not in the cell body. We also show that atrial natriuretic peptide, a hormone that is known to modulate insulin secretion from beta cells (PMID: 28864549), elicits distinct Ca²⁺ signaling in the cilium but not the cell body. cGMP likely acts via cyclic nucleotide-gated ion channels, and we observe CNGA3 channels in the cilium of both mouse islet cells and MIN6 cells. The cyclic nucleotides appear to diffuse freely between cytosol and cilium (production of cGMP in the cytosol elicits Ca²⁺ responses in the cilium), which is in contrast to Ca²⁺, which we show have a very limited ability to move between the two compartments.
- 4) We also now show in Fig. 1L that activation of GABAB1 receptors results in enrichment of the well-characterized cilia receptor Patched in primary cilia of islet cells. Although we don't know if this affects hedgehog signaling, it shows that there may be crosstalk between GABA receptors and other signaling pathways in the cilium.

3. Potential pitfall: As the above Mizuno et al. clearly demonstrated, confusion in the cilia calcium field originates at least partially from trivializing the importance of binding affinity of biosensors against calcium ions. Little to no detectable calcium signal does not mean that there is no calcium increase. It is simply non-detectable with the sensor used. A biosensor with a smaller K_d (i.e., higher affinity) could detect the otherwise pseudo-negative signal. The authors tested only one sensor, namely GCaMP5G, and discussed obtained data qualitatively, which is prone to misinterpretation.

We agree that biosensor properties are important when assessing cellular Ca²⁺ signaling. We have repeated key experiments using a second Ca²⁺ indicator, G-GECO1 (see e.g. Suppl. Fig. 3 and 4 and Fig. 7O-R). GCaMP5G has a K_d of around 450 nM, while G-GECO1 has a K_d of around 750 nM. We do not detect any difference in cilia Ca²⁺ response between islet cells expressing the two different sensors. We agree that it might be interesting to use a sensor with even higher affinity, but as can be seen from our in situ calibration of the indicator in Fig 3B-C, we are able to detect a clear increase in fluorescence with a step increase in Ca²⁺ from 100 nM to 300 nM inside the cilium. This is already a very small

increase, and most Ca^{2+} regulated processes require micromolar Ca^{2+} . We therefore believe that the indicator we used is able to report Ca^{2+} within the relevant physiological range.

Minor: Main texts describing Fig. 4D-I are apparently misaligned with figure numbers.

This has now been fixed (Fig. 8 in current version of the manuscript).

June 13, 2022

Re: JCB manuscript #202108101R-A

Dr. Olof Idevall-Hagren
Uppsala University
Medical Cell Biology
Husargatan 3
BMC, Box 571
Uppsala 75123
Sweden

Dear Dr. Idevall-Hagren,

Thank you for submitting your revised manuscript entitled "The β -cell primary cilium is an autonomous Ca^{2+} compartment for paracrine GABA signalling". The manuscript has been seen by the original reviewers whose full comments are appended below. We apologize for the delay in communicating our decision to you. While the reviewers continue to be overall positive about the work in terms of its suitability for JCB, some important issues remain.

You will see that while reviewers #1 and #2 express clear interest in the study, they feel more experiments are needed to support the conclusions. In particular, it would be essential that you address the following points with new data:

1- Localization of GABAR-B1 to cilia is novel and important. Figure 4B would need to be accompanied by measurements of fluorescence intensity and violin or scatter plots.

2- Clarify the pathway between GABAR-B1 and ciliary Ca^{2+} entry. Per reviewer #1, the way that drug treatments are conducted in the manuscript is not sufficiently rigorous to support conclusions. Careful dosage supplemented by siRNA studies is required to determine which Ca^{2+} channel is regulated by GABAR-B1.

3- Repeat key stainings (e.g. GABAR-B1) with another marker of cilia, e.g. ARL13B.

4- Acknowledge that overexpression of ciliary membrane proteins does affect the composition of cilia (see May et al., JCB 2021), and please present a cogent argument for why results are not affected by expression levels of the probes.

Our general policy is that papers are considered through only one revision cycle; however, given the interest in the study we are open to one additional short round of revision. Please note that I will expect to make a final decision without additional reviewer input upon resubmission, although reviewer input could be eventually requested.

Please submit the final revision within two months, along with a cover letter that includes a point by point response to the remaining reviewer comments.

Thank you for this interesting contribution to Journal of Cell Biology. You can contact me or the scientific editor listed below at the journal office with any questions, cellbio@rockefeller.edu.

Sincerely,

Maxence Nachury
Monitoring Editor
Journal of Cell Biology

Lucia Morgado-Palacin, PhD
Scientific Editor
Journal of Cell Biology

Reviewer #1 (Comments to the Authors (Required)):

I appreciate the effort of the authors to address my concerns and to try to improve the manuscript. I really like the new data regarding GABA-B1 receptor localization.

However, I am confused by the interpretation of the data added regarding CNG and L-type ion channels. The authors claim that albeit present in primary cilia CNG channels are not direct targets of GABA-B1. Instead L-type voltage gated Ca²⁺ channels are presumably downstream of the receptor. There are major problems with the data: All conclusions are based on pharmacology using inhibitors at high concentrations (100uM), potentially introducing non-specific or off target effects. Further, the CNG channels are disregarded as downstream targets based on data obtained with L-cis-diltiazem. However, this drug requires the CNGB1 subunit, whose cilia localization is not discussed. Lastly, the evidence for L-type voltage gated Ca channels in cilia is very weak. The paper the authors cite based its conclusions solely on antibody staining and lacks controls. To my knowledge this is the only publication that claims ciliary-localized voltage gated channels. I feel this data adds more confusion than clarity, thus I cannot recommend publication in its current form.

Reviewer #2 (Comments to the Authors (Required)):

In this revised manuscript, Sanchez and colleagues use genetically encoded fluorescent reporters to assess the calcium dynamics in primary cilia of pancreatic beta cells from isolated pancreatic islets. They report that:

- 1) In contrast to previous reports in other cell lines (Delling et al., 2013; DeCaen et al., 2013), primary cilia of pancreatic beta cells contain lower resting calcium concentrations than the beta cell cytoplasm.
- 2) Again in contrast to previous reports (Su et al., 2013; Delling et al., 2013), the primary cilium calcium environment in beta cells is functionally separated from the cytoplasm by calcium extrusion mechanisms, such that calcium signals from the cytoplasm are restricted to the ciliary base.
- 3) In agreement with this observation, spontaneous calcium spikes in beta cell primary cilia do not propagate to the cytoplasm and the calcium concentrations in cilia of pancreatic islets supposedly are isolated from the cell body.
- 4) These calcium spikes are mediated via GABA-B1 receptors.

In the revised version of the manuscript the authors additionally report that:

- 5) The calcium spikes in cilia require voltage-dependent calcium channel activation.

The topic is clearly of great interest to the bigger community of cell biologists interested in calcium signaling, beta cell and cilia biology. As far as I can judge the presented experiments, they appear very difficult and have been performed rigorously.

Some concerns raised from the initial submission have been addressed and the authors substantially revised the manuscript and have included a large amount of new data. However, a major point of criticism has not been adequately addressed: the authors are in large parts studying structures, which they assume are primary cilia based solely on one experimental strategy, i.e. immunofluorescence microscopy using antibodies directed against acetylated tubulin. Many aspects about the calcium environment the authors are describing are in stark contrast to previous studies and are somewhat reminiscent of calcium signaling in neurons. Moreover, acetylated tubulin is indeed present in axons and is an abundant component of the neuronal cytoskeleton. Therefore, this reviewer finds it important that the authors provide unambiguous data that confirm that the highly unusual "ciliary structures" they are investigating in isolated pancreatic islets are in fact primary cilia and not of neuronal origin.

Interestingly, the authors also provide solid evidence that many (very basic) findings in the proposed pancreatic islet "cilia" are different from a beta cell-based (MIN6) model, such as sub-ciliary localization of receptors and the presence or lack of a ciliary calcium barrier. Despite these clear and very fundamental differences between the MIN6 cell and the ex vivo islet "cilia", the authors utilize the MIN6 model to investigate the involvement of cyclic nucleotides, which seem to freely diffuse from the soma into cilia of MIN6 cells. Given the discrepancy of Ca²⁺ diffusion between soma and cilia of the MIN6 cell versus the pancreatic islet model (see Fig. 7A-C), it seems far-fetched to assume that the diffusion of cyclic nucleotides would be similar in both models. Hence, the significance of this data is unclear.

Major points (selected from previous review):

- Supplemental Figure 1H shows that the consequence of transducing pancreatic islets with their calcium sensor Smo-GCaMP5G-mCherry are massively long "primary cilia" (sometimes branched, see Figs. 2, S3) that are far from the physiological situation. Such unphysiological cilia are known to result from overexpression of cilia-localized membrane proteins, such as specific GPCRs, including Smo. Importantly, the high overexpression leads not only to extreme lengthening (as presented here) but most importantly also to an alteration of the ciliary protein composition, such that it is unclear whether the studied cilia actually contain the known or suggested proteins that may explain observed effects (ion channels and transporters). Therefore, all data resulting from the analysis of such artificially long cilia may be leading to false interpretations. This includes (although not all micrographs contain scale bars): Figs. 1H, 2C, 2F, 2H, 3B, 4C, 5A, 5I, 5J, 6D, 7C, 7D, 7F, 7G, 7I, S1C, S1L, S1K, S3C, S3D. Moreover, as the original micrographs are not presented, it is not unlikely that the following figures were also derived from artificially long and therefore non-physiological cilia: Figs. 1J, 1K, 2A, 2B, 2D, 2E, 2G, 2I, 3C, 3D, 3F, 3G, 4A, 4B, 4D, 4E, 4F, 4G, 4H, 4I, 4J, 5B, 5C, 5D, 5E-H, 6A-C, 6, 7E, 7H. While abnormally long cilia might indeed be irrelevant to assess some of the basic properties of the used reagents, presented in Figs. 1F, 1G, all other results should be reevaluated.

(Author response:)

We agree that the overexpression of the Ca²⁺ indicator causes as lengthening of the primary cilia, something that has been noted in previous publications using similar tools. We have now also quantitatively evaluated this effect and it shows that biosensor expression induces

a 50% increase in cilia length which is also accompanied by a slight change in overall morphology, with a larger fraction of cilia presenting with dilated or swollen tips (Suppl. Fig. 1E, F). Importantly, the overexpression did not alter the distribution of GABAB1 receptors, which were still confined to an approximately 3 μm long compartment at the cilia base. What was not so clear in the previous version of the manuscript is that naïve (non-transduced) islet cells have cilia that are both long and display a range of morphologies. We have now quantified this and presented the data in Fig. 2G-I. This data (all immunostainings of naïve islets) shows that there are cells that have two cilia, cells that have cilia with dilated or swollen tips and also examples of cilia-cilia contacts. We therefore believe that the overexpression of the biosensor amplifies an already existing heterogeneity in islet cell cilia morphology. To further investigate to what extent biosensor expression may interfere with cilia signaling, we have correlated the cilia Ca^{2+} response to biosensor expression and cilia length and do not find evidence that the sensor expression or length of cilium has an impact (Fig. 7G-I and Suppl. Fig. 2H). In addition, we have also repeated some experiments using an unrelated biosensor (G-GECO1 targeted to the cilium via 5HT6-receptor from Su et al, 2013) without observing any difference from the results obtained with Smo-GCaMP5G (Suppl. Fig. 3; Fig. 5E-G; Fig 6K,N; Fig 7O-R). We also thank the reviewer for pointing out the missing scale bars, which have now been added to all micrographs.

(Reviewer response:)

It is very helpful that the authors include more micrographs from non-transduced islets, however, I respectfully disagree that the authors can unambiguously conclude that they are describing cilia morphologies. I do believe that some of the structures identified are indeed cilia, as the gamma-tubulin co-staining in Fig.1B is quite convincing. However, these are all structures with "normal" primary cilia morphology and adequate lengths. According to literature, isolated islets should be de-innervated and de-vascularized, however, some structures presented in this study, especially the very long structures (examples stated previously) are reminiscent of neurites. Therefore, I strongly recommend that the authors provide further evidence to exclude that the acetylated tubulin positive structures of extra-ordinary length and morphology are not remnants of neurites present in the pancreatic islet preparations, and that they are in fact positive for other ciliary markers. This seems of particular interest, as GABA-B1 receptors have been described in pre-synapses, which would lead to very similar subcellular localizations as presented here.

- Since the presented data are conflicting with a number of previous studies (see above) it is not sufficient to base all interpretations on a single technique that investigates extremely long cilia, which represent artifacts of the experimental approach.

(author response:)

We do not have reason to believe that the lengthening of the cilia caused by receptor overexpression influences the Ca^{2+} response in the cilia (see answers above). For example, we find that the size of both the GABAB1 compartment at the cilia base (Fig. 1E-G) and the size of the carbachol-induced Ca^{2+} microdomain at the cilia base are quite uniform and show only weak correlation to cilia length. We are also not aware of any other technique for measuring Ca^{2+} inside cilia. Traditional Ca^{2+} dyes or soluble genetically encoded sensors do not accumulate inside cilia, so targeting by some means is required. We have repeated some experiments using an alternative cilia-targeting sequence and Ca^{2+} indicator (5HT6-G-GECO1; Su et al, 2013), and could confirm observations made with SMO-GCaMP5G-mCh.

(reviewer response:)

I respectfully disagree and have provided reasons to believe why receptor overexpression may influence a ciliary Ca^{2+} response. Targeting large quantities of proteins to cilia may not only alter the environment by their presence in cilia but also by occupying the transport machinery required to target other proteins efficiently to cilia.

I do agree that not many other techniques are available to assess Ca^{2+} inside cilia, except for patch-clamp experiments (DeCaen et al., 2013; Kleene & Kleene, 2017) that are extremely difficult to perform by non-experts. Using "alternative" targeting sequences that are also based on ciliary GPCR were not expected to provide substantially different results, other modes of targeting to cilia would have been preferable.

- Some changes in fluorescent signal intensities are somewhat ambiguous, as in presented micrographs often very long cilia are located in part above a cell body (or other structure). Often case, increases in cilia intensities are only visible directly above other structures that also show a change in fluorescence intensity (see Figs. 2C, 2F top middle, 2Hb).

(author response)

This is also a valid point. Since recordings are done with TIRF microscopy, which in our case has an evanescent wave penetration depth of around 65 nm, a 200 nm diameter cilium

would essentially prevent much of the signal from underlying cells to interfere with the signal from the cilium. Therefore, we believe that most of the signal we detect at regions where the cilia overlap with cell bodies emanate from the cilia. This is also supported by the calculations in Suppl. Fig. 2.

(reviewer response)

While the dimensions of cilia and evanescence waves in TIRF microscopy are clear, the data in the first version of the manuscript (Figure 2) clearly showed very strong signals from the cell bodies that clearly added to the signals in the proposed cilia, exactly where those "cilia" were present above the cell bodies (now Figure 7A and D; formerly Figs. 2).

It is somewhat concerning that the original data from Fig. 2F has been removed without comment and that generally the original data cannot be properly assessed by the reader. If the calcium responses "in cilia" are derived from similar cases, where the authors might be misled by signals from the cell body this may lead to gross misinterpretation. The authors should provide convincing examples of representative original data.

Major point on newly presented data:

- The newly added data on MIN6-based pseudoislets (Figs. 4D-E) nicely shows an involvement of GABA-1B receptors in baclofen-mediated responses, however, as -again- no original data is shown on the calcium reporter in these cells, it is not possible to compare the pseudoislets to the isolated islets from mice (or human), which would be very valuable information. As stated previously, from the few original micrographs provided it is hard to assess the presented data. Hence, the calcium response in cilia from MIN6-based pseudoislets, which are much shorter and hence will be found predominantly above the cell bodies, is prone to misinterpretation. In order to assess such data, this requires proper background subtraction. Whether this has been done is unclear from the methods section.
- With the exception of the well-established antibodies against acetylated tubulin and gamma tubulin, and the antibodies against GABA-B1 receptors, for which nice siRNA-based specificity controls have been performed in this study (see Figs. 4D, S1) the majority of the antibody reagents used in this study are not commonly used antibodies and proof of specificity is neither provided by the manufacturer nor by additional data within the presented study. This is probably not as important for well-known ciliary proteins, such as IFT88, SSTR3 or PATCHED, but a proper characterization should be provided for the anti-CNGA3 antibody to substantiate the presented findings.

Reviewer #3 (Comments to the Authors (Required)):

The authors properly addressed my initial concerns.

UPPSALA
UNIVERSITET

Olof Idevall-Hagren, PhD
Department of Medical Cell Biology
Uppsala University
SE-75123 Uppsala, Sweden

Phone: +46-184714426
E-mail: olof.idevall@mcb.uu.se

Uppsala, September 1st, 2022

Dear Dr. Nachury,

Thank you for the possibility to revise our manuscript entitled “The β -cell primary cilium is an autonomous compartment for paracrine GABA signalling”. As you will see below, we have addressed all comments raised by the reviewers, in most cases by performing additional experiments. We have primarily focused on the four points highlighted in the editorial assessment.

1) Localization of GABAB1 receptors to the primary cilium.

We have performed additional immunostainings on mouse islets using Arl13b which confirms our previous findings that the GABAB1 receptor localizes to the cilia base (Fig. 1D and quantified in Suppl. Fig. 1B). The specificity of the GABAB1 immunoreactivity is confirmed using a second primary antibody (Suppl. Fig. 1A) and by siRNA-mediated knockdown in MIN6 pseudoislets (Fig. 4D and Suppl. Fig. 1E). The effect of GABAB1-mediated Ca^{2+} signaling in cilia have also been quantified and it is now shown that the response to GABA is lost following GABAB1 receptor knockdown (Fig. 4E).

The localization of the GABAB1 receptor to the base of the cilium is supported by its partial overlap with acetylated tubulin, Arl13b and the Smo-GCaMP-mCh biosensor. In addition, the receptor positive compartment is adjacent to the centriole as revealed by γ -tubulin labeling. In all these experiments the subciliary confinement of GABAB1 is consistent, with an invariant length of 2 ~ 3 μm .

The identification of cilia by acetylated tubulin is undisputable given its colocalization with Patched, Smoothed, SSTR3, IFT88 stainings and the sensitivity of acetylated tubulin labeled structures to SAG treatments. There are no neurites left in isolated islets of Langerhans after keeping them for 3 days in culture and in any case, to the best of our knowledge, GABAB1 receptors has never been shown to localize to any neuronal compartment in the way we are showing in this manuscript.

2. Clarify the pathway between GABAR-B1 and ciliary Ca^{2+} entry.

We have performed additional experiments using a lower concentration of the voltage-dependent Ca^{2+} channel inhibitor verapamil (25 μM) and also including a more L-type-specific inhibitor, nifedipine. We first confirmed that all voltage-dependent Ca^{2+} channel blockers used in this study (verapamil, diltiazem and nifedipine) could block voltage-dependent Ca^{2+} influx into the cell bodies of islet beta-cells (Suppl. Fig. 4F). Next, we tested their ability to suppress GABA-induced Ca^{2+} signalling in the primary cilium of mouse islet cells. We find that both diltiazem and verapamil (also at the lower concentration) completely suppress Ca^{2+} influx, whereas nifedipine is less efficient. While nifedipine is selective for L-type VDCCs, the other two inhibitors are pore blockers and also target other members of the voltage-dependent Ca^{2+} channel family (Fig. 5L). The fact that nifedipine lacks effect in the cilium but strongly suppress L-type VDCC activity in the cell body speaks in favor of non-L-type VDCC involvement in GABA action in the

cilium. Consistently, immunostaining with KO-verified antibodies targeting the two L-type VDCCs CaV1.2 and CaV1.3 revealed little enrichment of the channels in the cilium (Suppl. Fig. 4G,H). In an attempt to further dissect the mechanism coupling GABAB1 receptor activation to VDCC activation, we tested if this was dependent on Gi-signaling by treating mouse islets with pertussis toxin. We find that treatment strongly suppress Gi signaling in the cell body (measured with a cAMP biosensor in mouse islet cells; Suppl. Fig. 4I-J) but is without effect on GABA-induced Ca²⁺ signaling in the primary cilium (Fig. 5M). Finally, we attempt to confirm the presence of active CNG channels in the primary cilium. We previously showed that the CNG channel blocker L-cis-diltiazem failed to suppress GABA-induced Ca²⁺ increases in the cilium, and it was pointed out by one of the reviewers that this drug required the presence of CNG-b subunits. We now confirm the presence of CNG-A3 subunits in the primary cilium by showing antibody specificity through siRNA-mediated silencing of CNG-A3 expression (Suppl. Fig. 4A-C). Immunostaining of CNG-b1 subunits did not reveal any localization to the primary cilium (Suppl. Fig. 4D). Consistent with the lack of CNG-b1 subunits, we find that L-cis-diltiazem also fails to suppress receptor-triggered (ANP) activation of CNG channels in the primary cilium (Suppl. Fig. 4E). Based on these results, we have modified our conclusion and now state that GABAB1 receptor activation in the primary cilium triggers Ca²⁺ influx through a non-L-type Ca²⁺ channel. Mouse islets beta-cells also express R-type and P/Q-type VDCCs and these are likely candidates. We believe that a rigorous identification of the channel involved in the GABA-induced Ca²⁺ signaling is highly relevant but will probably require large amounts of work and therefore it is more suitable for a follow up study.

3. Repeat key stainings with another cilia marker

We now show co-stainings of cilia (Arl13b) and GABAB1 receptors in mouse islets (Fig. 1D and suppl. Fig. 1B).

4. Acknowledge that overexpression of ciliary membrane proteins does affect the composition of cilia (see May et al., JCB 2021), and please present a cogent argument for why results are not affected by expression levels of the probes.

We now cite the above-mentioned publication and have also performed experiment to address this issue in our experimental model. We show that overexpression of the cilia-targeted biosensors is without effect on the distribution of GABAB1 receptors (Suppl. Fig. 2F) and that overexpression of two different cilia-localized Ca²⁺ indicators result in similar Ca²⁺ responses to stimuli (Suppl. Fig. 5A-D). Additionally, we now show that activation of the Hedgehog pathway using SAG results in exit of Patched from the cilium of both islet cells and MIN6 pseudo-islets overexpressing a cilia-targeted biosensor as well as in cells from non-transduced islets (Suppl. Fig. 5F) and that SAG is still able to promote cilia lengthening in beta cells overexpressing the cilia-targeted biosensor (Suppl. Fig. 5G). These results are summarized in the following paragraph: “*The overexpression of cilia-localized proteins is known to affect cilia composition and function*³². *Importantly, Smoothened overexpression in islet cells did not influence the distribution of endogenous GABA_{B1} receptors nor influence hedgehog signaling, determined as the exit of Patched from the cilium upon treatment with the Smoothened agonist SAG, nor prevent the SAG-induced lengthening of cilia (Suppl. Fig. 5).*”

Below follows detailed responses to all reviewers' comments. In the end I've also added a list of things that have been corrected in the current version.

Reviewer #1 (Comments to the Authors (Required)):

I appreciate the effort of the authors to address my concerns and to try to improve the manuscript. I really like the new data regarding GABA-B1 receptor localization.

However, I am confused by the interpretation of the data added regarding CNG and L-type ion channels. The authors claim that albeit present in primary cilia CNG channels are not direct targets of GABA-B1. Instead L-type voltage gated Ca²⁺ channels are presumably downstream of the receptor. There are major problems with the data:

All conclusions are based on pharmacology using inhibitors at high concentrations (100uM), potentially introducing non-specific or off target effects. Further, the CNG channels are disregarded as downstream targets based on data obtained with L-cis-diltiazem. However, this drug requires the CNGB1 subunit, whose cilia localization is not discussed. Lastly, the evidence for L-type voltage gated Ca channels in cilia is very weak. The paper the authors cite based its conclusions solely on antibody staining and lacks controls. To my knowledge this is the only publication that claims ciliary-localized voltage gated channels. I feel this data adds more confusion than clarity, thus I cannot recommend publication in its current form.

Reply: *We thank the reviewer for this comment and valuable input. We have now tried to address this through the following experiments:*

We included a structurally distinct L-type VDCC inhibitor (nifedipine, 10 uM) and show that this, as well as the previously used inhibitors verapamil (25 and 100 uM) and diltiazem (100 uM) effectively block voltage-dependent Ca²⁺ influx into the cell bodies of islet cells. This shows that the drugs are targeting the correct channels. Next, we tested to what extent they were able to prevent GABA-induced Ca²⁺ influx in the cilium. We find that while verapamil (low and high concentrations) as well as diltiazem effectively block the effect of GABA, nifedipine does not. Since nifedipine is the most selective L-type channel blocker, while the other drugs also target other VDCCs, we have changed the conclusion and now state that the GABA response is dependent on a voltage-gated Ca²⁺ channel, but not L-type channels (this is also supported by immunofluorescence staining for CaV1.2 and CaV1.3 subunits of L-type VDCCs). These results are shown in Fig. 5L and Suppl. Fig. 4A-H. In an attempt to understand the signal transduction from GABAB1-receptors to VDCCs, we tested whether this depends on Gi. Pertussis toxin treatments strongly suppressed Gi-induced lowering of cAMP in islet cell bodies (Suppl. Fig. 4I,J) but was without effect on GABA-induced cilia Ca²⁺ increases (Fig. 5M). This is in line with our observation that the G-protein binding subunit of the GABA-B receptor is absent from cilia. Finally, we attempted to clarify the role of CNG channels in controlling Ca²⁺ influx in cilia. We confirm by siRNA the specific localization of CNGA3 to the primary cilium. On recommendation from the reviewer, we also immunostained for CNGB1 subunits and find that these are excluded from the cilia. Along these lines, L-cis-diltiazem, which lacked effect on GABA-induced Ca²⁺ influx, was also unable to prevent cGMP-induced Ca²⁺ influx into primary cilia triggered by atrial natriuretic peptide (Suppl. Fig. 4E). It therefore seems that the CNG channels in the cilia differs from the typical channels with a 3A3:1B1 configuration found in e.g. photoreceptor cells. Importantly, CNGA3 can form functional homo-tetramers in heterologous systems.

Reviewer #2 (Comments to the Authors (Required)):

In this revised manuscript, Sanchez and colleagues use genetically encoded fluorescent reporters to assess the calcium dynamics in primary cilia of pancreatic beta cells from isolated pancreatic islets. They report that:

- 1) In contrast to previous reports in other cell lines (Delling et al., 2013; DeCaen et al., 2013), primary cilia of pancreatic beta cells contain lower resting calcium concentrations than the beta cell cytoplasm.
 - 2) Again in contrast to previous reports (Su et al., 2013; Delling et al., 2013), the primary cilium calcium environment in beta cells is functionally separated from the cytoplasm by calcium extrusion mechanisms, such that calcium signals from the cytoplasm are restricted to the ciliary base.
 - 3) In agreement with this observation, spontaneous calcium spikes in beta cell primary cilia do not propagate to the cytoplasm and the calcium concentrations in cilia of pancreatic islets supposedly are isolated from the cell body.
 - 4) These calcium spikes are mediated via GABA-B1 receptors.
- In the revised version of the manuscript the authors additionally report that:
- 5) The calcium spikes in cilia require voltage-dependent calcium channel activation.

The topic is clearly of great interest to the bigger community of cell biologists interested in calcium signaling, beta cell and cilia biology. As far as I can judge the presented experiments, they appear very difficult and have been performed rigorously.

Some concerns raised from the initial submission have been addressed and the authors substantially revised the manuscript and have included a large amount of new data. However, a major point of criticism has not been adequately addressed: the authors are in large parts studying structures, which they assume are primary cilia based solely on one experimental strategy, i.e. immunofluorescence microscopy using antibodies directed against acetylated tubulin. Many aspects about the calcium environment the authors are describing are in stark contrast to previous studies and are somewhat reminiscent of calcium signaling in neurons. Moreover, acetylated tubulin is indeed present in axons and is an abundant component of the neuronal cytoskeleton. Therefore, this reviewer finds it important that the authors provide unambiguous data that confirm that the highly unusual "ciliary structures" they are investigating in isolated pancreatic islets are in fact primary cilia and not of neuronal origin.

Interestingly, the authors also provide solid evidence that many (very basic) findings in the proposed pancreatic islet "cilia" are different from a beta cell-based (MIN6) model, such as sub-ciliary localization of receptors and the presence or lack of a ciliary calcium barrier. Despite these clear and very fundamental differences between the MIN6 cell and the ex vivo islet "cilia", the authors utilize the MIN6 model to investigate the involvement of cyclic nucleotides, which seem to freely diffuse from the soma into cilia of MIN6 cells. Given the discrepancy of Ca^{2+} diffusion between soma and cilia of the MIN6 cell versus the pancreatic islet model (see Fig. 7A-C), it seems far-fetched to assume that the diffusion of cyclic nucleotides would be similar in both models. Hence, the significance of this data is unclear.

Reply: *All experiments related to cyclic nucleotide signaling presented in Figure 5 were performed in intact mouse islets, so they are directly comparable to the cilia Ca^{2+} imaging experiments in the manuscript. As for the diffusion barrier, it also exists in MIN6 cell cilia – it's just more prominent in mouse islet cells because of the longer cilia (similar to what we observe in pseudo-islets generated from MIN6 cells). We don't think MIN6 cells are different, but that the formation of islet-like cell clusters has an impact on cilia growth. With regards to the distribution of GABAB1-receptors, the reviewer is correct that the localization differs between cells of mouse islets and MIN6 cells (grown both as monolayer and pseudoislet). Although we don't know the reason for this, one possibility is that the MIN6 cells secrete more GABA into the medium, which according to our observations would result in mobilization of the receptors into the cilium. Interestingly, the Ca^{2+} response to baclofen or GABA is immediate in MIN6 cell cilia but delayed by several minutes in mouse islet cells, perhaps reflecting the difference in receptor distribution under resting conditions.*

Major points (selected from previous review):

- Supplemental Figure 1H shows that the consequence of transducing pancreatic islets with their calcium sensor Smo-GCaMP5G-mCherry are massively long "primary cilia" (sometimes branched, see Figs. 2, S3) that are far from the physiological situation. Such unphysiological cilia are known to result from overexpression of cilia-localized membrane proteins, such as specific GPCRs, including Smo. Importantly, the high overexpression leads not only to extreme lengthening (as presented here) but most importantly also to an alteration of the ciliary protein composition, such that it is unclear whether the studied cilia actually contain the known or suggested proteins that may explain observed effects (ion channels and transporters). Therefore, all data resulting from the analysis of such artificially long cilia may be leading to false interpretations. This includes (although not all micrographs contain scale bars): Figs. 1H, 2C, 2F, 2H, 3B, 4C, 5A, 5I, 5J, 6D, 7C, 7D, 7F, 7G, 7I, S1C, S1L, S1K, S3C, S3D. Moreover, as the original micrographs are not presented, it is not unlikely that the following figures were also derived from artificially long and therefore non-physiological cilia: Figs. 1J, 1K, 2A, 2B, 2D, 2E, 2G, 2I, 3C, 3D, 3F, 3G, 4A, 4B, 4D, 4E, 4F, 4G, 4H, 4I, 4J, 5B, 5C, 5D, 5E-H, 6A-C, 6, 7E, 7H. While abnormally long cilia might indeed be irrelevant to assess some of the basic properties of the used reagents, presented in Figs. 1F, 1G, all other results should be reevaluated.

(Author response:)

We agree that the overexpression of the Ca^{2+} indicator causes a lengthening of the primary

cilia, something that has been noted in previous publications using similar tools. We have now also quantitatively evaluated this effect and it shows that biosensor expression induces a 50% increase in cilia length which is also accompanied by a slight change in overall morphology, with a larger fraction of cilia presenting with dilated or swollen tips (Suppl. Fig. 1E, F). Importantly, the overexpression did not alter the distribution of GABAB1 receptors, which were still confined to an approximately 3 μm long compartment at the cilia base. What was not so clear in the previous version of the manuscript is that naïve (non-transduced) islet cells have cilia that are both long and display a range of morphologies. We have now quantified this and presented the data in Fig. 2G-I. This data (all immunostainings of naïve islets) show that there are cells that have two cilia, cells that have cilia with dilated or swollen tips and also examples of cilia-cilia contacts. We therefore believe that the overexpression of the biosensor amplifies an already existing heterogeneity in islet cell cilia morphology. To further investigate to what extent biosensor expression may interfere with cilia signaling, we have correlated the cilia Ca^{2+} response to biosensor expression and cilia length and do not find evidence that the sensor expression or length of cilium has an impact (Fig. 7G-I and Suppl. Fig. 2H). In addition, we have also repeated some experiments using an unrelated biosensor (G-GECO1 targeted to the cilium via 5HT6-receptor from Su et al, 2013) without observing any difference from the results obtained with Smo-GCaMP5G (Suppl. Fig. 3; Fig. 5E-G; Fig 6K,N; Fig 7O-R). We also thank the reviewer for pointing out the missing scale bars, which have now been added to all micrographs.

(Reviewer response:)

It is very helpful that the authors include more micrographs from non-transduced islets, however, I respectfully disagree that the authors can unambiguously conclude that they are describing cilia morphologies. I do believe that some of the structures identified are indeed cilia, as the gamma-tubulin co-staining in Fig.1B is quite convincing. However, these are all structures with "normal" primary cilia morphology and adequate lengths. According to literature, isolated islets should be de-innervated and de-vascularized, however, some structures presented in this study, especially the very long structures (examples stated previously) are reminiscent of neurites. Therefore, I strongly recommend that the authors provide further evidence to exclude that the acetylated tubulin positive structures of extra-ordinary length and morphology are not remnants of neurites present in the pancreatic islet preparations, and that they are in fact positive for other ciliary markers. This seems of particular interest, as GABA-B1 receptors have been described in pre-synapses, which would lead to very similar subcellular localizations as presented here.

Reply: *We have now performed additional immunostainings of mouse islets using antibodies against GABAB1-R and Arl13b. As can be seen from the new Fig. 1D and Suppl. Fig. 1B, Arl13b, similar to acetylated tubulin, labels long primary cilia that also contain the GABAB1-R confined to the cell-proximal part. As can be seen from the quantifications, the average cilia length and size of GABAB-1 compartment are similar in these stainings and those previously shown for acetylated tubulin and GABAB1-R.*

• Since the presented data are conflicting with a number of previous studies (see above) it is not sufficient to base all interpretations on a single technique that investigates extremely long cilia, which represent artifacts of the experimental approach.

(author response:)

We do not have reason to believe that the lengthening of the cilia caused by receptor overexpression influences the Ca^{2+} response in the cilia (see answers above). For example, we find that the size of both the GABAB1 compartment at the cilia base (Fig. 1E-G) and the size of the carbachol-induced Ca^{2+} microdomain at the cilia base are quite uniform and show only weak correlation to cilia length. We are also not aware of any other technique for measuring Ca^{2+} inside cilia. Traditional Ca^{2+} dyes or soluble genetically encoded sensors

do not accumulate inside cilia, so targeting by some means is required. We have repeated some experiments using an alternative cilia-targeting sequence and Ca²⁺ indicator (5HT6-G-GECO1; Su et al, 2013), and could confirm observations made with SMO-GCaMP5G-mCh.

(reviewer response:)

I respectfully disagree and have provided reasons to believe why receptor overexpression may influence a ciliary Ca²⁺ response. Targeting large quantities of proteins to cilia may not only alter the environment by their presence in cilia but also by occupying the transport machinery required to target other proteins efficiently to cilia.

I do agree that not many other techniques are available to assess Ca²⁺ inside cilia, except for patch-clamp experiments (DeCaen et al., 2013; Kleene & Kleene, 2017) that are extremely difficult to perform by non-experts. Using "alternative" targeting sequences that are also based on ciliary GPCR were not expected to provide substantially different results, other modes of targeting to cilia would have been preferable.

Reply: *We have now performed experiments to address to what extent canonical cilia signaling is still functional in mouse islet cells expressing the cilia-localized Ca²⁺ indicator. As can be seen in Suppl. Fig. 5F-G, we show that although cilia are longer, they still respond to the addition of the Smoothened agonist SAG with exit of Patched (both in mouse islets and in MIN6 pseudoislets), and activation of Smoothened retains the ability to induce cilia lengthening in these cells. We agree that it would have been good to perform experiment with a sensor that has an alternative cilium targeting sequence, such as Arl13b, but we feel that we have already extensively characterized two different sensors with different targeting motifs and different Ca²⁺ sensors, and that a third one is not motivated.*

- Some changes in fluorescent signal intensities are somewhat ambiguous, as in presented micrographs often very long cilia are located in part above a cell body (or other structure). Often case, increases in cilia intensities are only visible directly above other structures that also show a change in fluorescence intensity (see Figs. 2C, 2F top middle, 2Hb).

(author response)

This is also a valid point. Since recordings are done with TIRF microscopy, which in our case has an evanescent wave penetration depth of around 65 nm, a 200 nm diameter cilium would essentially prevent much of the signal from underlying cells to interfere with the signal from the cilium. Therefore, we believe that most of the signal we detect at regions where the cilia overlap with cell bodies emanate from the cilia. This is also supported by the calculations in Suppl. Fig. 2.

(reviewer response)

While the dimensions of cilia and evanescence waves in TIRF microscopy are clear, the data in the first version of the manuscript (Figure 2) clearly showed very strong signals from the cell bodies that clearly added to the signals in the proposed cilia, exactly where those "cilia" were present above the cell bodies (now Figure 7A and D; formerly Figs. 2).

It is somewhat concerning that the original data from Fig. 2F has been removed without comment and that generally the original data cannot be properly assessed by the reader. If the calcium responses "in cilia" are derived from similar cases, where the authors might be misled by signals from the cell body this may lead to gross misinterpretation. The authors should provide convincing examples of representative original data.

Reply: *The reviewer is correct that we replaced figure 2F in the original version of the manuscript with one that better illustrates the response and matches the averaged data. We are very sorry that we forgot to mention this in our previous rebuttal. Although there may be a component of contamination from cytosolic or sub-plasma membrane fluorescence that influence the signal from the cilium that overlaps with the cell body, we still observe different Ca²⁺ kinetics at the cilia base (see Fig. 7Q,R) in favor of more rapid extrusion in the cilium. The potential contamination of somatic signal would make identification of such a*

difference more difficult, and hence we do not believe that we are simply observing cytosolic signals but rather that we are underestimating the difference between cell body and cilium.

Major point on newly presented data:

- The newly added data on MIN6-based pseudoislets (Figs. 4D-E) nicely shows an involvement of GABA-1B receptors in baclofen-mediated responses, however, as -again- no original data is shown on the calcium reporter in these cells, it is not possible to compare the pseudoislets to the isolated islets from mice (or human), which would be very valuable information. As stated previously, from the few original micrographs provided it is hard to assess the presented data. Hence, the calcium response in cilia from MIN6-based pseudoislets, which are much shorter and hence will be found predominantly above the cell bodies, is prone to misinterpretation. In order to assess such data, this requires proper background subtraction. Whether this has been done is unclear from the methods section.

Reply: *As indicated above, aggregation of MIN6 cells into pseudoislets results in cilia elongation. We have now included a micrograph of a pseudoislet expressing a cilia-targeted biosensor in Figure 4E.*

- With the exception of the well-established antibodies against acetylated tubulin and gamma tubulin, and the antibodies against GABA-B1 receptors, for which nice siRNA-based specificity controls have been performed in this study (see Figs. 4D, S1) the majority of the antibody reagents used in this study are not commonly used antibodies and proof of specificity is neither provided by the manufacturer nor by additional data within the presented study. This is probably not as important for well-known ciliary proteins, such as IFT88, SSTR3 or PATCHED, but a proper characterization should be provided for the anti-CNGA3 antibody to substantiate the presented findings.

Reply: *We have now performed siRNA-based specificity control for the CNGA3 antibody used in this study in MIN6 pseudoislets. It is shown in Suppl. Fig. 4A-C.*

Reviewer #3 (Comments to the Authors (Required)):

The authors properly addressed my initial concerns.

Corrections made to the manuscript:

When we went through all images, we noticed some errors that have now been corrected:

- 1) *Kymograph in Figure 6A has been replaced with a normalized version of the same kymograph.*
- 2) *Kymograph in Figure 6B has been replaced with a normalized version of the same kymograph.*
- 3) *Figure 6C incorrectly depicted a Ca²⁺ flash in a mouse islet beta cell and not a clonal MIN6 beta cell. The traces and normalized kymograph have been replaced to show data from a MIN6 cell.*
- 4) *Kymographs in 6D have been replaced with normalized kymographs (same kymographs as in the previous version are shown, but images have been normalized and background corrected).*
- 5) *Kymographs in 8A have been replaced with normalized kymographs (same kymographs as in the previous version are shown, but images have been normalized and background corrected).*
- 6) *Suppl. Figs. 1C-E have been updated (Suppl. Figs. 1C-D in new version). The previous version was based on 2 replicates. The version now is based on 3 replicates. The results are the same (GABAB1-R KD results in significant reduction in GABAB1-R immunoreactivity in the cilium of MIN6 cells).*

October 3, 2022

RE: JCB Manuscript #202108101RR

Dr. Olof Idevall-Hagren
Uppsala University
Medical Cell Biology
Husargatan 3
BMC, Box 571
Uppsala 75123
Sweden

Dear Dr. Idevall-Hagren:

Thank you for submitting your revised manuscript entitled "The β -cell primary cilium is an autonomous Ca^{2+} compartment for paracrine GABA signalling". Your paper was re-reviewed by the original reviewer #1, and they are now satisfied with revisions. Thus, we would be happy to publish your paper in JCB pending final revisions to address, via text edits, their remaining comment re: the effect of CNG-A3 knock down on ciliary Ca^{2+} influx. Please, also attend to the following formatting guidelines (see details below).

To avoid unnecessary delays in the acceptance and publication of your paper, please read the following information carefully. Please go through all the formatting points paying special attention to those marked with asterisks.

A. MANUSCRIPT ORGANIZATION AND FORMATTING:

1) Text limits: Character count for Articles and Tools is < 40,000, not including spaces. Count includes title page, abstract, introduction, results, discussion, and acknowledgments. Count does not include materials and methods, figure legends, references, tables, or supplemental legends.

2) Figures limits: Articles and Tools may have up to 10 main text figures.

***** Please note that main text figures should be provided as individual, editable files.**

3) Figure formatting:

Molecular weight or nucleic acid size markers must be included on all gel electrophoresis.

***** Scale bars must be present on all microscopy images, including inset magnifications. Please include scale bars in main Figs. 1E (inset magnifications), 2A (inset magnifications), 2D (inset magnifications), 2L (inset magnifications), 3D (inset magnifications), 4D-E (inset magnifications), 5I (inset magnifications), 9B (inset magnifications), 9F, and supplemental Figs. 1A, 2D, 2F, 3A, 3C-D, 5G, 9B.**

Also, please avoid pairing red and green for images and graphs to ensure legibility for color-blind readers. If red and green are paired for images, please ensure that the particular red and green hues used in micrographs are distinctive with any of the colorblind types. If not, please modify colors accordingly or provide separate images of the individual channels.

4) Statistical analysis:

***** Error bars on graphic representations of numerical data must be clearly described in the figure legend.**

***** The number of independent data points (n) represented in a graph must be indicated in the legend. Please, indicate whether N refers to technical or biological replicates (i.e. number of analyzed cells, samples or animals, number of independent experiments).**

If independent experiments with multiple biological replicates have been performed, we recommend using distribution-

reproducibility SuperPlots (please, see Lord et al., JCB 2020) to better display the distribution of the entire dataset, and report statistics (such as means, error bars, and P values) that address the reproducibility of the findings.

*** Statistical methods should be explained in full in the materials and methods in a separate section.

For figures presenting pooled data the statistical measure should be defined in the figure legends.

Please also be sure to indicate the statistical tests used in each of your experiments (both in the figure legend itself and in a separate methods section) as well as the parameters of the test (for example, if you ran a t-test, please indicate if it was one- or two-sided, etc.).

As you used parametric tests in your study (i.e. t-tests), you should have first determined whether the data was normally distributed before selecting that test. In the stats section of the methods, please indicate how you tested for normality. If you did not test for normality, you must state something to the effect that "Data distribution was assumed to be normal but this was not formally tested."

5) Abstract and title:

The abstract should be no longer than 160 words and should communicate the significance of the paper for a general audience.

The title should be less than 100 characters including spaces. Make the title concise but accessible to a general readership.

6) Materials and methods:

Should be comprehensive and not simply reference a previous publication for details on how an experiment was performed. The text should not refer to methods "...as previously described."

Also, the materials and methods should be included with the main manuscript text and not in the supplementary materials.

7) Please be sure to provide the sequences for all your primers/oligos and RNAi constructs in the materials and methods.

You must also indicate in the methods the source, species, and catalog numbers (where appropriate) for all your antibodies. Please include species for all your antibodies.

8) Microscope image acquisition:

The following information must be provided about the acquisition and processing of images:

- a. Make and model of microscope
- b. Type, magnification, and numerical aperture of the objective lenses
- c. Temperature
- d. imaging medium
- e. Fluorochromes
- f. Camera make and model
- g. Acquisition software
- h. Any software used for image processing subsequent to data acquisition. Please include details and types of operations involved (e.g., type of deconvolution, 3D reconstitutions, surface or volume rendering, gamma adjustments, etc.).

10) Supplemental materials:

There are strict limits on the allowable amount of supplemental data. Articles/Tools may have up to 5 supplemental figures. There is no limit for supplemental tables. Currently, you have 7 supplemental figures but that is fine as we could give you a bit of extra space.

*** Please note that supplemental figures and tables should be provided as individual, editable files.

*** A summary of all supplemental material should appear at the end of the Materials and Methods section (please see any recent JCB paper for an example of this summary).

11) eTOC summary:

*** A ~40-50 word summary that describes the context and significance of the findings for a general readership should be included on the title page. The statement should be written in the present tense and refer to the work in the third person. It should begin with "First author name(s) et al..." to match our preferred style.

12) Conflict of interest statement:

*** JCB requires inclusion of a statement in the acknowledgements regarding competing financial interests. If no competing financial interests exist, please include the following statement: "The authors declare no competing financial interests."

13) Author contribution:

*** A separate author contribution section is required following the Acknowledgments in all research manuscripts. All authors should be mentioned and designated by their first and middle initials and full surnames and the CRediT nomenclature should be used (<https://casrai.org/credit/>).

14) ORCID IDs: ORCID IDs are unique identifiers allowing researchers to create a record of their various scholarly contributions in a single place. At resubmission of your final files, please consider providing an ORCID ID for as many contributing authors as possible.

15) Materials and data sharing:

All animal and human studies must be conducted in compliance with relevant local guidelines, such as the US Department of Health and Human Services Guide for the Care and Use of Laboratory Animals or MRC guidelines, and must be approved by the authors' Institutional Review Board(s). A statement to this effect with the name of the approving IRB(s) must be included in the Materials and Methods section.

*** As a condition of publication, authors must make protocols and unique materials (including, but not limited to, cloned DNAs; antibodies; bacterial, animal, or plant cells; and viruses) described in our published articles freely available upon request by researchers, who may use them in their own laboratory only. All materials must be made available on request and without undue delay. Please, indicate whether the cell lines and reagents generated in this study have been deposited in public repositories. If not, please state that they would be made available to the scientific community upon request in the 'Data availability' section.

All datasets included in the manuscript must be available from the date of online publication, and the source code for all custom computational methods, apart from commercial software programs, must be made available either in a publicly available database or as supplemental materials hosted on the journal website. Numerous resources exist for data storage and sharing (see Data Deposition: <https://rupress.org/jcb/pages/data-deposition>), and you should choose the most appropriate venue based on your data type and/or community standard. If no appropriate specific database exists, please deposit your data to an appropriate publicly available database.

16) Please note that JCB now requires authors to submit Source Data used to generate figures containing gels and Western blots with all revised manuscripts. This Source Data consists of fully uncropped and unprocessed images for each gel/blot displayed in the main and supplemental figures. The Source Data files will be directly linked to specific figures in the published article.

Since your paper includes cropped gel and/or blot images, please be sure to provide one Source Data file for each figure that contains gels and/or blots along with your revised manuscript files. File names for Source Data figures should be alphanumeric without any spaces or special characters (i.e., SourceDataF#, where F# refers to the associated main figure number or SourceDataFS# for those associated with Supplementary figures). The lanes of the gels/blots should be labeled as they are in the associated figure, the place where cropping was applied should be marked (with a box), and molecular weight/size standards should be labeled wherever possible.

B. FINAL FILES:

Thank you for your attention to these final processing requirements. Please revise and format the manuscript and upload materials within 7 days. Please let us know if any complication preventing you from meeting this deadline arises and we can work with you to determine a suitable revision period.

Thank you for this interesting contribution, we look forward to publishing your paper in Journal of Cell Biology.

Sincerely,

Maxence Nachury
Monitoring Editor
Journal of Cell Biology

Lucia Morgado-Palacin, PhD
Scientific Editor
Journal of Cell Biology

Reviewer #1 (Comments to the Authors (Required)):

I appreciate the the amount of work the authors have invested in the revisions. I feel most of my concerns have been addressed. Although it would be nice to know the molecular identity of the ciliary channel mediating ciliary Ca²⁺ influx, I do agree with the authors that this may take time.

The only comment I have is that it would be very helpful for the audience if the authors could comment on the effect of CNG-A3 knock down on ciliary CA²⁺ influx. I may have missed it but I cannot find it in the figures. I am sure the authors have tried this.